 

# Research

biomechanics

bird flight, wing, suspension, turbulence, gust rejection

**Authors for correspondence:**
Richard J. Bomphrey
e-mail: rbomphrey@rvc.ac.uk
Shane P. Windsor
e-mail: shane.windsor@bristol.ac.uk

†These authors contributed equally to this work.
‡Present address: Airbus UK, Barnwell House, Filton, Bristol, BS34 7PA, UK.

# Bird wings act as a suspension system that rejects gusts

Jorn A. Cheney[1,†], Jonathan P. J. Stevenson[2,†], Nicholas E. Durston[2,‡], Jialei Song[1,3], James R. Usherwood[1], Richard J. Bomphrey[1] and Shane P. Windsor[2]

[1]Structure and Motion Laboratory, Royal Veterinary College, Hatfield AL9 7TA, UK
[2]Department of Aerospace Engineering, University of Bristol, Bristol BS8 1TR, UK
[3]School of Mechanical Engineering, Dongguan University of Technology, Guangdong, People's Republic of China

JAC, 0000-0002-9952-2612; JPJS, 0000-0002-3132-9824; NED, 0000-0002-4621-9468; JS, 0000-0003-3244-7766; JRU, 0000-0001-8794-4677; RJB, 0000-0002-4748-0510; SPW, 0000-0002-7597-4497

Musculoskeletal systems cope with many environmental perturbations without neurological control. These passive preflex responses aid animals to move swiftly through complex terrain. Whether preflexes play a substantial role in animal flight is uncertain. We investigated how birds cope with gusty environments and found that their wings can act as a suspension system, reducing the effects of vertical gusts by elevating rapidly about the shoulder. This preflex mechanism rejected the gust impulse through inertial effects, diminishing the predicted impulse to the torso and head by 32% over the first 80 ms, before aerodynamic mechanisms took effect. For each wing, the centre of aerodynamic loading aligns with the centre of percussion, consistent with enhancing passive inertial gust rejection. The reduced motion of the torso in demanding conditions simplifies crucial tasks, such as landing, prey capture and visual tracking. Implementing a similar preflex mechanism in future small-scale aircraft will help to mitigate the effects of gusts and turbulence without added computational burden.

## 1. Introduction

Birds routinely fly in gusty wind flows, and often in close proximity to obstacles such as terrain and buildings, conditions which challenge engineered air vehicles of a similar size [1,2]. The gusts encountered by birds can be of a similar magnitude to their flight speed [3,4] and are largely unpredictable. As such, deviations in flight path and/or orientation caused by gusts have the potential to disrupt critical behaviours such as landing, prey capture and visual tracking. Therefore, the capacity of birds to cope with gusts could limit their foraging success [5], the conditions in which they can forage, their nesting sites, as well as increase their flight costs [6]. The robust flight of birds observed in gusty unpredictable winds leads to the hypothesis that most flying birds must possess fast, stabilizing responses that reject gust effects.

Terrestrial animals are known to cope with perturbations through a combination of both active neurological control and passive stabilizing responses of the musculoskeletal system known as preflexes [7]. These responses can be difficult to separate unless the passive preflex occurs within the reflexive delay of the central nervous system (CNS). Birds may cope with gusts in a similar way, but the response times and dynamics by which they deal with aerodynamic perturbations are not known, as simultaneous measurements of both the bird's kinematics and the properties of the unsteady airflows they encounter are required. Most previous studies of bird flight mechanics (but see [8]) have taken place in steady laboratory conditions, such as corridors [9–11], or wind tunnels [12–17], or outdoors [18–21], where the

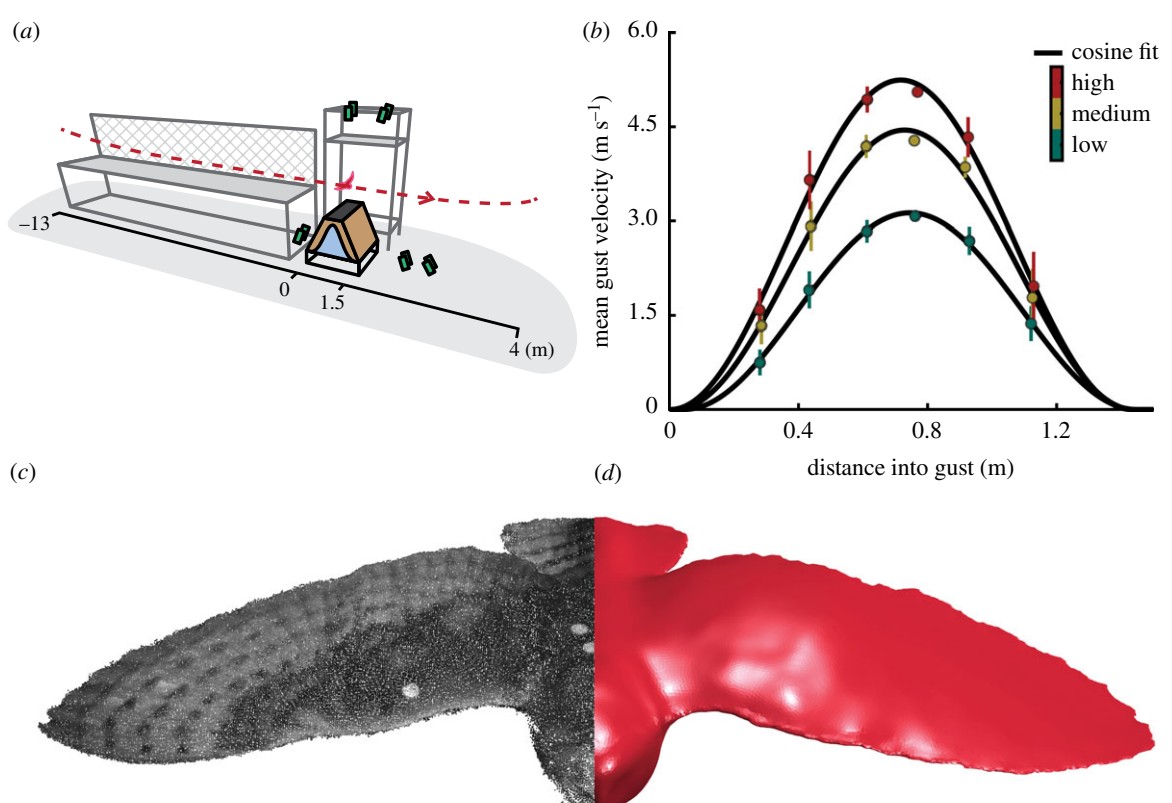

**Figure 1.** Measuring the response of a gliding bird to a vertical gust perturbation. (*a*) Schematic of the flight corridor with the trajectory of the bird (red dashed line), placement of high-speed video cameras for reconstruction (green), and gust generator (pale brown) with gust profile overlaid (pale blue). (*b*) Vertical gust profile along the flight path for three different gust intensities (spanwise mean ± s.e.m., *n* = 6 spanwise measures). (*c*) Reconstructed surface geometry of the barn owl as a point cloud and (*d*) CFD geometry, generated from the point cloud, used to estimate aerodynamic coefficients and expected gust impulse for a rigid model bird. (Online version in colour.)

moment-to-moment variation in the wind that the birds encountered could not be quantified.

Here, we investigate how birds cope with rapidly changing airflow by studying their flight through a controlled gust with a combination of high-speed video-based three-dimensional surface reconstruction, computational fluid dynamics (CFD) and quasi-steady glide modelling to understand how birds reject gusts through wing morphing, i.e. changing the shape and posture of their wings. Our experiment consisted of having a barn owl (*Tyto alba*) (figure 1*a*) glide through a range of fan-generated vertical gusts to impose a transient perturbation of different magnitudes (figure 1*b*) on the bird, with the strongest gust having a similar magnitude to the flight speed of the bird (7.72 ± 0.25 m s$^{-1}$, mean ± s.d., *n* = 13 flights). We measured rapid wing morphing and the perturbation impulse experienced by the torso and head using three-dimensional reconstruction (figure 1*c*) from 10 high-speed video cameras. From these measurements, we interpreted the effects of wing inertia on the glide path of the head and torso using a mass-calibrated computed tomography (CT) scan to compute the movement of the wings relative to the centre of mass of the whole bird. We computed the aerodynamic effects of wing movement by contrasting the glide results of the live articulated bird to the modelled glide path for a rigid bird whose aerodynamic polar was computed using CFD (figure 1*d*). We quantified the effect, and rejection, of the perturbation over time using mass-normalized impulse (change in velocity).

This comprehensive approach, using well-defined perturbation stimuli and high-precision measurements of the

moving anatomy, showed that the trajectory of the barn owl was relatively unaffected by even the strongest gust. Instead, the bird rejected the potentially negative consequences of the gusts by immediate, probably passive, wing elevation about the shoulder. This fast response was then followed by more complex changes in wing shape, potentially under active musculoskeletal control. Based on these observations we propose two independent mechanisms for gust rejection in birds: (i) an inertial mechanism, whereby the relative motion of the mass of the wings reduces the motion of the torso and head; and (ii) an aerodynamic mechanism based on changes in the orientation and shape of the wings, including decreasing their angle of attack, which reduces the lift generated. These two mechanisms separate the forces into those acting internally, the forces acting on the joints responsible for moving the segments of the bird relative to the centre of mass, and those acting externally, responsible for moving the centre of mass, and producing locomotion.

## 2. Results

### (a) Gust rejection kinematics

The bird consistently maintained a stable trajectory of its head and torso throughout all trials (electronic supplementary material, Movie S1) as if they were on a suspension system. The bird achieved this, in part, by two dramatic movements: an almost instantaneous change in wing elevation (figure 2*a*,*c*, electronic supplementary material, Movie S2), where rotation about the shoulder caused the wing to rise with the gust,

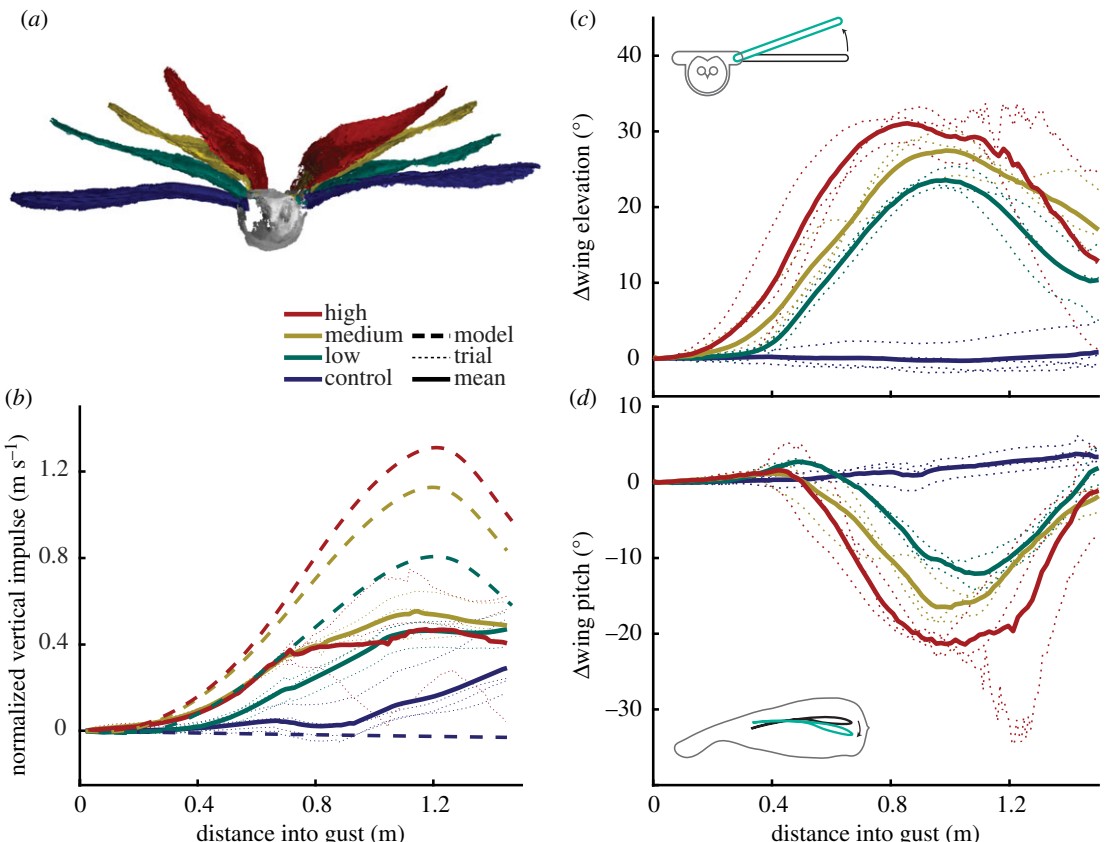

**Figure 2.** Gust rejection through dose-dependent changes in wing elevation and wing pitch angle. (a) Three-dimensional reconstructions of the bird 90% through the gust perturbation, wings colour-coded by gust intensity. (b) Vertical mass-normalized impulse of the centre of mass of the bird's torso (solid lines, mean response; light dashed lines, individual trials) was less than predicted by our glide model (heavy dashed line). (c) Wings elevated about the shoulder in dose-dependent response to gust intensity. The inset schematic depicts 20° elevation (teal outline) from the reference (black outline). (d) Wings pitched downward after increase in wing elevation. Inset schematic depicts −10° pitch (teal outline) from the reference (black outline). Bird is facing right. (b–d) Presented kinematic data encompasses the space occupied by the '1-cosine' gust, and does not extend beyond it. (c,d) Kinematics represent the average of the two wings. (Online version in colour.)

followed by a substantial reduction in wing pitch angle, where the wing rotated about its long axis (figure 2d). Both movements were important for gust alleviation and increased proportionally with gust intensity. Wing elevation delivered a rapid rejection of the gust owing to inertial effects, and aided the delayed wing pitch to decrease the lift generated by the wings. Together these reduced the peak vertical mass-normalized impulse (change in velocity) applied to the bird's torso and head by around half (42–63%, figure 2b) compared to that predicted by a quasi-steady glide model of a rigid bird (for the remaining results, references to the torso also apply to the head, as they were tracked as a whole).

## (b) Timing of inertial and aerodynamic components of gust rejection

We calculated the magnitude of gust rejection from the perspective of minimizing the perturbation applied to the torso. *Inertial gust rejection* describes the difference between the observed movement of the torso and that of the whole bird centre of mass, as calculated using the mass-calibrated CT scan (figure 3a,b); this rejection is owing to forces internal to the bird, i.e. passive or active elevation of its wing masses with respect to the torso. *Aerodynamic gust rejection* describes the difference between the observed movement of the centre of mass and that expected from the quasi-steady, rigid-bird glide model with CFD-derived polar (figure 3a,c); this

rejection is owing to changes in the external aerodynamic force. Both gust rejection metrics compare theoretical expectations for a rigid bird to observations of our live articulated bird. We quantified these gust rejection magnitudes by their impulse (electronic supplementary material). The internal impulse acting on the torso would equal that of the observed centre of mass of the whole bird if the wings were not mobile about the shoulder. The external impulse acting on the bird would equal that acting on the glide model if the bird did not reduce its aerodynamic force production through complex morphing.

Throughout the first half of the gust, the mass-normalized vertical impulse applied to the torso was substantially lower than that of the whole bird's centre of mass (figure 3a) because of the rapid upward motion of wing mass. Conservation of momentum dictates that, relative to the centre of mass of the whole bird, the upward momentum of the wings (19% of total mass) must be balanced by the downward momentum of the torso. This inertial effect reduced the impulse to the torso (figure 3b). The dose-dependent kinematics resulted in wing inertia rejecting a greater magnitude of the gust impulse with increasing gust intensity, but the proportion of the total expected impulse rejected by wing inertia did not discernably differ across gust intensities (electronic supplementary material, figure S1). At the instant of peak inertial rejection, wing inertia rejected $32 \pm 5\%$ (mean ± s.e.m. $n = 9$ flights) of the total expected gust impulse. At this same instance, there

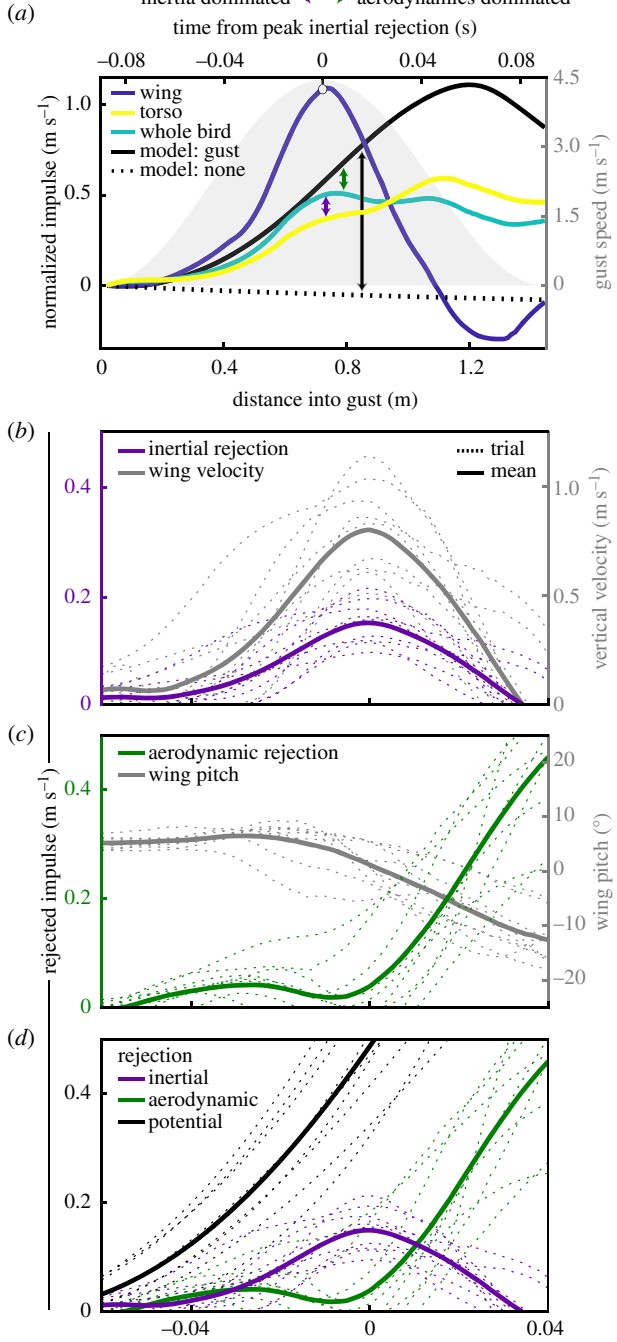

**Figure 3.** Inertial mechanisms reject the gust impulse applied to the torso before aerodynamic mechanisms take effect. (*a*) Representative medium-intensity gust trial displaying vertical mass-normalized impulse of wing, torso, whole bird and rigid-bird glide model with and without gust. Vertical arrows in the centre of the plot indicate how differences in paired parameters determine the components of gust rejection in the plots below (coded by colour). Mass-normalized impulse was computed from the onset of the gust. The white dot indicates the moment of peak inertial rejection. (*b–d*) Rejected mass-normalized impulse (left *y*-axis) versus time, aligned to the instant of peak inertial rejection and kinematic correlates (right *y*-axis). (*b*) Vertical wing velocity (grey) coincided with increasing rejection of the gust impulse by means of inertial mechanisms (purple). (*c*) Aerodynamic rejection mechanisms (green) remained at low levels until shortly past peak inertial rejection (*t* = 0), then increased coinciding with downward wing pitch (grey). (*d*) Inertial mechanisms (purple) rejected 32 ± 5% of the gust impulse applied (black, calculated from the glide model) prior to the initiation of aerodynamic mechanisms (green). (*b–d*) Trials from all gust intensities (excluding controls) are shown as dashed lines; solid lines show mean response. (*a*) Top arrows provide an indication of the transition from gust rejection dominated by inertial effects to that by aerodynamic effects.

was negligible aerodynamic rejection, accounting for only 6 ± 6% of the total rejection (mean ± s.e.m.; figure 3*c*), with the whole bird centre of mass velocity closely following that of the model of the rigid bird. Later, as the wing pitched downward, there was an increase in the difference between the measured vertical impulse applied to the whole bird centre of mass and to the rigid-bird glide model, indicating large aerodynamic-based gust rejection, which at the end of the gust reached 48 ± 4% of the potential impulse (figure 3*c*). It is, therefore, useful to think of these mechanisms as occurring in two distinct periods: the first inertia-dominated period occurs from the time of gust entry to peak upward velocity of the wings; the second aerodynamic-dominated period begins as the wings' upward momentum decreases. It is notable that the inertial period, while relatively brief, begins rapidly and rejects the gust impulse over the first 80.7 ± 9.3 ms (mean ± s.d., *n* = 9 flights) of the gust. Overall, the combination of inertial and aerodynamic mechanisms acted to reduce the effect of the gust on the body through two different means at two different time scales: the faster-acting inertial mechanism reduced the initial impulse applied to the torso, before the delayed aerodynamic mechanisms acted to reduce the impulse applied to the whole bird (figure 3*d*).

### (c) Distribution of mass and pressure along the wing

The mechanics that allow for inertial gust rejection depend upon the mass distribution as well as the spanwise centre of pressure; together they determine how each wing moves, and how the perturbation applied to each wing affects the body. The spanwise distribution of mass along the wing is heavily skewed proximally towards the shoulder, with a smaller local peak at the elbow (figure 4*b*). This mass distribution shifts the centre of percussion (elaborated upon below) inboard towards the shoulder relative to a uniform distribution. The centre of percussion was located 39.4% of the wing length from the shoulder (electronic supplementary material). The spanwise centre of pressure, as computed from CFD, was located 46.1% of the wing length from the shoulder. The position of the centre of pressure was stable with regards to changes in the angle of attack, moving outboard by 0.3% of wing length per degree increase.

## 3. Discussion

Prior studies of wing inertia have demonstrated that birds and bats enhance manoeuvring [12,22] directing their massive wings using active muscular control. The results presented here additionally demonstrate the benefits of wing inertia for gust rejection but, crucially, this mechanism does not require active control over the wings. In racquet and bat sports, the reaction force experienced at the hand when hitting the ball is reduced to zero if the ball is struck at a particular location on the racquet/bat called the 'sweet spot' [23], or the 'centre of percussion' [24]. The same phenomenon acts here (figure 4*a*), with each wing's centre of percussion coinciding closely with its centre of pressure during steady gliding (figure 4*b*). The location of centre of percussion is determined by the mass distributions of the wing. For the barn owl, it was estimated to be located just inboard (2.6 cm, or 6.7% semi-span) of the calculated centre of pressure from the CFD model. The exact magnitude of torso acceleration is determined by the mechanical properties of the shoulder joint,

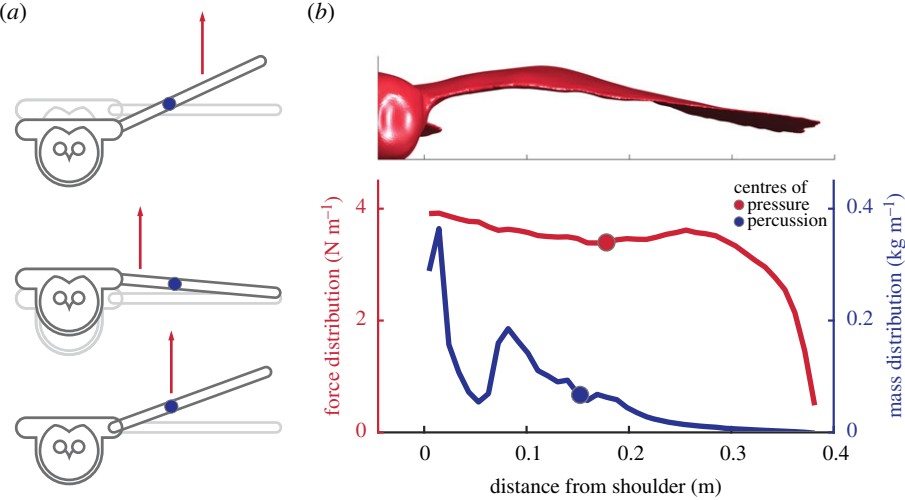

**Figure 4.** Aerodynamic force acting at the centre of percussion of the wing will initially only produce a rotation about the shoulder and cause no vertical displacement of the torso. (a) A force acting outboard of the centre of percussion (top) will rotationally accelerate the wingtip upward and accelerate the torso downward; a force inboard (middle) will act to rotationally accelerate the wingtip downward and accelerate the torso upward, while a force at the centre of percussion (bottom) will lift and rotate only the wing. (b) Mass distribution of the wing showing the alignment between centre of percussion and the spanwise centre of pressure during steady gliding flight from CFD simulations.

with increasing shoulder-joint stiffness shifting further outboard the point at which a perturbation produces no change in torso acceleration (see the electronic supplementary material). This suggests that birds may then be able to tune the dynamics involved with gust rejection by modifying the stiffness of their shoulder joint through muscle activity, potentially using a more compliant shoulder for atmospheric turbulence rejection (conferring straighter trajectories) or increasing shoulder stiffness for atmospheric energy harvesting.

## (a) A passive preflex

The inertial gust rejection mechanism discussed here functions as a preflex [25,26]. The mechanical properties of the wing and shoulder mitigate the fastest effects of the changing airflow, and only later does the CNS need to act to correct changes in trajectory. The inertial-based rejection reaches a maximum after a mean period of 80.7 ms, which is equal to the minimum behavioural reaction time measured in other bird species [27–29]. The proposed mechanism for gust rejection is intrinsically instantaneous, and is an effect of having a shoulder that allows the wings to elevate freely under increased load, which helps to explain how the entire inertial response can occur within the reflex delay. The speed of the event suggests that the CNS cannot modulate shoulder muscle activity sufficiently quickly and, therefore, that the passive mechanical properties of the shoulder joint and wing are responsible for immediate gust rejection. Consistent with this, wing shape is essentially constant from the start of the gust through the moment of peak inertial rejection (electronic supplementary material, Movie S3). After this inertial-based rejection, the pitch and shape of the wing begin to change more substantially (electronic supplementary material, Movie S2), altering the aerodynamic forces produced by the wings on a time scale that allows for potential CNS control.

## (b) Sensitivity to potential error

The conclusion that inertial rejection is both a rapid and strong gust rejection mechanism in birds is robust to

systematic error in the CFD-derived aerodynamic polar. At its peak, inertial rejection exceeded aerodynamic rejection regardless of whether we applied a 50% increase in drag across the drag curve, a 20% increase in lift to the linear portion of the lift curve, or a moderate 6.5% increase in lift across the entire lift curve (detailed in the electronic supplementary material). None of these changes affected the absolute magnitude of inertial rejection, but they affected the quasi-steady glide model, which in turn determined the relative magnitudes of inertial and aerodynamic rejection. The relative impact of inertial rejection and its insensitivity to our aerodynamic model provide confidence in the substantial role played by inertial effects in avian gust rejection. Further, we have conservatively estimated the effect of inertial rejection by ignoring the added fluid mass around the wings; accounting for the added mass would have further enhanced inertial rejection.

## (c) The delay in aerodynamic rejection

Why is aerodynamic rejection initially weaker than inertial rejection? Despite a multitude of possible aerodynamic rejection mechanisms, inertial rejection increases more rapidly than aerodynamic rejection. The most likely aerodynamic rejection mechanisms that could reduce the experienced vertical force/impulse include: (i) changes to wing posture, which can reject gusts by reducing angle of attack, and wing elevation, which tilts the resultant aerodynamic force vector inwards; (ii) changes to wing shape, where aeroelastic feather bending modifies wing camber and angle of attack, and wing retraction, which reduces wing area; and (iii) the dynamics of wing elevation, which reduces the relative velocity between the wings and the gust, and effectively reduces the local angle of attack of the wing, most notably at the tip.

Aerodynamic rejection did undergo an immediate rise, similar to inertial rejection, but it saturated before it had an appreciable effect (approx. 0.05 m s⁻¹; figure 3c) and maintained that saturated level until around the instant of peak inertial rejection. This saturation may have been a response to soft stall, a feature found in many bird wings and

observable in our simulated lift curve from CFD (electronic supplementary material, figure S5; [30]). Soft stall describes a feature of the lift curve, where the lift coefficient becomes insensitive to changes in the angle of attack, i.e. at high angles of attack, the curve forms a nominally flat horizontal line. The majority of the mechanisms discussed above reject gusts by means of reducing the lift coefficient through reductions in the angle of attack. We hypothesize that the delay in aerodynamic gust rejection persists until each wing's lift coefficient becomes sensitive to changes in angle of attack, i.e. the wing is no longer stalled. This difference in lift coefficient sensitivity may explain why aeroelastic feather bending, which confers nearly instantaneous load alleviation/gust rejection under small perturbations [31], did not have a more pronounced effect in this study. Future work could explore the relative contributions of inertial and aerodynamic mechanisms under weaker gusts that do not produce stall.

## 4. Conclusion

The hinged wings performed the same role as a suspension system in a terrestrial vehicle, by dramatically reducing the perturbation applied to the body. The concept of the shoulder joint acting as a hinged suspension system has a number of implications for wing design in both animals and engineered flyers. Gust rejection enabled by wings that can rotate about the shoulder in flight may explain why wing-locking mechanisms are rare, even among specialist gliders and soaring birds [32]. Balancing torque from the wing lift with muscular force from the pectoralis muscle allows rapid changes in wing loading to rotate the wing and reduce the disturbance experienced by the head and torso. This is advantageous as it reduces deviations of the flight path and stabilizes the torso and head, which simplifies tasks such as landing, prey capture and visual tracking. The effectiveness of inertial gust rejection depends not only on wing hinge properties, but also on the aerodynamic shape (centre of pressure) and mass distribution of the wing (centre of percussion) and could represent a trade-off in the evolution of wing shape and structure. Certain lift distributions minimize various forms of drag [33], but moderate deviation from these distributions often impose only marginal increases in drag. Deviation from an aerodynamically optimal lift distribution, as a means to shift centre of pressure, may present relatively small costs relative to the benefits of steadier, simpler and more robust flight. A suitably tuned, hinged-wing design could also be useful in small-scale aircraft. These vehicles often need to operate in turbulent conditions close to obstacles, where gust rejection is important to keep deviations of the flight path to a minimum as well as to help stabilize any sensors carried by the aircraft. Implementing a similar preflex mechanism in future small-scale aircraft would help to reject gusts and turbulence with reduced computational burden.

## 5. Material and methods

### (a) Bird

A captive-bred, adult, female barn owl (*T. alba*) was used in these experiments. The bird was trained to fly between handlers on command and was familiar with flying in loud and bright environments.

All work was approved by the Ethics and Welfare Committee of the Royal Veterinary College (URN 2015 1358) and the University of Bristol Animal Welfare and Ethical Review Body (UIN UB/15/070) and complied with all relevant ethical regulations.

### (b) Experimental setup

The experiments were conducted in a purpose-built indoor flight corridor at the Royal Veterinary College (Hatfield, UK). The corridor was 2 m wide, bounded by a structural wall on one side and framed mesh on the other, with a suspended floor to elevate the flight path of the bird and permit camera views from above and below within the gusted region. During each trial, the bird flapped along the corridor to gain speed before entering a smooth, steady glide prior to the gust. After negotiating the disturbance, it flared and landed at the receiving handler, having flown a total distance of approximately 17 m.

We imaged approximately 3.5 m of each flight through the gust using an array of 10 synchronized high-speed cameras, arranged in upper and lower sets (figure 1a), comprising pairs of either Photron SA3 (1 MP), SA-Z (1 MP) or WX100 (4 MP) models (Photron Europe Limited, West Wycombe, UK). Camera placement ensured that the bird's dorsal and ventral surfaces were always visible to a sufficient number of cameras to enable full surface reconstruction before, during and after the gust. Refer to the electronic supplementary material, figure S2 for typical views of the bird in flight. Cameras recorded at 1000 frames per second (fps) with a shutter speed of approximately 1/4000 s. Custom stroboscopic LED lamps were used to provide even, flicker-free exposure of the bird against the background scenery which, where possible, was covered in black material to facilitate automated masking of the bird during later image processing.

A motion-capture system (Qualisys AB; Göteborg, Sweden) tracked the motion of retro-reflective markers on the dorsal surface of the bird's head, torso, wing tips and tail tips. The system consisted of 12 cameras operating at 180 fps mounted on the support tower looking down onto the flight path of the bird. This system covered the region of the gust as well as a larger pre-gust region than the high-speed video cameras. The only markers used in this study were those on the torso, to estimate pre-gust acceleration.

### (c) Gust

The gust was generated by a 2 × 2 bank of 0.5 m diameter axial fans (MB50; Broughton EAP Ltd, Redditch, UK). The flow was directed through a convergent nozzle to form a columnar, vertical jet. The nozzle outlet (1.6 m wide, 0.6 m long) was situated at the far end of the corridor approximately 0.7 m below the mean flight height of the bird. Entrainment of the stagnant surrounding air caused the jet to spread out as it moved away from the nozzle increasing the size of the gust beyond the dimensions of the nozzle.

The bird flew through three gust intensities—low, medium and high—with three repeats for each condition, and additionally flew four control trials. In the control trials, the flow was diverted away from the nozzle and the nozzle exit was covered to ensure a consistent acoustic noise but no gust. Ordering was pseudo-random, except for the controls, which were carried out in groups of two at both the start and end of the session to test for pre-emptive corrections by the bird before entering the gust zone. This ordering minimized the risk of accidentally changing the orientation of the calibrated cameras when moving the fans to divert the flow.

A sonic anemometer (HS-50; Gill Instruments Ltd., Lymington, UK) was used to survey the gust velocity field across two planar horizontal grids of 30 points (5 × 6) spaced 0.45 m vertically apart, with the lower plane being 0.40 m above the nozzle exit. All flights passed between these planes, with all but one

within 0.2 m of the upper plane. Fitted peak intensity was similar between the two planes: peak speed of the lower plane was greater by 6%, 4% and 3% for low, medium and high intensity, respectively. Owing to the similarities in intensity and the proximity of flights to the upper plane, we considered the upper plane as representative of the perturbation experienced by the bird. An additional vertical transect was measured for the high-intensity gust to confirm that the centreline jet velocity remained consistent between the planes. All measurement points were sampled at 50 Hz for a minimum of 30 s, with those on the upper plane spaced more widely to capture the lateral spreading of the jet. Refer to the electronic supplementary material, figure S3 for the measured mean velocity fields. The upper plane measurements were used to characterize the gust for the glide model. For each gust intensity, the span-averaged vertical velocity distribution across this plane was fitted by a symmetrical '1-cosine' profile (figure 1b), which was defined by two parameters: gust length and peak velocity magnitude. The fitted gust length was 1.40–1.43 m for all intensities, and the peak velocities were 3.1, 4.5 and 5.2 m s$^{-1}$ for low, medium and high intensities, respectively.

### (d) Reconstructions

The three-dimensional surface geometry of the bird was reconstructed using commercial photogrammetry software (PHOTOSCAN v.1.3.5; Agisoft LLC, St Petersburg, Russia) and custom Python scripting. For each high-speed video frame, the bird was first masked from the images using median background subtraction. This mask was then refined by building a mesh of the bird using the camera views, filling any gaps and projecting the mesh back onto the images to ensure no points on the bird had been discarded during initial background removal. Common features were then automatically identified and matched between multiple views, which provided an initial sparse reconstruction of the bird when combined with camera calibrations. The sparse reconstruction served as a foundation for disparity map calculations between camera pairs. Finally, the masked bird was reconstructed from disparity maps and camera calibrations. The full process generated a three-dimensional point cloud of the bird in corridor-aligned coordinates for each video frame. Each cloud point was also assigned a greyscale value, based on the matched image pixels from which it was obtained. These greyscale values were used to filter out any spurious points (mainly dark edge noise) after processing. The surface reconstructions had a spatial resolution of approximately 100 points cm$^{-2}$ and a surface accuracy of $-0.68 \pm 1.67$ mm (mean ± s.d.; where 50% of the absolute error was less than 1.1 mm and 95% less than 3.5 mm; computed with CLOUDCOMPARE 2.6.2), based on the reconstruction of a fibre-glass bird model of a known geometry measured using a high accuracy laser scan (Romer Absolute Arm, RA-7525-SI, accuracy 0.063 mm). Negative accuracy implies that the known geometry was larger in one or more dimensions than the point cloud. Most likely, our primary error was along the vertical axis between the upper and lower camera sets, and our reconstruction estimated the lower and upper surfaces too close to one another.

Camera calibration involved three steps: (i) intrinsic calibration; (ii) individual extrinsic calibration of upper and lower cameras sets; and (iii) alignment of both sets' coordinate systems to the corridor reference frame. The intrinsic properties of each camera-lens pairing, including optical distortion, were calculated from 40 to 80 images of a large, flat checkerboard filling the camera's field of view. The extrinsic parameters of each camera set (camera positions, orientations and scale) were then calculated using images of a patterned board with corner targets used to define the absolute scale. As the pattern on the board could only be seen from a single set of cameras at a time, images of a T-shaped wand with spherical reference points were used to

align the upper and lower camera sets. Finally, the corridor reference frame was defined in relation to images of an L-shaped wand with spherical markers, directed along the flight path and levelled in the plane normal to gravity. For ease of reference, the data shown here has been shifted so the corridor coordinate system origin coincided with the position of the start of the gust.

### (e) Point cloud motion tracking

The motion of the point clouds was quantified by segmenting the cloud into torso, wing and tail sections using custom-written graphical user interfaces (Matlab, 2019b; The MathWorks Inc., Natick, USA). These sections were tracked by aligning them with a selected gliding pre-gust frame using an iterative closest point (ICP) alignment method (Matlab, 2019b; The MathWorks Inc., Natick, USA). Wing movements were then compared to a reference pose, with the wingtip pointing orthogonal to the body-axis and co-linear with the shoulder, and in a 'flat' state as estimated by a plane fitted to the wing (electronic supplementary material, figure S4). The ICP results quantified the movement as translation and rotation matrices. Rotation matrices were converted to Euler angles for wing movement description and computed in the order: sweep (Z), elevation (X') and pitch (Y"). Wing elevation is described relative to the horizontal plane, and wing pitch is described as the long-axis rotation of the wing after accounting for its swept and elevated orientation.

### (f) Mass properties

The mass properties of the bird were calculated based on CT-scan data from a naturally deceased barn owl of similar overall size. The cadaver weighed 296 g, and the live owl 319 g. The cadaver, with wings extended, was depth-scanned (LightSpeed RT16 CT scanner; General Electric, Boston, USA) to produce a three-dimensional voxel array in 16-bit greyscale. Calibration phantoms (Gamex 467, Sun Nuclear, Melbourne, Australia), scanned alongside the bird, supplied the necessary calibration for the conversion of greyscale values to tissue density and, in turn, mass distribution of the bird using the known voxel dimensions. The segmented torso and left wing from the scan were aligned with the point clouds to estimate the mass distribution of the bird in flight. We assumed that the inertial dynamics of the left wing were equal to those of the right wing. The CT and point cloud alignment was performed for a single typical glide posture to establish the mass and centre of mass position for each point cloud segment. Centre of mass position was then tracked subsequently when aligning point clouds using rigid-body transforms that allowed for rotation and translation. This approach was ideal for the duration that wing flexion was negligible (e.g. the time-course dominated by the inertial rejection mechanics. electronic supplementary material, Movie S3). However, the effect of wing shape change on the centre of mass was not completely ignored because we allowed the wing to translate. For example, when the elbow flexed, the rigid-body transform that best fit the flexed state of the wing moved the centre of mass proximally as expected.

### (g) Quasi-steady, rigid-bird glide model

A two-dimensional flight-path model for a hypothetical, rigid gliding owl under the influence of a '1-cosine' gust was used to estimate the effect of the wing motion of the bird on its trajectory. The model was free to accelerate both horizontally and vertically, but could not rotate, as the real bird displayed no tendency to pitch during the gust. The initial velocity was based on the path of the segmented torso, and acceleration from the motion-capture system. The motion-capture data covered a larger pre-gust region and provided a better estimate of acceleration than the point-cloud data. The equations of translational motion, with quasi-steady aerodynamic forcing, were solved in uniform time steps of 10 μs (for details see

the electronic supplementary material, Text). Aerodynamic coefficients (electronic supplementary material, figure S4) were determined by CFD simulations.

## (h) Aerodynamic data (computational fluid dynamics)

CFD modelling was used to generate aerodynamic force data for the quasi-steady glide model. Commercial software (Mimics, Materialise NV, Leuven, Belgium; and SPACECLAIM, v. 19.1, ANSYS Inc., Canonsburg, USA) was used to create a high-fidelity surface geometry from a point cloud of the owl in a typically neutral glide posture. This point cloud measurement was performed prior to the experiment. Comparisons of this single geometry to the configurations adopted by the bird are provided for: all trials, pre-gust (electronic supplementary material, Movie S4); and control trials, evenly sampled through the measurement region (electronic supplementary material, Movie S5). Together these demonstrate that the geometry is representative of the posture prior to the gust, and throughout the glide when no gust was present.

The fluid mesh was generated in ANSYS Mesh as a hybrid mesh (Ansys Inc., Canonsburg, USA). The simulation domain (domain size: $9000 \times 6000 \times 6000 \text{ mm}^3$) was discretized by approximately 15 million non-uniform volume elements. Two bodies of influence controlled mesh size: mesh size was 5 mm for the inner body of influence and 12 mm for the outer body of influence. Within the inner body, adjacent to the bird surface, 15 o-shaped inflation layers were used to resolve the boundary layer: the first layer thickness was 0.5 mm ($y^+$ approx. 10) and subsequent layers had a growth ratio of 1.2. Except for the inflation layer, we used an unstructured tetrahedron mesh to reduce the number of mesh elements and computing cost. Several images of the tessellated fluid mesh across a plane through the arm-wing are shown in the electronic supplementary material, figure S6. We found that if we further refined the near-surface mesh by decreasing the first layer thickness to 0.1 mm ($y^+$ approx. 3), there was only a 3% difference in lift.

The boundary conditions for the simulation constrained velocity, pressure and turbulence intensity. The velocity at the inlet was constrained to 7.88 m s$^{-1}$, and at the outlet followed the Neumann boundary condition, while pressure at both the inlet and outlet followed the Neumann boundary condition. The far-field boundary conditions, perpendicular to the flow, were constrained to be symmetric. Turbulence intensity was specified as 1% at the inlet.

The CFD simulations treated air as an incompressible and viscous flow. At the speed of the flights, Mach number was approximately 0.02, which suggests air compressibility was negligible. Turbulent effects though were not negligible. For viscous flow at Reynolds numbers for the gliding flights (approx. $8.8 \times 10^4$ as defined by mean chord length and flight speed), turbulent effects are significant. To fully simulate the turbulence would require direct numerical simulation (DNS) and a fluid mesh density that captures the Kolmogorov scale. As DNS is computationally impractical for bird flight, we approximated the behaviour of the fluid with the k-ω SST turbulence model in ANSYS FLUENT (v. 19.1; ANSYS Inc., Canonsburg, USA). The shear stress transport (SST) model uses a blending function which combines the merits of the standard k-ω model, for near-surface simulation and the standard k-ε model, for the domain away from the surface, which is appropriate for bird flight simulations that require a high-accuracy boundary layer. We compared the effects of different turbulence models

(Spalart-Allmaras, Reynolds Stress, k-ε standard and k-ω standard) on the computed values of lift and drag, and found all models were within 5% of each other for a 5° angle of attack simulation. The criteria for simulation convergence was determined by the unitless 'scaled' residuals of three quantities, when: the continuity residual reached less than $5 \times 10^{-3}$; the orthogonal velocity-component residuals reached less than $5 \times 10^{-7}$; and the turbulence kinetic energy reached less than $3 \times 10^{-4}$. Turbulence kinetic energy was typically the last criterion met. The 'scaled' residuals are determined by normalizing the absolute residuals by the maximum residual value among the first five iterations of the simulation.

The output of the CFD simulations was sufficient to provide weight support, produced a wake similar to published wakes for *T. alba*, and produced a polar similar to those published for other birds. Lift accounted for 94.2% of body weight after accounting for acceleration. The wake, as visualized by Q-criterion [34], was similar to wakes measured behind the same individual, and the spanwise downwash distribution was also qualitatively and quantitatively similar to that measured using particle tracking [33,35]. Finally, the simulated lift-drag polars were consistent with other published data [30]. Importantly, they exhibit similar values for the coefficient of lift (approx. 0.8) at high (greater than 20°) angle of attack, for which much of the glide model operates; but note our drag values are larger owing to the inclusion of the body (electronic supplementary material, figure S5). The lift and drag polars were computed from transient simulations for 10 angles of attack between −5° and 50°. The simulation timestep was 0.2 ms and lift and drag values were determined from the final, steady timestep.

Ethics. All work was approved by the Ethics and Welfare Committee of the Royal Veterinary College (URN 2015 1358) and the University of Bristol Animal Welfare and Ethical Review Body (UIN UB/15/070) and complied with all relevant ethical regulations.

Data accessibility. Relevant data is included as electronic supplementary material within file 'Dataset S1'

Authors' Contributions. J.A.C., J.P.J.S., N.E.D., J.R.U., R.J.B. and S.P.W. conceived and designed the study and collected the flight data. J.A.C., N.E.D. and J.P.J.S. processed the data. J.A.C. and J.P.J.S. analysed the flight data. J.S., J.A.C., J.R.U. and R.J.B. performed the CFD analysis. J.R.U., J.P.J.S. and J.A.C. developed the centre of percussion model. N.E.D., J.P.J.S. and S.P.W. performed the CT analysis. J.A.C., J.P.J.S. and S.P.W. wrote the paper with input from all authors.

Competing interests. The work presented here supports, in part, patent application (UK Patent Application no 1903806.6) to the Royal Veterinary College, with J.A.C., J.P.J.S., N.E.D., J.R.U., R.J.B. and S.P.W. being involved in the application.

Funding. This project has received funding from the European Research Council (ERC) under the European Union's Horizon 2020 research and innovation programme (grant agreement no 679355). This material is based upon work supported by the Air Force Office of Scientific Research, Air Force Material Command, USAF under Award No. FA9550-16-1-0034. J.R.U. was funded by a Wellcome Trust Fellowship 202854/Z/16/Z.

Acknowledgements. We thank Masateru Maeda and Steve Amos for helpful discussions, Nathan Phillips for helpful feedback and assistance during set-up and data collection and Maja Lorenc for assistance during data collection. We also thank Lloyd and Rose Buck for their falconry expertise during the flight testing. We thank Tony Lapsansky and our anonymous reviewers for helpful feedback on the manuscript.

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
