## [Reviewer comments · Proceedings of the Royal Society B: Biological Sciences]

Review History

RSPB-2020-0905.R0 (Original submission)

Review form: Reviewer 1

Recommendation

Accept with minor revision (please list in comments)

Scientific importance: Is the manuscript an original and important contribution to its field?

Excellent

General interest: Is the paper of sufficient general interest?

Good

Quality of the paper: Is the overall quality of the paper suitable?

Excellent

Is the length of the paper justified?

Yes

Should the paper be seen by a specialist statistical reviewer?

No

Do you have any concerns about statistical analyses in this paper? If so, please specify them explicitly in your report.

No

It is a condition of publication that authors make their supporting data, code and materials available - either as supplementary material or hosted in an external repository. Please rate, if applicable, the supporting data on the following criteria.

Is it accessible?

Yes

Is it clear?

Yes

Is it adequate?

Yes

Do you have any ethical concerns with this paper?

No

Comments to the Author

General comments:

The role of wing flexibility is not discussed anywhere in this paper despite it having a large potential impact on the interpretation and conclusions.

No comment is made in the discussion about the differences between their three gust test conditions (low, mid, high) after introducing the idea in Figure 1 and showing the different shapes and velocities in Figure 2.

I believe that their statements on the contribution of different components contributions to conservation of momentum should be backed up by the actual calculation. It comes off as qualitative rather than quantitative.

Regarding CFD, while the code is listed it might be nice to mention what flow assumptions are assumed for those not familiar with the code, i.e. is it low speed? etc

Specific line references:

Line 55: Tennekes book is not a reliable source of information. This book does not cite any of the original work that it references, nor does it explain how it arrives at its stated conclusions. See Pennycuick's review of the original work:

<https://www.nature.com/articles/381126b0.pdf?origin=ppub>

Line 92: About what axis is wing pitch angle defined?

Figure 2A: There visually appears to be asymmetries in the wing shape on either side. Are the presented results for one wing or for an average of both wings?

Figure 2B,C,D: It may be more informative to plot these panels with the mean gust velocity similar to Fig. 3a

Line 109: Where is the origin of the torso and the CG of the whole bird? The materials section (Line 351) states that a single glide configuration was scanned. So, does that mean that the whole bird CG and wing CG was calculated by assuming a rigid body rotation of the wings about the shoulder? Has the effect of mass distribution change due to the wing morphing been accounted for in this calculation? If so how?

Line 113: How does the force provided by the gust come into play here? The conservation of momentum will be the inertial effect of the wings, inertial effect of the torso and the aerodynamic force due to the gust. A net positive acceleration of the wing mass does not necessarily indicate a net downward (negative) acceleration of the body unless all other forces acting on the body is zero, which is not the case here. Also the previous sentences and figure show that the torso does not have downward momentum just a lower value than the wings.

Line 115/117: The nomenclature of "impulse" suggests that the impulse due to the gust has been calculated which is the time integral of the resultant force in Newton-seconds however in

Fig. 3 this is a velocity measurement of meters/second. I assume this is the velocity difference between the different components of the bird. Perhaps change the nomenclature throughout the paper to “rejected velocity”?

Line 117: Does aerodynamic rejection refer to the difference between the rigid CFD body to translate vertically compared to the whole bird CG? If so this is not really an aerodynamic rejection but rather another inertial mechanism?

Line 170: Is this elliptical lift distribution in contrast to the recently published JEB paper? <https://jeb.biologists.org/content/223/3/jeb214809>

Line 201: What is the minimum time required for muscle activation? How do you know that 80ms was too fast for the muscles to activate? Or that they hadn’t actively “tensed” up before entering the gust? Provide citations and further discussion to support this sentence. This is a big result of your paper and mentioned multiple times in the abstract and intro.

Line 228: Does this mean that this bird was found to be optimally loaded? This is a big statement that requires further explanation and supporting numerical data.

Line 383: It is not clear how comparison of the CFD mesh to trial recordings demonstrated “that the bird exhibited similar and consistent approach postures in all flights”

Line 378: CFD at lower Reynolds numbers can be difficult. Provide the CFD test Reynolds number, some validation of the CFD method and discuss how this could impact the results.

Line 558: What method was used to calculate the appropriate moment of inertia about the hinge calculated? And was this assumed to be constant throughout the sequence?

Review form: Reviewer 2

Recommendation

Major revision is needed (please make suggestions in comments)

Scientific importance: Is the manuscript an original and important contribution to its field?

Good

General interest: Is the paper of sufficient general interest?

Good

Quality of the paper: Is the overall quality of the paper suitable?

Acceptable

Is the length of the paper justified?

Yes

Should the paper be seen by a specialist statistical reviewer?

No

Do you have any concerns about statistical analyses in this paper? If so, please specify them explicitly in your report.

No

It is a condition of publication that authors make their supporting data, code and materials available - either as supplementary material or hosted in an external repository. Please rate, if applicable, the supporting data on the following criteria.

Is it accessible?

Yes

Is it clear?

Yes

Is it adequate?

Yes

Do you have any ethical concerns with this paper?

No

Comments to the Author

This work assesses how birds change their wing shape in response to vertical gusts by combining experimental testing of live owls and computational fluid dynamic modeling of a rigid owl model. The authors aim to show that bird wings experience passive upwards translation and rotation (preflex) in response to vertical gusts in order to reduce the translation of the head and body, allowing for more stable flight. This response is said to mimic a 'suspension system'. The authors have undoubtedly taken on an ambitious task to understand a complex problem. While the experiments are fairly strong and show interesting results, the integration of the CFD analysis is flawed. It is my opinion that this paper would be much stronger if the authors eliminate the CFD analysis and expanded the assessment of their experimental analysis.

Major comments:

1. This analysis ignores the effects of aeroelasticity on the gust response and attributes the gust rejection solely to the preflex mechanism. The reduction in peak vertical velocity is calculated by comparing the results of the in-flight (aeroelastic) bird data to a completely rigid CFD model. Wing and feather flexibility will play some, if not a large, role in the vertical translation/acceleration. Aeroelastic deformations are even observable in the supplemental S1 movie. The authors cannot attribute the rejection they are calculating solely to the preflex mechanism. The only way to numerically quantify the role of preflex from this CFD analysis would be to make the CFD model into an FSI model which accounts for the role of aeroelasticity. However, accurately modeling the aeroelasticity of a complete bird is an entirely different challenge in and of itself which I would not recommend. The simpler solution would be to report your observations about the wing/body translation and rotation directly from your experimental measurements. While I understand that this would eliminate Fig. 3, you could replace it with an assessment of how the mass distribution is changing in time as the owl flies through the gust. Plots on the tail's response during the gust would also be useful, as video S1 shows that the tail experiences an upwards pitching motion followed by a downwards pitching motion, the latter of which I presume is a landing response.
2. The CFD analysis has no validation. The authors have attempted to justify the validity of their results by claiming that the lift and drag is similar to other reported values, but the paper cited does not include data from an owl and the other species tested in that paper have a wide spread of lift and drag results (ranging from Cl max of 0.8 to 1.4). This is neither validation nor justification of the use of CFD. Comparison to other lift/drag data specifically from an owl would be needed.
3. The details on how the CFD code was implemented are very fuzzy, especially in the main portion of the text. Even in the methods section, some key aspects of the modeling remain unclear. What wing/body geometry is used? Is it from the collected in flight data and if so, at what stage in flight before the gust? How are the aerodynamic gusts applied? Does it consider just the 2D variation in gust intensity like the data shown in Figure 1b, or the 3D variation in intensity like the data shown in Figure S2? Does the CFD model assume that the bird flies directly over the center of the gust? (From the results presented in Figure 2, the wing deflections are highly asymmetric which implies that it is not directly flying over the point of symmetry of the gust (or that the gust itself is asymmetric)). What was the convergence criteria for the numerical solution? What mesh shape is used (C-grid, o-grid, etc.)?
4. Although the title claims that this preflex mechanism acts as a suspension system, this is neither quantitatively or qualitatively defined in the text. The only justification for why this acts as a suspension system is because "the bird consistently maintained a stable trajectory of its head

and torso". The only way to quantitatively claim this is to use the results to mathematically model the owl as a suspension system which is done using a mass-spring-damper analysis. If you wish to keep the suspension analogy, I recommend doing this.

General Comments:

Line 39: "Implementing a similar reflex mechanism in future aircraft..." Phrasing is important here, and 'future aircraft' to the public implies transonic aircraft. The reflex shown here undergoes large scale displacement which would almost certainly be unsuitable for transonic aircraft. It's important to clarify that the target application is (presumably) small scale aircraft or UAVs.

Line 55: "The gusts encountered by birds can be of a similar magnitude to their flight speed (4)". There is no qualitative data (or reference) of gust speeds and correspond flight speeds in Tennekes' paper. To support this claim, the authors can either conduct a literature review of papers reporting the flight speed and gust speed of a variety of species, or find a paper which does.

Fig. 1B: The measurements at the peak of the gusts appear to have no error bars. Is this because they are small or missing? Similarly, the error bars on the edge of the gusts are quite large (up to 0.3-0.4 m/s (or upwards of 10-15%) at a distance of 0.4m into the gust) and even overlap with the error bars of the data points for the other flow velocities, yet no discussion of error in the analysis is really presented. The authors report an n of 6 (presumably for the error calculations) but in the methods section they state that anemometer measurements were taken at 50 hz for 30 s which totals 1.5k measurements. This is confusing. Shouldn't the latter number be used in error calculations? Lastly, the methods section states that the velocity measurements were collected in a 5x6 grid, so why is only one cross sectional profile shown? The full velocity could be plotted as a 3D contour map and would be more informative.

Figure 1C: The leading edge of the point clouds and the corresponding mesh have a significant amount of rough edges and features. Are you confident that these represent physical features observed on the bird or are they the result of errors in the 3D reconstruction? This is important because leading edge protrusions are well known to affect the flow, for example by tripping the flow from laminar to turbulent. This could affect the results of your CFD analysis.

Line 73: More detail is needed about the CFD model earlier on in the text. Is it a static or dynamic model? Is it rigid or flexible? What turbulence model was implemented? Some of these details are later discussed in the methods section, but the reader needs a better understanding of the CFD was performed before introducing your results so they can assess them properly.

Line 90: This is the only time that the suspension system is discussed somewhat qualitatively and in my opinion, this does not adequately demonstrate that the reflex mechanism acts as one. To truly prove that this mechanism acts a suspension system, you should use your experimental results to model it as a mass-spring-damper system which mathematically describes a suspension system.

Line 92: "...A substantial reduction in wing pitch angle" is not the only mechanism that may "modify the lift generated by the wings". Assuming the pitching axis does not coincide with the aerodynamic center, changing the angle of attack will also change the pitching moment of the owl. This should not be ignored. Additionally, was any tail motion observed during the in-flight tests? If so, this motion might be a mechanism to counteract a change in pitching moment from the gust.

Figure 2: It would be useful to superimpose the gust profile onto these plots so the reader can compare the velocity, elevation, and pitch response relative to the spatial strength of the gust. Furthermore, there is a fair amount of variability between test runs, so it would be useful to, for

example, plot shaded error bands on the plotted averages.

Paragraph @106: The classification of how the inertial vs aerodynamic gust response was determined is not sufficiently detailed. I presume mechanism 1) @ line 109 corresponds to the former, and mechanism 2) corresponds to the latter? Furthermore, do the mechanisms presented @ 109 “Calculating the difference between 1)...2)” mean the differences in the vertical velocity? As written, this statement implies the metric is taken by calculating the difference between the centers of mass, not velocity.

Line 113: The conservation of moment calculations would be useful to perform here to physically show the reader how this is true. However, I believe you also need to include the momentum of the fluid especially since you are assessing a dynamic gust.

Line 115: The use of the word ‘impulse’ throughout the text is misleading. It appears that this calculation is simply the difference between the CFD and the experimental results. Impulse is defined as the force acting over time, but it does not appear that the authors performed time integration to actually calculate the impulse.

Line 117/119: “At this stage...” “Later, as the wing pitched down...”. It would be useful to mark these stages on the plots to direct the reader exactly to where these trends are occurring.

Line 120-122: “As the wing pitched downwards, there was an increase in the difference between the measured vertical velocity of the whole bird center of mass and the rigid body simulation, indicating large aerodynamic-based gust rejection of $48 \pm 4\%$ of the potential impulse (Fig 3c).” The authors cannot confirm that this is due to the translation and rotation of the wings. The wing and feather aeroelasticity plays a substantial role in mitigating gusts, which is not accounted for here since the calculations are performed by taking the difference between the rigid CFD model and the flexible in-flight experiments.

Line 188: The authors state that “birds may then be able to tune the dynamics involved with gust rejection by modifying the stiffness of their shoulder joint”. If the wing does indeed act as a suspension system as the authors claim, then changing the stiffness of the shoulder would effect the timing of the inertial rejection (dictated by mass-spring-damper equations of motion). Stiffening the shoulder muscle would cause the inertial rejection to reach it’s maximum sooner. This is contradictory the statement made in line 200, claiming that the inertial response must be passive since it is not faster than the reaction time measured in other studies. That is, unless the shoulder stiffness is assumed to already be at a maximum for these in-flight tests.

Figure 3: The results in Figure 3 are only presented for one gust intensity (medium), right? Do these results change at the lower and higher intensities? Since this was only assessed at one gust intensity, I don’t think you can conclude that “passive mechanical properties of the shoulder joint and wing are responsible for immediate gust rejection” because this time frame was “equal to the minimum behavioural reaction time measured in other bird species”. The data shown in Figure 2c indicates that for the high gust intensities, the wing elevation changes more rapidly (the response onset is sooner and the slope is steeper) but you have not yet proven that this difference between gust intensities is only due to the aerodynamic rejection.

Line 292: The spacing of the velocity measurements is .45m which appears to be approximately the same as the span of the owl. Some comment would be useful on the spatial sampling of the velocity profiles. Figure S2 shows only 3 velocity measurements overlap with where the bird should fly (when the wings are fully extended). Presumably when the wings are elevated during a gust, no data points capture the velocity over the wings.

Line 332: The surface accuracy of the reconstruction is said to be 1.67mm, however there are many features of the bird that will be well below this thickness (namely, the trailing edge feather

thickness). How does that effect your reconstruction?

Line 341: Are the mass properties segmented into each wing section (i.e. handwing, armwing etc.) or is the mass tracking done by translation and rotation of the whole wing mass? This could yield very different results if the whole wing is treated as one lump mass that can only undergo translation/rotation. Upon looking at the supplementary material, it appears that the mass is recorded for each voxel, so is each voxel mapped to the deformed bird geometry?

Line 370: "The model was free to accelerate both horizontally and vertically, but could not rotate, as the real bird displayed no tendency to pitch during the gust." This is a major assumption. The lack of observation of the bird pitching during a gust does not mean that, in the absence of wing motions, the bird would not pitch. Elaborating on my comment regarding line 92, the wing's reduction in angle of attack is likely an active mechanism to, in part, minimize the pitching moments caused by the gust. When the gust hits the nose end of the bird first, this would cause a nose-up pitching moment. This has been shown in similar experiments done with gliders, so it is not accurate to assume that there would be no pitch in your rigid model.

Line 402: The authors justify their turbulent model by claiming that it "is appropriate for bird flight simulations that require a high accuracy boundary layer"; however, their first inflation layer near the wall has a y^+ value of 10 which does not resolve the viscous sublayer. Y^+ values of approximately 1 are recommended for best results using low Re (k - ω) models. The authors report a difference of 3% in lift between $y^+ 10$ and $y^+ 3$ results, but there would presumably be an even larger error between $y^+ 10$ and $y^+ 1$. And how does this affect the drag error?

403: From my perspective, comparing the results of different turbulent models is not sufficient for validating the results of the modelling.

411: The authors justify their CFD results by claiming that their data is similar to the results presented in [25]. However, this citation does not have any aerodynamic coefficient data for an owl. Furthermore, the lift curves for all of the species tested range from a CL_{max} of 1.4 for the woodcock, to a minimum of CL_{max} of around 0.8 for the swift. This is a huge range of lift curves to compare results to, so it is not sufficient to simply say that your results "are similar to other published data (25)".

Were any corrections made to account for the bird flying too high/low left/right? Since the gust intensity is dependent on distance from the gust source, differences in flight height above the gust could change your measurements. Some mention of this in the results would be sufficient.

Review form: Reviewer 3 (Anthony Lapsansky)

Recommendation

Accept with minor revision (please list in comments)

Scientific importance: Is the manuscript an original and important contribution to its field?

Excellent

General interest: Is the paper of sufficient general interest?

Acceptable

Quality of the paper: Is the overall quality of the paper suitable?

Good

Is the length of the paper justified?

Yes

Should the paper be seen by a specialist statistical reviewer?

No

Do you have any concerns about statistical analyses in this paper? If so, please specify them explicitly in your report.

No

It is a condition of publication that authors make their supporting data, code and materials available - either as supplementary material or hosted in an external repository. Please rate, if applicable, the supporting data on the following criteria.

Is it accessible?

Yes

Is it clear?

Yes

Is it adequate?

No

Do you have any ethical concerns with this paper?

No

Comments to the Author

A thorough study with an innovative experimental design. I am convinced of the general conclusions (but see my comments about the aerodynamic consequences of wing elevation + retraction). The figures are clean, and the methods are detailed. Overall, my suggestions are to emphasize the importance of the work to help draw in a general audience, more explicitly state the rationale behind the experimental design (to give yourselves due credit), and explain the interpretation of the results in more detail to help hold the attention/understanding of the general audience. Given the page limitations of Proceedings B and the complexity of the methods of the study, it seems that you are stuck between a rock and a hard place. With that in mind, I have tried to avoid asking for additional details unless it seems vitally important for the work to be readable by a general audience. Specific examples below.

General comments:

- The text switches between referring to the head and torso and just the torso. I suspect that you are referencing the head + torso throughout (i.e. the whole bird minus the wings). Maybe use something like "torso + head" or "torso/head" throughout, just to avoid any confusion.
- As a suggestion, I think that a more detailed introduction would help to draw the interest of non-experts and help them to appreciate the importance of this study. Specifically, I think that the introduction could be improved by discussing the importance of rejecting perturbations in flight more concretely. You mention in the abstract that reduced motion of the torso simplifies certain tasks, but the actual costs of failing to reject perturbations remain somewhat opaque. Not rejecting gusts has implications for cost of transport and risk of injury - both for animals and engineered vehicles. I think that putting your work in this context early will help emphasize the importance for the general audience. But do what you can with the space permitted.

- Re: "The time-varying magnitude of the two gust rejection mechanisms was quantified by calculating the difference between 1) the torso and the whole bird centre of mass computed using the mass calibrated CT scan (Fig 3a,b); and 2) the whole bird centre of mass and the rigid body CFD-based simulation (Fig 3c)."

I don't think that it would be immediately evident to a general reader that any difference between the torso and the whole bird center of mass indicates an inertial response and that any difference between the whole bird center of mass and the center of mass of the CFD model indicates an aerodynamic response. I don't disagree. But I do think that it would be very helpful to walk through that logic in a bit more detail either here or in the introduction, rather than stating these concepts as evident.

In other words, I think that the experimental design is an innovative way to separate between inertial and aerodynamic responses, but I don't think that you give yourselves due credit for coming up with a powerful experimental design!

- Why is the velocity of the whole bird, wing, and torso + head the important factor in measuring gust rejection, rather than their position? Is it because the gust is too brief to cause major changes to position? I think it would be helpful if you could briefly mention why the focus is placed on vertical velocity rather than position (or acceleration, for that matter).

- It seems strange that the rapid elevation of the wings does not have any effect on the whole bird center of mass relative to the CFD model. If the wings are moving upward with the air, that should mean that the angle of incidence of the wings is much lower in the actual bird than in the CFD-based model, which I would expect would cause the whole bird to fall a bit (relative to the CFD-based model). Evidently, that is not the case ("there was negligible aerodynamic rejection accounting for only $6 \pm 6\%$ of the total rejection"). Do you have any thoughts on why the aerodynamic rejection was so low even while the wings were rapidly elevated?

- I would suggest making the distinction between the CFD simulation and the CFD-based mathematical model clearer. Perhaps you could refer to the mathematical model without using the word "CFD". It took a bit of time before I understood the distinction, so I worry that others will have the same trouble.

- Unless I am missing something about the mathematical model, I think that you are missing two important mechanisms of aerodynamic rejection, and so should not be attributing the response entirely to changes in wing pitch. In addition to the change in wing pitch, the elevation of the wings also means that the vertical component of the lift vector acting on each wing is reduced quite a bit (though the overall lift force is roughly the same). Also, the wings are retracted over the gust (at least based on the example Fig. 2A). The combination of wing elevation and wing retraction means that the effective wingspan is greatly reduced over the gust. For example, the effective wingspan of the red wings in Figure 2A is about 50% of that of the blue wings, which must have a huge effect on the vertical force they produce. Therefore, I think that elevating the wings has both an inertial and aerodynamic effect, which makes it harder to cleanly separate the two mechanisms of gust rejection.

Luckily, this does not alter your major conclusion about the importance of inertial rejection. However, I do think that you need to discuss these two other mechanisms of aerodynamic rejection rather than put all the emphasis on changes in wing pitch.

- The discussion is great, overall. One additional topic that you might consider including (space permitting) is the speed at which an inertial response leads to a change in position relative to the speed at which an increment of force leads to a change in position. Is there something about the lag time between a change in net force and the subsequent change in position (versus throwing one's weight around) that makes an inertial response especially effective? The answer

may be no. Just something to think about.

Specific comments:

Line 53 – I suggest starting a new paragraph at “Birds...”

Line 63 – Preference thing, but it is not immediately obvious if “our observations” refers to observations during this study or before this study. It is not all that important, but changing the wording slightly (perhaps by removing “Based on our observations”) might clear up any minor confusion.

Line 79 – Please specify at what height from the gust generator these measurements were taken.

Line 92 – See my above comment about the aerodynamic impact of changes in wing elevation.

Line 93 – Wing pitch also seems to be proportionate to gust strength, as stated in the legend for figure 2. I would suggest rewording slightly to make that clearer.

Line 118 – Specify s.e.m or standard deviation

Figure 2 – The purple-green arrows on top of figure 2 are a bit confusing to me. I am not sure if they indicate a continuum between ‘inertia dominated’ and ‘aerodynamics dominated’ rejection, or if they are simply labeled for the arrows below. I feel like the latter is true. But you might consider the unintended interpretation.

Line 143 – Typo, add ‘the’ in “centre of _ plot”

Line 167 – I would appreciate a more detailed presentation of how the center of percussion is calculated for the owl wing. Perhaps you could include an excel sheet or script in which this parameter is calculated from the mass distribution of the wing.

The paragraph at 166 – Are there any differences in the center of percussion and center of pressure between the two wings? And is there any variation in the span-wise position of the center of pressure with the angle of attack in the CFD model?

Line 170 – Please specify that these measurements are from the CFD model.

Line 378 – How did the overall lift force calculated via CFD compare to the animal’s weight?

Line 533 – Because you have space in the supplemental material, I would suggest that you proceed very slowly through this calculation. It is an important component of your conclusion, so it would be great if a general reader could follow the math without too much effort. For example, the addition of K in the same step as substituting and rearranging takes some work to follow. A drawing may help. Also, should it be ‘ or simply ?

Figure S2 – I have trouble interpreting the top row of the figure without referencing the methods. Perhaps including labels like "top view" and "side view", and a profile of the owl (in gray, like in the lower row) in the top row would help.

Figure S4 – That’s a dense mesh!

Decision letter (RSPB-2020-0905.R0)

22-May-2020

Dear Dr Cheney:

I am writing to inform you that your manuscript RSPB-2020-0905 entitled "Bird wings act as a suspension system that rejects gusts" has, in its current form, been rejected for publication in Proceedings B.

This action has been taken on the advice of referees, who have recommended that substantial revisions are necessary. With this in mind we would be happy to consider a resubmission, provided the comments of the referees are fully addressed. However please note that this is not a provisional acceptance.

Sincerely,
Dr Sasha Dall
<mailto:proceedingsb@royalsociety.org>

Associate Editor
Comments to Author:
Associate Editor: Doug Altshuler

The authors have combined live measurements from flying owls and computational fluid dynamics on an owl model to argue that birds are able to compensate for gusts using a preflex that reduces head and body motion. I agree with the three referees that these are very exciting results that are likely to have a significant impact in the field of animal flight. The three reviews contain a number of suggestions for revision but one common theme was that the presentation was a bit dense. The writing was good but there is a lot of material, and some methodological concerns about the CFD were raised. I'm not sure whether the best option is to save the CFD for a separate manuscript, as suggested directly by one of the referees, or whether further editing can handle this. In any case, it would be helpful to see a revised manuscript that is responsive to these reviews.

Reviewer(s)' Comments to Author:
Referee: 1

Comments to the Author(s)

General comments:

The role of wing flexibility is not discussed anywhere in this paper despite it having a large potential impact on the interpretation and conclusions.

No comment is made in the discussion about the differences between their three gust test conditions (low, mid, high) after introducing the idea in Figure 1 and showing the different shapes and velocities in Figure 2.

I believe that their statements on the contribution of different components contributions to conservation of momentum should be backed up by the actual calculation. It comes off as qualitative rather than quantitative.

Regarding CFD, while the code is listed it might be nice to mention what flow assumptions are assumed for those not familiar with the code, i.e. is it low speed? etc

Specific line references:

Line 55: Tennekes book is not a reliable source of information. This book does not cite any of the original work that it references, nor does it explain how it arrives at its stated conclusions. See Pennycuik's review of the original work:

<https://www.nature.com/articles/381126b0.pdf?origin=ppub>

Line 92: About what axis is wing pitch angle defined?

Figure 2A: There visually appears to be asymmetries in the wing shape on either side. Are the presented results for one wing or for an average of both wings?

Figure 2B,C,D: It may be more informative to plot these panels with the mean gust velocity similar to Fig. 3a

Line 109: Where is the origin of the torso and the CG of the whole bird? The materials section (Line 351) states that a single glide configuration was scanned. So, does that mean that the whole bird CG and wing CG was calculated by assuming a rigid body rotation of the wings about the shoulder? Has the effect of mass distribution change due to the wing morphing been accounted for in this calculation? If so how?

Line 113: How does the force provided by the gust come into play here? The conservation of momentum will be the inertial effect of the wings, inertial effect of the torso and the aerodynamic force due to the gust. A net positive acceleration of the wing mass does not necessarily indicate a net downward (negative) acceleration of the body unless all other forces acting on the body is zero, which is not the case here. Also the previous sentences and figure show that the torso does not have downward momentum just a lower value than the wings.

Line 115/117: The nomenclature of "impulse" suggests that the impulse due to the gust has been calculated which is the time integral of the resultant force in Newton-seconds however in Fig. 3 this is a velocity measurement of meters/second. I assume this is the velocity difference between the different components of the bird. Perhaps change the nomenclature throughout the paper to "rejected velocity"?

Line 117: Does aerodynamic rejection refer to the difference between the rigid CFD body to translate vertically compared to the whole bird CG? If so this is not really an aerodynamic rejection but rather another inertial mechanism?

Line 170: Is this elliptical lift distribution in contrast to the recently published JEB paper?

<https://jeb.biologists.org/content/223/3/jeb214809>

Line 201: What is the minimum time required for muscle activation? How do you know that 80ms was too fast for the muscles to activate? Or that they hadn't actively "tensed" up before entering the gust? Provide citations and further discussion to support this sentence. This is a big result of your paper and mentioned multiple times in the abstract and intro.

Line 228: Does this mean that this bird was found to be optimally loaded? This is a big statement that requires further explanation and supporting numerical data.

Line 383: It is not clear how comparison of the CFD mesh to trial recordings demonstrated "that the bird exhibited similar and consistent approach postures in all flights"

Line 378: CFD at lower Reynolds numbers can be difficult. Provide the CFD test Reynolds number, some validation of the CFD method and discuss how this could impact the results.

Line 558: What method was used to calculate the appropriate moment of inertia about the hinge calculated? And was this assumed to be constant throughout the sequence?

Referee: 2

Comments to the Author(s)

This work assesses how birds change their wing shape in response to vertical gusts by combining experimental testing of live owls and computational fluid dynamic modeling of a rigid owl model. The authors aim to show that bird wings experience passive upwards translation and rotation (preflex) in response to vertical gusts in order to reduce the translation of the head and body, allowing for more stable flight. This response is said to mimic a 'suspension system'. The authors have undoubtedly taken on an ambitious task to understand a complex problem. While the experiments are fairly strong and show interesting results, the integration of the CFD analysis is flawed. It is my opinion that this paper would be much stronger if the authors eliminate the CFD analysis and expanded the assessment of their experimental analysis.

Major comments:

1. This analysis ignores the effects of aeroelasticity on the gust response and attributes the gust rejection solely to the preflex mechanism. The reduction in peak vertical velocity is calculated by comparing the results of the in-flight (aeroelastic) bird data to a completely rigid CFD model. Wing and feather flexibility will play some, if not a large, role in the vertical translation/acceleration. Aeroelastic deformations are even observable in the supplemental S1 movie. The authors cannot attribute the rejection they are calculating solely to the preflex mechanism. The only way to numerically quantify the role of preflex from this CFD analysis would be to make the CFD model into an FSI model which accounts for the role of aeroelasticity. However, accurately modeling the aeroelasticity of a complete bird is an entirely different challenge in and of itself which I would not recommend. The simpler solution would be to report your observations about the wing/body translation and rotation directly from your experimental measurements. While I understand that this would eliminate Fig. 3, you could replace it with an assessment of how the mass distribution is changing in time as the owl flies through the gust. Plots on the tail's response during the gust would also be useful, as video S1 shows that the tail experiences an upwards pitching motion followed by a downwards pitching motion, the latter of which I presume is a landing response.
2. The CFD analysis has no validation. The authors have attempted to justify the validity of their results by claiming that the lift and drag is similar to other reported values, but the paper cited does not include data from an owl and the other species tested in that paper have a wide spread of lift and drag results (ranging from Cl max of 0.8 to 1.4). This is neither validation nor justification of the use of CFD. Comparison to other lift/drag data specifically from an owl would be needed.
3. The details on how the CFD code was implemented are very fuzzy, especially in the main portion of the text. Even in the methods section, some key aspects of the modeling remain unclear. What wing/body geometry is used? Is it from the collected in flight data and if so, at what stage in flight before the gust? How are the aerodynamic gusts applied? Does it consider just the 2D variation in gust intensity like the data shown in Figure 1b, or the 3D variation in intensity like the data shown in Figure S2? Does the CFD model assume that the bird flies directly over the center of the gust? (From the results presented in Figure 2, the wing deflections are highly asymmetric which implies that it is not directly flying over the point of symmetry of the gust (or that the gust itself is asymmetric)). What was the convergence criteria for the numerical solution? What mesh shape is used (C-grid, o-grid, etc.)?
4. Although the title claims that this preflex mechanism acts as a suspension system, this is neither quantitatively or qualitatively defined in the text. The only justification for why this acts as a suspension system is because "the bird consistently maintained a stable trajectory of its head

and torso". The only way to quantitatively claim this is to use the results to mathematically model the owl as a suspension system which is done using a mass-spring-damper analysis. If you wish to keep the suspension analogy, I recommend doing this.

General Comments:

Line 39: "Implementing a similar reflex mechanism in future aircraft...". Phrasing is important here, and 'future aircraft' to the public implies transonic aircraft. The reflex shown here undergoes large scale displacement which would almost certainly be unsuitable for transonic aircraft. It's important to clarify that the target application is (presumably) small scale aircraft or UAVs.

Line 55: "The gusts encountered by birds can be of a similar magnitude to their flight speed (4)". There is no qualitative data (or reference) of gust speeds and correspond flight speeds in Tennekes' paper. To support this claim, the authors can either conduct a literature review of papers reporting the flight speed and gust speed of a variety of species, or find a paper which does.

Fig. 1B: The measurements at the peak of the gusts appear to have no error bars. Is this because they are small or missing? Similarly, the error bars on the edge of the gusts are quite large (up to 0.3-0.4 m/s (or upwards of 10-15%) at a distance of 0.4m into the gust) and even overlap with the error bars of the data points for the other flow velocities, yet no discussion of error in the analysis is really presented. The authors report an n of 6 (presumably for the error calculations) but in the methods section they state that anemometer measurements were taken at 50 hz for 30 s which totals 1.5k measurements. This is confusing. Shouldn't the latter number be used in error calculations? Lastly, the methods section states that the velocity measurements were collected in a 5x6 grid, so why is only one cross sectional profile shown? The full velocity could be plotted as a 3D contour map and would be more informative.

Figure 1C: The leading edge of the point clouds and the corresponding mesh have a significant amount of rough edges and features. Are you confident that these represent physical features observed on the bird or are they the result of errors in the 3D reconstruction? This is important because leading edge protrusions are well known to affect the flow, for example by tripping the flow from laminar to turbulent. This could affect the results of your CFD analysis.

Line 73: More detail is needed about the CFD model earlier on in the text. Is it a static or dynamic model? Is it rigid or flexible? What turbulence model was implemented? Some of these details are later discussed in the methods section, but the reader needs a better understanding of the CFD was performed before introducing your results so they can assess them properly.

Line 90: This is the only time that the suspension system is discussed somewhat qualitatively and in my opinion, this does not adequately demonstrate that the reflex mechanism acts as one. To truly prove that this mechanism acts a suspension system, you should use your experimental results to model it as a mass-spring-damper system which mathematically describes a suspension system.

Line 92: "...A substantial reduction in wing pitch angle" is not the only mechanism that may "modify the lift generated by the wings". Assuming the pitching axis does not coincide with the aerodynamic center, changing the angle of attack will also change the pitching moment of the owl. This should not be ignored. Additionally, was any tail motion observed during the in-flight tests? If so, this motion might be a mechanism to counteract a change in pitching moment from the gust.

Figure 2: It would be useful to superimpose the gust profile onto these plots so the reader can compare the velocity, elevation, and pitch response relative to the spatial strength of the gust. Furthermore, there is a fair amount of variability between test runs, so it would be useful to, for example, plot shaded error bands on the plotted averages.

Paragraph @106: The classification of how the inertial vs aerodynamic gust response was determined is not sufficiently detailed. I presume mechanism 1) @ line 109 corresponds to the former, and mechanism 2) corresponds to the latter? Furthermore, do the mechanisms presented @ 109 “Calculating the difference between 1)...2)” mean the differences in the vertical velocity? As written, this statement implies the metric is taken by calculating the difference between the centers of mass, not velocity.

Line 113: The conservation of moment calculations would be useful to perform here to physically show the reader how this is true. However, I believe you also need to include the momentum of the fluid especially since you are assessing a dynamic gust.

Line 115: The use of the word ‘impulse’ throughout the text is misleading. It appears that this calculation is simply the difference between the CFD and the experimental results. Impulse is defined as the force acting over time, but it does not appear that the authors performed time integration to actually calculate the impulse.

Line 117/119: “At this stage...” “Later, as the wing pitched down...”. It would be useful to mark these stages on the plots to direct the reader exactly to where these trends are occurring.

Line 120-122: “As the wing pitched downwards, there was an increase in the difference between the measured vertical velocity of the whole bird center of mass and the rigid body simulation, indicating large aerodynamic-based gust rejection of $48 \pm 4\%$ of the potential impulse (Fig 3c).” The authors cannot confirm that this is due to the translation and rotation of the wings. The wing and feather aeroelasticity plays a substantial role in mitigating gusts, which is not accounted for here since the calculations are performed by taking the difference between the rigid CFD model and the flexible in-flight experiments.

Line 188: The authors state that “birds may then be able to tune the dynamics involved with gust rejection by modifying the stiffness of their shoulder joint”. If the wing does indeed act as a suspension system as the authors claim, then changing the stiffness of the shoulder would effect the timing of the inertial rejection (dictated by mass-spring-damper equations of motion). Stiffening the shoulder muscle would cause the inertial rejection to reach it’s maximum sooner. This is contradictory the statement made in line 200, claiming that the inertial response must be passive since it is not faster than the reaction time measured in other studies. That is, unless the shoulder stiffness is assumed to already be at a maximum for these in-flight tests.

Figure 3: The results in Figure 3 are only presented for one gust intensity (medium), right? Do these results change at the lower and higher intensities? Since this was only assessed at one gust intensity, I don’t think you can conclude that “passive mechanical properties of the shoulder joint and wing are responsible for immediate gust rejection” because this time frame was “equal to the minimum behavioural reaction time measured in other bird species”. The data shown in Figure 2c indicates that for the high gust intensities, the wing elevation changes more rapidly (the response onset is sooner and the slope is steeper) but you have not yet proven that this difference between gust intensities is only due to the aerodynamic rejection.

Line 292: The spacing of the velocity measurements is .45m which appears to be approximately the same as the span of the owl. Some comment would be useful on the spatial sampling of the velocity profiles. Figure S2 shows only 3 velocity measurements overlap with where the bird should fly (when the wings are fully extended). Presumably when the wings are elevated during a gust, no data points capture the velocity over the wings.

Line 332: The surface accuracy of the reconstruction is said to be 1.67mm, however there are many features of the bird that will be well below this thickness (namely, the trailing edge feather thickness). How does that effect your reconstruction?

Line 341: Are the mass properties segmented into each wing section (i.e. handwing, armwing etc.) or is the mass tracking done by translation and rotation of the whole wing mass? This could yield very different results if the whole wing is treated as one lump mass that can only undergo translation/rotation. Upon looking at the supplementary material, it appears that the mass is recorded for each voxel, so is each voxel mapped to the deformed bird geometry?

Line 370: "The model was free to accelerate both horizontally and vertically, but could not rotate, as the real bird displayed no tendency to pitch during the gust." This is a major assumption. The lack of observation of the bird pitching during a gust does not mean that, in the absence of wing motions, the bird would not pitch. Elaborating on my comment regarding line 92, the wing's reduction in angle of attack is likely an active mechanism to, in part, minimize the pitching moments caused by the gust. When the gust hits the nose end of the bird first, this would cause a nose-up pitching moment. This has been shown in similar experiments done with gliders, so it is not accurate to assume that there would be no pitch in your rigid model.

Line 402: The authors justify their turbulent model by claiming that it "is appropriate for bird flight simulations that require a high accuracy boundary layer"; however, their first inflation layer near the wall has a y^+ value of 10 which does not resolve the viscous sublayer. Y^+ values of approximately 1 are recommended for best results using low Re (k - ω) models. The authors report a difference of 3% in lift between $y^+ 10$ and $y^+ 3$ results, but there would presumably be an even larger error between $y^+ 10$ and $y^+ 1$. And how does this affect the drag error?

403: From my perspective, comparing the results of different turbulent models is not sufficient for validating the results of the modelling.

411: The authors justify their CFD results by claiming that their data is similar to the results presented in [25]. However, this citation does not have any aerodynamic coefficient data for an owl. Furthermore, the lift curves for all of the species tested range from a CL_{max} of 1.4 for the woodcock, to a minimum of CL_{max} of around 0.8 for the swift. This is a huge range of lift curves to compare results to, so it is not sufficient to simply say that your results "are similar to other published data (25)".

Were any corrections made to account for the bird flying too high/low left/right? Since the gust intensity is dependent on distance from the gust source, differences in flight height above the gust could change your measurements. Some mention of this in the results would be sufficient.

Referee: 3

Comments to the Author(s)

A thorough study with an innovative experimental design. I am convinced of the general conclusions (but see my comments about the aerodynamic consequences of wing elevation + retraction). The figures are clean, and the methods are detailed. Overall, my suggestions are to emphasize the importance of the work to help draw in a general audience, more explicitly state the rationale behind the experimental design (to give yourselves due credit), and explain the interpretation of the results in more detail to help hold the attention/understanding of the general audience. Given the page limitations of Proceedings B and the complexity of the methods of the study, it seems that you are stuck between a rock and a hard place. With that in mind, I have tried to avoid asking for additional details unless it seems vitally important for the work to be readable by a general audience. Specific examples below.

General comments:

- The text switches between referring to the head and torso and just the torso. I suspect that you are referencing the head + torso throughout (i.e. the whole bird minus the wings). Maybe use something like "torso + head" or "torso/head" throughout, just to avoid any confusion.

- As a suggestion, I think that a more detailed introduction would help to draw the interest of non-experts and help them to appreciate the importance of this study. Specifically, I think that the introduction could be improved by discussing the importance of rejecting perturbations in flight more concretely. You mention in the abstract that reduced motion of the torso simplifies certain tasks, but the actual costs of failing to reject perturbations remain somewhat opaque. Not rejecting gusts has implications for cost of transport and risk of injury - both for animals and engineered vehicles. I think that putting your work in this context early will help emphasize the importance for the general audience. But do what you can with the space permitted.

- Re: "The time-varying magnitude of the two gust rejection mechanisms was quantified by calculating the difference between 1) the torso and the whole bird centre of mass computed using the mass calibrated CT scan (Fig 3a,b); and 2) the whole bird centre of mass and the rigid body CFD-based simulation (Fig 3c)."

I don't think that it would be immediately evident to a general reader that any difference between the torso and the whole bird center of mass indicates an inertial response and that any difference between the whole bird center of mass and the center of mass of the CFD model indicates an aerodynamic response. I don't disagree. But I do think that it would be very helpful to walk through that logic in a bit more detail either here or in the introduction, rather than stating these concepts as evident.

In other words, I think that the experimental design is an innovative way to separate between inertial and aerodynamic responses, but I don't think that you give yourselves due credit for coming up with a powerful experimental design!

- Why is the velocity of the whole bird, wing, and torso + head the important factor in measuring gust rejection, rather than their position? Is it because the gust is too brief to cause major changes to position? I think it would be helpful if you could briefly mention why the focus is placed on vertical velocity rather than position (or acceleration, for that matter).

- It seems strange that the rapid elevation of the wings does not have any effect on the whole bird center of mass relative to the CFD model. If the wings are moving upward with the air, that should mean that the angle of incidence of the wings is much lower in the actual bird than in the CFD-based model, which I would expect would cause the whole bird to fall a bit (relative to the CFD-based model). Evidently, that is not the case ("there was negligible aerodynamic rejection accounting for only $6 \pm 6\%$ of the total rejection"). Do you have any thoughts on why the aerodynamic rejection was so low even while the wings were rapidly elevated?

- I would suggest making the distinction between the CFD simulation and the CFD-based mathematical model clearer. Perhaps you could refer to the mathematical model without using the word "CFD". It took a bit of time before I understood the distinction, so I worry that others will have the same trouble.

- Unless I am missing something about the mathematical model, I think that you are missing two important mechanisms of aerodynamic rejection, and so should not be attributing the response entirely to changes in wing pitch. In addition to the change in wing pitch, the elevation of the wings also means that the vertical component of the lift vector acting on each wing is reduced quite a bit (though the overall lift force is roughly the same). Also, the wings are retracted over the gust (at least based on the example Fig. 2A). The combination of wing elevation and wing retraction means that the effective wingspan is greatly reduced over the gust. For example, the effective wingspan of the red wings in Figure 2A is about 50% of that of the blue wings, which must have a huge effect on the vertical force they produce. Therefore, I think that elevating the wings has both an inertial and aerodynamic effect, which makes it harder to cleanly separate the two mechanisms of gust rejection.

Luckily, this does not alter your major conclusion about the importance of inertial rejection. However, I do think that you need to discuss these two other mechanisms of aerodynamic rejection rather than put all the emphasis on changes in wing pitch.

- The discussion is great, overall. One additional topic that you might consider including (space permitting) is the speed at which an inertial response leads to a change in position relative to the speed at which an increment of force leads to a change in position. Is there something about the lag time between a change in net force and the subsequent change in position (versus throwing one's weight around) that makes an inertial response especially effective? The answer may be no. Just something to think about.

Specific comments:

Line 53 – I suggest starting a new paragraph at “Birds...”

Line 63 – Preference thing, but it is not immediately obvious if “our observations” refers to observations during this study or before this study. It is not all that important, but changing the wording slightly (perhaps by removing “Based on our observations”) might clear up any minor confusion.

Line 79 – Please specify at what height from the gust generator these measurements were taken.

Line 92 – See my above comment about the aerodynamic impact of changes in wing elevation.

Line 93 – Wing pitch also seems to be proportionate to gust strength, as stated in the legend for figure 2. I would suggest rewording slightly to make that clearer.

Line 118 – Specify s.e.m or standard deviation

Figure 2 – The purple-green arrows on top of figure 2 are a bit confusing to me. I am not sure if they indicate a continuum between ‘inertia dominated’ and ‘aerodynamics dominated’ rejection, or if they are simply labeled for the arrows below. I feel like the latter is true. But you might consider the unintended interpretation.

Line 143 – Typo, add ‘the’ in “centre of _ plot”

Line 167 – I would appreciate a more detailed presentation of how the center of percussion is calculated for the owl wing. Perhaps you could include an excel sheet or script in which this parameter is calculated from the mass distribution of the wing.

The paragraph at 166 – Are there any differences in the center of percussion and center of pressure between the two wings? And is there any variation in the span-wise position of the center of pressure with the angle of attack in the CFD model?

Line 170 – Please specify that these measurements are from the CFD model.

Line 378 – How did the overall lift force calculated via CFD compare to the animal’s weight?

Line 533 – Because you have space in the supplemental material, I would suggest that you proceed very slowly through this calculation. It is an important component of your conclusion, so it would be great if a general reader could follow the math without too much effort. For example, the addition of K in the same step as substituting and rearranging takes some work to follow. A drawing may help. Also, should it be $\frac{K}{\rho A} \frac{dV}{dt}$ or simply $\frac{dV}{dt}$?

Figure S2 - I have trouble interpreting the top row of the figure without referencing the methods. Perhaps including labels like "top view" and "side view", and a profile of the owl (in gray, like in the lower row) in the top row would help.

Figure S4 - That's a dense mesh!

Author's Response to Decision Letter for (RSPB-2020-0905.R0)

See Appendix A.

RSPB-2020-1748.R0

Review form: Reviewer 1

Recommendation

Accept as is

Scientific importance: Is the manuscript an original and important contribution to its field?

Excellent

General interest: Is the paper of sufficient general interest?

Excellent

Quality of the paper: Is the overall quality of the paper suitable?

Acceptable

Is the length of the paper justified?

Yes

Should the paper be seen by a specialist statistical reviewer?

No

Do you have any concerns about statistical analyses in this paper? If so, please specify them explicitly in your report.

No

It is a condition of publication that authors make their supporting data, code and materials available - either as supplementary material or hosted in an external repository. Please rate, if applicable, the supporting data on the following criteria.

Is it accessible?

Yes

Is it clear?

Yes

Is it adequate?

Yes

Do you have any ethical concerns with this paper?

No

Comments to the Author

A good idea. Changes are acceptable.

Review form: Reviewer 2

Recommendation

Reject - article is scientifically unsound

Scientific importance: Is the manuscript an original and important contribution to its field?

Acceptable

General interest: Is the paper of sufficient general interest?

Good

Quality of the paper: Is the overall quality of the paper suitable?

Marginal

Is the length of the paper justified?

Yes

Should the paper be seen by a specialist statistical reviewer?

No

Do you have any concerns about statistical analyses in this paper? If so, please specify them explicitly in your report.

No

It is a condition of publication that authors make their supporting data, code and materials available - either as supplementary material or hosted in an external repository. Please rate, if applicable, the supporting data on the following criteria.

Is it accessible?

Yes

Is it clear?

Yes

Is it adequate?

Yes

Do you have any ethical concerns with this paper?

No

Comments to the Author

It's disheartening that the authors ignored the overwhelming majority of my initial critiques and presumably the critiques of the other reviewers as well. It's especially evident that the authors didn't even take time to correct some of the softball/easy fixes like changing wording or adding clarifying labels to figures. As such, my comments largely remain the same; however, the few changes that were made to this manuscript only bring up additional concerns.

Major comments:

1. This analysis ignores the effects of aeroelasticity on the gust response and attributes the gust rejection solely to the preflex mechanism. The reduction in peak vertical velocity is calculated by comparing the results of the in-flight (aeroelastic) bird data to a completely rigid CFD model. Wing and feather flexibility will play some, if not a large, role in the vertical translation/acceleration. Aeroelastic deformations are even observable in the S1 movie. The authors cannot attribute the rejection they are calculating solely to the preflex mechanism. The only way to numerically quantify the role of preflex from this CFD analysis would be to make turn the CFD model into an FSI model which accounts for the role of aeroelasticity. However, accurately modeling the aeroelasticity of a complete bird is an entirely different challenge in and of itself which I would not recommend. The simpler solution would be to report your observations directly from your experimental measurements. At the VERY LEAST, the authors should have addressed the role of aeroelasticity in their revised paper's discussion even if they cannot quantify how much it effects the aerodynamic rejection.
2. The CFD analysis has no validation. The authors have attempted to justify the validity of their results by claiming that the lift and drag is similar to other reported values, but the paper cited does not include data from an owl and the other species tested in that paper have a wide spread of lift and drag results (ranging from Cl max of 0.8 to 1.4). This is neither validation nor justification of the use of CFD. Comparison to other lift/drag data specifically from an owl would be needed. To make matters worse, the revised paper added that the "CFD simulations...produced a wake similar to published wakes for T. alba, and produced a polar similar to those published for other birds." without citing what data is being referred to.
3. Although the title claims that this preflex mechanism acts as a suspension system, this is neither quantitatively or qualitatively defined in the text. Their only justification for why this acts as a suspension system is because "the bird consistently maintained a stable trajectory of its head and torso". But that's not the definition of a suspension system. Luckily, suspension systems are very easily defined mathematically and thus, the only way to quantitatively claim this is to use the results to mathematically model the owl as a suspension system which is done using a mass-spring-damper analysis. Otherwise, you have not actually proven the claim stated by the title of your paper. You could, alternatively, change the title and remove comparisons to a suspension system.

General Comments:

"Implementing a similar preflex mechanism in future aircraft...". Phrasing is important here, and 'future aircraft' to the public implies transonic aircraft. The preflex shown here undergoes large scale displacement which would almost certainly be unsuitable for transonic aircraft. It's important to clarify that the target application is (presumably) small scale aircraft or UAVs.

Fig. 1B: The measurements at the peak of the gusts appear to have no error bars. Is this because they are small or missing? Similarly, the error bars on the edge of the gusts are quite large (up to 0.3-0.4 m/s (or upwards of 10-15%) at a distance of 0.4m into the gust) and even overlap with the error bars of the data points for the other flow velocities, yet no discussion of error in the analysis is really presented. The authors report an n of 6 (presumably for the error calculations) but in the methods section they state that anemometer measurements were taken at 50 hz for 30 s which totals 1.5k measurements. This is confusing. Shouldn't the latter number be used in error calculations? Lastly, the methods section states that the velocity measurements were collected in a 5x6 grid, so why is only one cross sectional profile shown? The full velocity could be plotted as a 3D contour map and would be more informative.

Figure 1C: The leading edge of the point clouds and the corresponding mesh have a significant amount of rough edges and features. Are you confident that these represent physical features observed on the bird or are they the result of errors in the 3D reconstruction? This is important because leading edge protrusions are well known to affect the flow, for example by tripping the flow from laminar to turbulent. This could affect the results of your CFD analysis.

More detail is needed about the CFD model earlier on in the text. Is it a static or dynamic model? Is it rigid or flexible? What turbulence model was implemented? Some of these details are later discussed in the methods section, but the reader needs a better understanding of the CFD was performed before introducing your results so they can assess them properly.

Line 111: This is the only time that the suspension system is discussed somewhat qualitatively. The kinematic characteristics or mathematical representation of a suspension system aren't even defined in the introduction or analysis. Furthermore, this data does not adequately demonstrate that the reflex mechanism acts as one. To truly prove that this mechanism acts a suspension system, you should use your experimental results to model it as a mass-spring-damper system which mathematically describes a suspension system.

Figure 2: It would be useful to superimpose the gust profile onto these plots so the reader can compare the velocity, elevation, and pitch response relative to the spatial strength of the gust. Furthermore, there is a fair amount of variability between test runs, so it would be useful to, for example, plot shaded error bands on the plotted averages.

The conservation of moment calculations would be useful to perform here to physically show the reader how this is true. However, I believe you also need to include the momentum of the fluid especially since you are assessing a dynamic gust.

The use of the word 'impulse' throughout the text is misleading. It appears that this calculation is simply the difference between the CFD and the experimental results. Impulse is defined as the force acting over time, but it does not appear that the authors performed time integration to actually calculate the impulse.

"At this stage..." "Later, as the wing pitched down..." It would be useful to mark these stages on the plots to direct the reader exactly to where these trends are occurring.

"As the wing pitched downwards, there was an increase in the difference between the measured vertical velocity of the whole bird center of mass and the rigid body simulation, indicating large aerodynamic-based gust rejection of $48 \pm 4\%$ of the potential impulse (Fig 3c)." The authors cannot confirm that this is due to the translation and rotation of the wings. The wing and feather aeroelasticity plays a substantial role in mitigating gusts, which is not accounted for here since the calculations are performed by taking the difference between the rigid CFD model and the flexible in-flight experiments. At the very least, the authors could acknowledge that the effects of aeroelasticity is present and that it's effects can't be isolated.

The authors state that "birds may then be able to tune the dynamics involved with gust rejection by modifying the stiffness of their shoulder joint". If the wing does indeed act as a suspension system as the authors claim, then changing the stiffness of the shoulder would effect the timing of the inertial rejection (dictated by mass-spring-damper equations of motion). Stiffening the shoulder muscle would cause the inertial rejection to reach it's maximum sooner. This is contradictory the statement made earlier, claiming that the inertial response must be passive since it is not faster than the reaction time measured in other studies. That is, unless the shoulder stiffness is assumed to already be at a maximum for these in-flight tests.

Figure 3: The results in Figure 3 are only presented for one gust intensity (medium), right? Do these results change at the lower and higher intensities? Since this was only assessed at one gust intensity, I don't think you can conclude that "passive mechanical properties of the shoulder joint and wing are responsible for immediate gust rejection" because this time frame was "equal to the minimum behavioural reaction time measured in other bird species". The data shown in Figure 2c indicates that for the high gust intensities, the wing elevation changes more rapidly (the response onset is sooner and the slope is steeper) but you have not yet proven that this difference between gust intensities is only due to the aerodynamic rejection.

Line 270: The authors have attempted to discuss the potential sensitivity of their main conclusion (specifically that the inertial effects are 'rapid and strong gust rejection mechanisms') to errors in their CFD simulations. First and foremost, a proper error or sensitivity assessment is not meant to determine whether or not the main conclusion is categorically true or false; instead, it is meant to quantify by HOW MUCH the data and/or conclusion is affected. I would also argue that the findings regarding the aerodynamic rejections are just as important as the inertial rejections (the data in Fig. 3 shows that the aerodynamic rejection substantially exceeds the inertial rejection, so perhaps it is actually more important?). Secondly, no documentation or data from this assessment is included with the manuscript's supplementary material. Furthermore, drag is frequently associated with errors substantially larger than 10% because the magnitude of drag forces is so small, so this does not seem like an appropriate % increase to test. It also seems unusual that the sensitivity analysis done for a 20% increase in lift but only a 10% increase in drag. Lastly, the conclusion that inertial rejection is "insensitive to our aerodynamic model" is arguably common sense. Since you were assessing the effects of errors in your aerodynamic loadings, it would have been more appropriate to assess the changes in aerodynamic rejection. Discussion on the sensitivity to the aerodynamic rejection is almost entirely ignored which is where you would logically see the largest impact of increasing the lift and drag values.

The surface accuracy of the reconstruction is said to be 1.67mm, however there are many features of the bird that will be well below this thickness (namely, the trailing edge feather thickness). How does that effect your reconstruction?

Are the mass properties segmented into each wing section (i.e. handwing, armwing etc.) or is the mass tracking done by translation and rotation of the whole wing mass? This could yield very different results if the whole wing is treated as one lump mass that can only undergo translation/rotation. Upon looking at the supplementary material, it appears that the mass is recorded for each voxel, so is each voxel mapped to the deformed bird geometry? Basically, are you calculating the mass distribution of the morphed wing by translating and rotating each voxel, or by translating and rotating each wing segment?

"The model was free to accelerate both horizontally and vertically, but could not rotate, as the real bird displayed no tendency to pitch during the gust." This is a major assumption. The lack of observation of the bird pitching during a gust does not mean that, in the absence of wing motions, the bird would not pitch. The wing's reduction in angle of attack is likely an active mechanism to, in part, minimize the pitching moments caused by the gust. When the gust hits the nose end of the bird first, this would cause a nose-up pitching moment. This has been shown in similar experiments done with gliders, so it is not accurate to assume that there would be no pitch in your rigid model.

The authors justify their turbulent model by claiming that it "is appropriate for bird flight simulations that require a high accuracy boundary layer"; however, their first inflation layer near the wall has a y^+ value of 10 which does not resolve the viscous sublayer. y^+ values of approximately 1 are recommended for best results using low Re (k - ω) models. The authors report a difference of 3% in lift between $y^+ 10$ and $y^+ 3$ results, but there would presumably be an even larger error between $y^+ 10$ and $y^+ 1$. And how does this affect the drag error? I appreciate the added details about convergence that the authors added in the resubmission

The authors justify their CFD results by claiming that their data is similar to the results presented in [33]. However, this citation does not have any aerodynamic coefficient data for an owl. Furthermore, the lift curves for all of the species tested range from a CL_{max} of 1.4 for the woodcock, to a minimum of CL_{max} of around 0.8 for the swift. This is a huge range of lift curves to compare results to, so it is not sufficient to simply say that your results "are similar to other published data (33)". Furthermore, in the resubmitted manuscript, the authors claim that their CFD produced wake distributions and polars similar to other works but did not cite the works to which they were referring to.

Review form: Reviewer 3 (Anthony Lapsansky)

Recommendation

Accept with minor revision (please list in comments)

Scientific importance: Is the manuscript an original and important contribution to its field?

Excellent

General interest: Is the paper of sufficient general interest?

Good

Quality of the paper: Is the overall quality of the paper suitable?

Excellent

Is the length of the paper justified?

Yes

Should the paper be seen by a specialist statistical reviewer?

No

Do you have any concerns about statistical analyses in this paper? If so, please specify them explicitly in your report.

No

It is a condition of publication that authors make their supporting data, code and materials available - either as supplementary material or hosted in an external repository. Please rate, if applicable, the supporting data on the following criteria.

Is it accessible?

No

Is it clear?

Yes

Is it adequate?

No

Do you have any ethical concerns with this paper?

No

Comments to the Author

Review for RSPB-2020-1748-Proof-hi

The manuscript is much improved, and I have no major concerns with the methods or conclusions. I do have quite a few comments to improve the clarity of the text, most of which are suggestions.

One overarching theme of those comments concerns the use of "trajectory" and "path" in the text. If possible, I would suggest removing these words wherever possible and focusing on impulse. It would also help to concisely explain why impulse is used to quantify gust rejection, rather than position, as early as the introduction. Maybe it is just me, but the plots in Figure 2B and Figure 3 evoke in my mind the flight paths of a bird through an upward gust. With the references to trajectory and path, it is hard to break away from viewing these plots as flight trajectories.

Thank you for addressing my comments from the previous version! The supplemental is much easier to follow and the text is much clearer.

Specific comments:

Line 36 – Specify that the center of percussion and mass are for the wing(s). The torso and head mentioned last, which may cause some confusion.

Paragraph at Line 66 – Two references are made to the glide path, but you have focused on the vertical impulse rather than the position of the bird. I think it would help if you explained how/why impulse was used to measure gust rejection. Right after the sentence ending in “ten high-speed video cameras,” you could add something like, “to quantify gust rejection, we compared the change in vertical velocity, hereafter impulse,...”.

Suggestions to improve the clarity of this paragraph:

“Here, we investigate how birds cope with rapidly changing airflow by studying the flight of a barn owl (*Tyto alba*), tracked using high-speed video-based 3-dimensional (3D) surface reconstruction, through a range of fan-generated vertical gusts. To successfully reject a gust perturbation, a bird must maintain the velocity of its torso and head through wing morphing, i.e. changing the shape and posture of their wings. We quantified how well the barn owl rejected the controlled gust through inertial mechanisms by comparing the impulse (change in velocity) of the barn owl’s torso and head to that of its center of mass, which includes the mass of the torso, head, and wing. In other words, we investigated how the movement of the mass of the wings mitigated the impulse experienced by the torso and head of the animal through the conservation of momentum. We quantified the aerodynamic effects of wing movement by contrasting the glide results of the live articulated bird to a model of the expected glide path for a rigid bird, with an aerodynamic polar computed from CFD (Fig 1D).”

Line 111-114 – Briefly define wing pitch and wing elevation. Alternatively, you could include a cartoon/drawing in Figure 2 with arrows to illustrate wing pitch and elevation. Currently, a lot is going on in Figure 2A and in Movie S2, so it is not clear to a novice reader what movements are elevation versus pitching (e.g. in pitching, the back of the wing “elevates” and in elevating, the wings pitch toward the midline).

Also, in Figure 2, both values start at 0, so I think that these are changes in wing elevation and wing pitch from the start of the gust. Is wing pitch/elevation, as included in the supplemental data sheet, calculated based on the horizontal?

Line 114 – Unclear what “proportion” refers to. Do the movements change in proportion relative to one another, or are they both proportional to gust intensity? Maybe “Both movements are likely to be important for gust alleviation and increase proportionally with gust intensity.”

Line 114-116 – Wing elevation also modifies the lift generated by the wings. You have done a great job at addressing the concerns of reviewers RE: the importance of wing pitch relative to other factors, but you are less successful here. I am guessing that the aerodynamic effect of wing elevation is also substantial. The effects of wing pitch might be more substantial. Either way, it might help to use the data you have from your CFD polar and data on wing elevation/pitch to roughly estimate the contribution of each mechanism. If the wings elevate up to 30 deg, that should be about a 15% drop in the vertical component of the lift vector ($\cos(30 \text{ deg}) = 0.866$). The change in lift based on wing pitch depends on the initial angle of attack but could be much larger based on your CFD-based lift polar. Pointing that out with some numbers might help placate reviewers/readers.

With that in mind, it might be interesting/convincing to include a plot of angle of attack vs. distance into the gust, accounting for the upward velocity of the gust. This is not required, but it might be convincing if the owl maintained a consistent angle of attack by pitching its wings forward despite the perturbation.

Figure 2 – I think it would be helpful to indicate that vertical impulse, wing elevation, and wing pitch plots are relative to pre-gust values. Including a delta symbol on the axes’ labels and a brief comment in the figure caption would accomplish this. To help the general reader understand wing pitch and wing elevation, consider adding a small cartoon version of each response to the corner of plots 2B through 2D.

Line 144 – Add a statement explaining why impulse, rather than position, was used to quantify gust rejection. Both “perturbation” and “movement” in this paragraph suggest that you tracked position instead of impulse. This would also be a good place to specify that velocity was calculated at the center of mass for the wing, torso/head, and whole bird, which I assume was the case.

Line 158 – for clarity, it would be helpful to define the instant of peak inertial rejection (e.g. “the point at which the impulse of the torso is most different from the impulse of the center of mass”)

Line 188 – Change to “The white dot” and add a period at the end of the sentence.

Line 188 – Figure 2B says Vertical Impulse whereas figure 3A says Normalized Impulse. I would suggest using “Normalized Vertical Impulse”.

Line 210 – I suggest introducing this paragraph briefly, if possible. Otherwise, it comes suddenly, given that the center of percussion is not mentioned in the introduction.

Line 295 – The text in this paragraph is not clear, at least to me. If space allows, it would be helpful to expand on this section.

Paragraph at 402 – Is this python script available upon request? Without it, I think it would be difficult to repeat the methodology described in this paragraph from the text alone. Does this python script take advantage of any specific libraries (OpenCV)? If so, please reference those libraries.

Line 416 – How was this accuracy defined? I asked because the value is negative.

Paragraph at 437 – As wing elevation and wing pitch are central components of this work, more specificity would help here. Specify which reference position (currently, the text says “a reference position”). At 0 m into the gust? Was the plane fitted to the top or bottom surface of the wing? What software was used segmentation and for the ICP Alignment method?

Let me know if you have any questions about these comments!

Regards,

Tony Lapsansky

anthony.lapsansky@umontana.edu

Decision letter (RSPB-2020-1748.R0)

21-Aug-2020

Dear Dr Cheney:

Your manuscript has now been peer reviewed and the reviews have been assessed by an Associate Editor. The reviewers' comments (not including confidential comments to the Editor) and the comments from the Associate Editor are included at the end of this email for your reference. As you will see, the reviewers and the Editors have raised some concerns with your manuscript and we would like to invite you to revise your manuscript to address them.

Research ethics:

Use of animals and field studies:

It is a condition of publication that you make available the data and research materials supporting the results in the article (<https://royalsociety.org/journals/authors/author-guidelines/#data>). Datasets should be deposited in an appropriate publicly available repository and details of the associated accession number, link or DOI to the datasets must be included in the Data Accessibility section of the article (<https://royalsociety.org/journals/ethics-policies/data-sharing-mining/>). Reference(s) to datasets should also be included in the reference list of the article with DOIs (where available).

Please submit a copy of your revised paper within three weeks. If we do not hear from you within this time your manuscript will be rejected. If you are unable to meet this deadline please let us know as soon as possible, as we may be able to grant a short extension.

Best wishes,
Dr Sasha Dall
mailto: proceedingsb@royalsociety.org

Associate Editor Board Member
Comments to Author:
Associate Editor: Doug Altshuler

The manuscript has now been seen by the three original referees who bring considerable expertise in both biology and engineering. The three referees did not reach consensus. Two are strongly in favor of acceptance, one as is and one with some additional minor concerns. The other referee raises three significant concerns that were not addressed from the first review. I am in agreement with the majority perspective that the work is sufficiently well performed, valuable and of broad interest for the readership of Proceedings B. However, it would be helpful to see direct responses to the remaining major concerns. This will be valuable for the editorial board as we consider the final decision. In the event of eventual acceptance, it will be important to see the authors' responses to these concerns in the online review history that will accompany the article.

Reviewer(s)' Comments to Author:

Referee: 2

Comments to the Author(s).

It's disheartening that the authors ignored the overwhelming majority of my initial critiques and presumably the critiques of the other reviewers as well. It's especially evident that the authors didn't even take time to correct some of the softball/easy fixes like changing wording or adding clarifying labels to figures. As such, my comments largely remain the same; however, the few changes that were made to this manuscript only bring up additional concerns.

Major comments:

1. This analysis ignores the effects of aeroelasticity on the gust response and attributes the gust rejection solely to the preflex mechanism. The reduction in peak vertical velocity is calculated by comparing the results of the in-flight (aeroelastic) bird data to a completely rigid CFD model.

Wing and feather flexibility will play some, if not a large, role in the vertical translation/acceleration. Aeroelastic deformations are even observable in the S1 movie. The authors cannot attribute the rejection they are calculating solely to the preflex mechanism. The only way to numerically quantify the role of preflex from this CFD analysis would be to make turn the CFD model into an FSI model which accounts for the role of aeroelasticity. However, accurately modeling the aeroelasticity of a complete bird is an entirely different challenge in and of itself which I would not recommend. The simpler solution would be to report your observations directly from your experimental measurements. At the VERY LEAST, the authors should have addressed the role of aeroelasticity in their revised paper's discussion even if they cannot quantify how much it effects the aerodynamic rejection.

2. The CFD analysis has no validation. The authors have attempted to justify the validity of their results by claiming that the lift and drag is similar to other reported values, but the paper cited does not include data from an owl and the other species tested in that paper have a wide spread of lift and drag results (ranging from Cl max of 0.8 to 1.4). This is neither validation nor justification of the use of CFD. Comparison to other lift/drag data specifically from an owl would be needed. To make matters worse, the revised paper added that the "CFD

simulations...produced a wake similar to published wakes for T. alba, and produced a polar similar to those published for other birds." without citing what data is being referred to.

3. Although the title claims that this reflex mechanism acts as a suspension system, this is neither quantitatively or qualitatively defined in the text. Their only justification for why this acts as a suspension system is because "the bird consistently maintained a stable trajectory of its head and torso". But that's not the definition of a suspension system. Luckily, suspension systems are very easily defined mathematically and thus, the only way to quantitatively claim this is to use the results to mathematically model the owl as a suspension system which is done using a mass-spring-damper analysis. Otherwise, you have not actually proven the claim stated by the title of your paper. You could, alternatively, change the title and remove comparisons to a suspension system.

General Comments:

"Implementing a similar reflex mechanism in future aircraft...". Phrasing is important here, and 'future aircraft' to the public implies transonic aircraft. The reflex shown here undergoes large scale displacement which would almost certainly be unsuitable for transonic aircraft. It's important to clarify that the target application is (presumably) small scale aircraft or UAVs.

Fig. 1B: The measurements at the peak of the gusts appear to have no error bars. Is this because they are small or missing? Similarly, the error bars on the edge of the gusts are quite large (up to 0.3-0.4 m/s (or upwards of 10-15%) at a distance of 0.4m into the gust) and even overlap with the error bars of the data points for the other flow velocities, yet no discussion of error in the analysis is really presented. The authors report an n of 6 (presumably for the error calculations) but in the methods section they state that anemometer measurements were taken at 50 hz for 30 s which totals 1.5k measurements. This is confusing. Shouldn't the latter number be used in error calculations? Lastly, the methods section states that the velocity measurements were collected in a 5x6 grid, so why is only one cross sectional profile shown? The full velocity could be plotted as a 3D contour map and would be more informative.

Figure 1C: The leading edge of the point clouds and the corresponding mesh have a significant amount of rough edges and features. Are you confident that these represent physical features observed on the bird or are they the result of errors in the 3D reconstruction? This is important because leading edge protrusions are well known to affect the flow, for example by tripping the flow from laminar to turbulent. This could affect the results of your CFD analysis.

More detail is needed about the CFD model earlier on in the text. Is it a static or dynamic model? Is it rigid or flexible? What turbulence model was implemented? Some of these details are later discussed in the methods section, but the reader needs a better understanding of the CFD was performed before introducing your results so they can assess them properly.

Line 111: This is the only time that the suspension system is discussed somewhat qualitatively. The kinematic characteristics or mathematical representation of a suspension system aren't even defined in the introduction or analysis. Furthermore, this data does not adequately demonstrate that the reflex mechanism acts as one. To truly prove that this mechanism acts a suspension system, you should use your experimental results to model it as a mass-spring-damper system which mathematically describes a suspension system.

Figure 2: It would be useful to superimpose the gust profile onto these plots so the reader can compare the velocity, elevation, and pitch response relative to the spatial strength of the gust. Furthermore, there is a fair amount of variability between test runs, so it would be useful to, for example, plot shaded error bands on the plotted averages.

The conservation of moment calculations would be useful to perform here to physically show the reader how this is true. However, I believe you also need to include the momentum of the fluid especially since you are assessing a dynamic gust.

The use of the word 'impulse' throughout the text is misleading. It appears that this calculation is simply the difference between the CFD and the experimental results. Impulse is defined as the force acting over time, but it does not appear that the authors performed time integration to actually calculate the impulse.

"At this stage..." "Later, as the wing pitched down...". It would be useful to mark these stages on the plots to direct the reader exactly to where these trends are occurring.

"As the wing pitched downwards, there was an increase in the difference between the measured vertical velocity of the whole bird center of mass and the rigid body simulation, indicating large aerodynamic-based gust rejection of $48 \pm 4\%$ of the potential impulse (Fig 3c)." The authors cannot confirm that this is due to the translation and rotation of the wings. The wing and feather aeroelasticity plays a substantial role in mitigating gusts, which is not accounted for here since the calculations are performed by taking the difference between the rigid CFD model and the flexible in-flight experiments. At the very least, the authors could acknowledge that the effects of aeroelasticity is present and that it's effects can't be isolated.

The authors state that "birds may then be able to tune the dynamics involved with gust rejection by modifying the stiffness of their shoulder joint". If the wing does indeed act as a suspension system as the authors claim, then changing the stiffness of the shoulder would effect the timing of the inertial rejection (dictated by mass-spring-damper equations of motion). Stiffening the shoulder muscle would cause the inertial rejection to reach it's maximum sooner. This is contradictory the statement made earlier, claiming that the inertial response must be passive since it is not faster than the reaction time measured in other studies. That is, unless the shoulder stiffness is assumed to already be at a maximum for these in-flight tests.

Figure 3: The results in Figure 3 are only presented for one gust intensity (medium), right? Do these results change at the lower and higher intensities? Since this was only assessed at one gust intensity, I don't think you can conclude that "passive mechanical properties of the shoulder joint and wing are responsible for immediate gust rejection" because this time frame was "equal to the minimum behavioural reaction time measured in other bird species". The data shown in Figure 2c indicates that for the high gust intensities, the wing elevation changes more rapidly (the response onset is sooner and the slope is steeper) but you have not yet proven that this difference between gust intensities is only due to the aerodynamic rejection.

Line 270: The authors have attempted to discuss the potential sensitivity of their main conclusion (specifically that the inertial effects are 'rapid and strong gust rejection mechanisms') to errors in their CFD simulations. First and foremost, a proper error or sensitivity assessment is not meant to determine whether or not the main conclusion is categorically true or false; instead, it is meant to quantify by HOW MUCH the data and/or conclusion is affected. I would also argue that the findings regarding the aerodynamic rejections are just as important as the inertial rejections (the data in Fig. 3 shows that the aerodynamic rejection substantially exceeds the inertial rejection, so perhaps it is actually more important?). Secondly, no documentation or data from this assessment is included with the manuscript's supplementary material. Furthermore, drag is frequently associated with errors substantially larger than 10% because the magnitude of drag forces is so small, so this does not seem like an appropriate % increase to test. It also seems unusual that the sensitivity analysis done for a 20% increase in lift but only a 10% increase in drag. Lastly, the conclusion that inertial rejection is "insensitive to our aerodynamic model" is arguably common sense. Since you were assessing the effects of errors in your aerodynamic loadings, it would have been more appropriate to assess the changes in aerodynamic rejection. Discussion on the sensitivity to the aerodynamic rejection is almost entirely ignored which is where you would logically see the largest impact of increasing the lift and drag values.

The surface accuracy of the reconstruction is said to be 1.67mm, however there are many features of the bird that will be well below this thickness (namely, the trailing edge feather thickness). How does that effect your reconstruction?

Are the mass properties segmented into each wing section (i.e. handwing, armwing etc.) or is the mass tracking done by translation and rotation of the whole wing mass? This could yield very different results if the whole wing is treated as one lump mass that can only undergo translation/rotation. Upon looking at the supplementary material, it appears that the mass is recorded for each voxel, so is each voxel mapped to the deformed bird geometry? Basically, are you calculating the mass distribution of the morphed wing by translating and rotating each voxel, or by translating and rotating each wing segment?

“The model was free to accelerate both horizontally and vertically, but could not rotate, as the real bird displayed no tendency to pitch during the gust.” This is a major assumption. The lack of observation of the bird pitching during a gust does not mean that, in the absence of wing motions, the bird would not pitch. The wing’s reduction in angle of attack is likely an active mechanism to, in part, minimize the pitching moments caused by the gust. When the gust hits the nose end of the bird first, this would cause a nose-up pitching moment. This has been shown in similar experiments done with gliders, so it is not accurate to assume that there would be no pitch in your rigid model.

The authors justify their turbulent model by claiming that it “is appropriate for bird flight simulations that require a high accuracy boundary layer”; however, their first inflation layer near the wall has a y^+ value of 10 which does not resolve the viscous sublayer. Y^+ values of approximately 1 are recommended for best results using low Re (k - ω) models. The authors report a difference of 3% in lift between $y^+ 10$ and y^+3 results, but there would presumably be an even larger error between $y^+ 10$ and y^+1 . And how does this affect the drag error? I appreciate the added details about convergence that the authors added in the resubmission

The authors justify their CFD results by claiming that their data is similar to the results presented in [33]. However, this citation does not have any aerodynamic coefficient data for an owl. Furthermore, the lift curves for all of the species tested range from a CL_{max} of 1.4 for the woodcock, to a minimum of CL_{max} of around 0.8 for the swift. This is a huge range of lift curves to compare results to, so it is not sufficient to simply say that your results “are similar to other published data (33)”. Furthermore, in the resubmitted manuscript, the authors claim that their CFD produced wake distributions and polars similar to other works but did not cite the works to which they were referring to.

Referee: 1

Comments to the Author(s).

A good idea. Changes are acceptable.

Referee: 3

Comments to the Author(s).

Review for RSPB-2020-1748-Proof-hi

The manuscript is much improved, and I have no major concerns with the methods or conclusions. I do have quite a few comments to improve the clarity of the text, most of which are suggestions.

One overarching theme of those comments concerns the use of “trajectory” and “path” in the text. If possible, I would suggest removing these words wherever possible and focusing on impulse. It would also help to concisely explain why impulse is used to quantify gust rejection, rather than position, as early as the introduction. Maybe it is just me, but the plots in Figure 2B and Figure 3 evoke in my mind the flight paths of a bird through an upward gust. With the references to trajectory and path, it is hard to break away from viewing these plots as flight trajectories. Thank you for addressing my comments from the previous version! The supplemental is much easier to follow and the text is much clearer.

Specific comments:

Line 36 – Specify that the center of percussion and mass are for the wing(s). The torso and head mentioned last, which may cause some confusion.

Paragraph at Line 66 – Two references are made to the glide path, but you have focused on the vertical impulse rather than the position of the bird. I think it would help if you explained how/why impulse was used to measure gust rejection. Right after the sentence ending in “ten high-speed video cameras,” you could add something like, “to quantify gust rejection, we compared the change in vertical velocity, hereafter impulse,...”.

Suggestions to improve the clarity of this paragraph:

“Here, we investigate how birds cope with rapidly changing airflow by studying the flight of a barn owl (*Tyto alba*), tracked using high-speed video-based 3-dimensional (3D) surface reconstruction, through a range of fan-generated vertical gusts. To successfully reject a gust perturbation, a bird must maintain the velocity of its torso and head through wing morphing, i.e. changing the shape and posture of their wings. We quantified how well the barn owl rejected the controlled gust through inertial mechanisms by comparing the impulse (change in velocity) of the barn owl’s torso and head to that of its center of mass, which includes the mass of the torso, head, and wing. In other words, we investigated how the movement of the mass of the wings mitigated the impulse experienced by the torso and head of the animal through the conservation of momentum. We quantified the aerodynamic effects of wing movement by contrasting the glide results of the live articulated bird to a model of the expected glide path for a rigid bird, with an aerodynamic polar computed from CFD (Fig 1D).”

Line 111-114 – Briefly define wing pitch and wing elevation. Alternatively, you could include a cartoon/drawing in Figure 2 with arrows to illustrate wing pitch and elevation. Currently, a lot is going on in Figure 2A and in Movie S2, so it is not clear to a novice reader what movements are elevation versus pitching (e.g. in pitching, the back of the wing “elevates” and in elevating, the wings pitch toward the midline).

Also, in Figure 2, both values start at 0, so I think that these are changes in wing elevation and wing pitch from the start of the gust. Is wing pitch/elevation, as included in the supplemental data sheet, calculated based on the horizontal?

Line 114 – Unclear what “proportion” refers to. Do the movements change in proportion relative to one another, or are they both proportional to gust intensity? Maybe “Both movements are likely to be important for gust alleviation and increase proportionally with gust intensity.”

Line 114-116 – Wing elevation also modifies the lift generated by the wings. You have done a great job at addressing the concerns of reviewers RE: the importance of wing pitch relative to other factors, but you are less successful here. I am guessing that the aerodynamic effect of wing elevation is also substantial. The effects of wing pitch might be more substantial. Either way, it might help to use the data you have from your CFD polar and data on wing elevation/pitch to roughly estimate the contribution of each mechanism. If the wings elevate up to 30 deg, that should be about a 15% drop in the vertical component of the lift vector ($\cos(30 \text{ deg}) = 0.866$). The change in lift based on wing pitch depends on the initial angle of attack but could be much larger based on your CFD-based lift polar. Pointing that out with some numbers might help placate reviewers/readers.

With that in mind, it might be interesting/convincing to include a plot of angle of attack vs. distance into the gust, accounting for the upward velocity of the gust. This is not required, but it might be convincing if the owl maintained a consistent angle of attack by pitching its wings forward despite the perturbation.

Figure 2 – I think it would be helpful to indicate that vertical impulse, wing elevation, and wing pitch plots are relative to pre-gust values. Including a delta symbol on the axes’ labels and a brief comment in the figure caption would accomplish this. To help the general reader understand wing pitch and wing elevation, consider adding a small cartoon version of each response to the corner of plots 2B through 2D.

Line 144 – Add a statement explaining why impulse, rather than position, was used to quantify gust rejection. Both “perturbation” and “movement” in this paragraph suggest that you tracked position instead of impulse. This would also be a good place to specify that velocity was calculated at the center of mass for the wing, torso/head, and whole bird, which I assume was the case.

Line 158 – for clarity, it would be helpful to define the instant of peak inertial rejection (e.g. “the point at which the impulse of the torso is most different from the impulse of the center of mass”)

Line 188 – Change to “The white dot” and add a period at the end of the sentence.

Line 188 – Figure 2B says Vertical Impulse whereas figure 3A says Normalized Impulse. I would suggest using “Normalized Vertical Impulse”.

Line 210 – I suggest introducing this paragraph briefly, if possible. Otherwise, it comes suddenly, given that the center of percussion is not mentioned in the introduction.

Line 295 – The text in this paragraph is not clear, at least to me. If space allows, it would be helpful to expand on this section.

Paragraph at 402 – Is this python script available upon request? Without it, I think it would be difficult to repeat the methodology described in this paragraph from the text alone. Does this python script take advantage of any specific libraries (OpenCV)? If so, please reference those libraries.

Line 416 – How was this accuracy defined? I asked because the value is negative.

Paragraph at 437 – As wing elevation and wing pitch are central components of this work, more specificity would help here. Specify which reference position (currently, the text says “a reference position”). At 0 m into the gust? Was the plane fitted to the top or bottom surface of the wing? What software was used segmentation and for the ICP Alignment method?

Let me know if you have any questions about these comments!

Regards,

Tony Lapsansky

anthony.lapsansky@umontana.edu

Author's Response to Decision Letter for (RSPB-2020-1748.R0)

See Appendix B.

Decision letter (RSPB-2020-1748.R1)

25-Sep-2020

Dear Dr Cheney

I am pleased to inform you that your manuscript entitled "Bird wings act as a suspension system that rejects gusts" has been accepted for publication in Proceedings B.

Open Access

Corresponding authors from member institutions (<http://royalsocietypublishing.org/site/librarians/allmembers.xhtml>) receive a 25% discount to these charges. For more information please visit <http://royalsocietypublishing.org/open-access>.

Paper charges

Sincerely,

Dr Sasha Dall

Associate Editor:

Comments to Author:

Associate Editor: Doug Altshuler

The authors have now revised the manuscript in response to the second round of reviews. I am satisfied that they have addressed the referees' concerns to a sufficient degree that it is now ready for our readership. I find this result to be interesting and important.

Appendix A

Comments to Author:

Associate Editor: Doug Altshuler

The authors have combined live measurements from flying owls and computational fluid dynamics on an owl model to argue that birds are able to compensate for gusts using a reflex that reduces head and body motion. I agree with the three referees that these are very exciting results that are likely to have a significant impact in the field of animal flight. The three reviews contain a number of suggestions for revision but one common theme was that the presentation was a bit dense. The writing was good but there is a lot of material, and some methodological concerns about the CFD were raised. I'm not sure whether the best option is to save the CFD for a separate manuscript, as suggested directly by one of the referees, or whether further editing can handle this. In any case, it would be helpful to see a revised manuscript that is responsive to these reviews.

Thank you for the feedback and organizing the referees and their comments.

As Reviewer 3 identified, we are stuck between a rock and a hard place, in that we must expand on the manuscript without really making it longer. We do entirely agree with Reviewer 3 that we should keep story simple and attractive to the general readership of Proc R Soc B, while we can also see the requirement for further details on methodology (particularly relating to CFD), and the robustness of the conclusions.

The solution to the rock and hard-place challenge has to be extensive use of Supplementary Information. This does mean, however, that some of the issues raised by the reviewers will not be explicitly addressed at the points in the manuscript requested: certain details are considered of interest to the specialist but would be distracting to the general reader. In doing so, we believe that we have achieved a manuscript that is approachable by a general audience, and now sufficiently detailed with the SI to satisfy a specialist reader.

It was suggested that we could remove the CFD from the work. We did not feel that this it was possible to remove the CFD—and aerodynamic modelling—from this work if we are to ‘tell the story’. It is integral to this work and while without it, we can compute the absolute magnitude of inertial effects, we cannot place those effects in context. What good is an inertial reflex, if it only account for a small fraction of the gust rejection? We believe our sensitivity analysis now demonstrates that any potential error in the CFD is not critical to the conclusions.

The CFD analysis and the model it parameterizes allows ‘counterfactual’ possibilities to be considered, thereby advancing the manuscript from merely reporting a series of observations on towards describing the mechanisms underlying the observation.

--

Below, we highlight particular concerns and how we addressed them:

The referees wondered about wing morphing and how we could justify neglecting it. To address this we added a new supplemental video to demonstrate that while the wings morph, they do not do so substantially (as shown by wing outlines) throughout the period from the start of the gust to the moment of peak inertial rejection.

Importantly, we have fixed an imprecise statement that suggested aerodynamic rejection could be explained solely by wing pitch. This was never intended and was inconsistent with the rest of the manuscript.

We have added substantial detail about the CFD and clarified its purpose by adjusting the wording to differentiate better between the quasi-steady gliding model and CFD simulation. Among the details about the CFD is information providing validation, and an exploration on the sensitivity of our conclusions to potential systematic error.

Reviewer(s)' Comments to Author:

All line numbers refer to the manuscript with track changes accepted

Referee: 1

Comments to the Author(s)

General comments:

The role of wing flexibility is not discussed anywhere in this paper despite it having a large potential impact on the interpretation and conclusions.

Wing flexibility could play an important role in both inertial and aerodynamic rejection.

With regards to inertial dynamics, we have added a supplemental video displaying negligible wing morphing effects on the wing's centre of mass over the period from the start of the gust to moment that inertial gust rejection peaks. At some point thereafter, wing flexibility will affect our ability to accurately resolve its centre of mass, but as we demonstrate, that moment is late enough to not affect our conclusions.

Concerning the aerodynamics, as shown: aerodynamic rejection begins to grow after the moment of peak inertial rejection. It is over this duration that obvious wing morphing occurs. It is not the intention of this paper to assess how the bird rejects the gust through aerodynamic means.

We have rectified the statement that suggested wing pitch was sufficient to explain the aerodynamic rejection. It is likely a substantial effect (20 degree change in angle of attack), and it aligns well with the timing of aerodynamic rejection, but it is certainly not the only effect. Elsewhere in the text (*e.g.*, line 89 & 255), we specifically referenced additional aerodynamic mechanisms, but line 93 was inconsistent. It now reads as:

Line 111: The bird achieved this, in part, by two dramatic movements: an almost instantaneous change in wing elevation (Fig 2A,C, Movie S2), followed by a substantial reduction in wing pitch angle (Fig 2D).

No comment is made in the discussion about the differences between their three gust test conditions (low, mid, high) after introducing the idea in Figure 1 and showing the different shapes and velocities in Figure 2.

We have added a statement to explain why the gust condition is ignored with respect to quantifying gust rejection. This statement was best supported with an additional figure which we have added as SI, reproduced below, along with the text in the main manuscript that cites it:

Line 154: This inertial effect reduced the impulse to the torso (Fig 3B). The dose-dependent kinematics resulted in wing inertia rejecting a greater magnitude of the gust impulse with increasing gust intensity, but the proportion of the total expected impulse rejected by wing inertia did not discernably differ across gust intensities (Fig S1). At the instant of peak inertial rejection, wing inertia rejected $32 \pm 5\%$ (mean \pm s.e.m. n=9 flights) of the total expected gust impulse.

Line 113: How does the force provided by the gust come into play here? The conservation of momentum will be the inertial effect of the wings, inertial effect of the torso and the aerodynamic force due to the gust. A net positive acceleration of the wing mass does not necessarily indicate a net downward (negative) acceleration of the body unless all other forces acting on the body is zero, which is not the case here. Also the previous sentences and figure show that the torso does not have downward momentum just a lower value than the wings.

We have added expanded upon the logic within the manuscript, and also expanded the SI's discussion of the mechanics. See response two below.

Line 117: Does aerodynamic rejection refer to the difference between the rigid CFD body to translate vertically compared to the whole bird CG? If so this is not really an aerodynamic rejection but rather another inertial mechanism?

Similar to above, we believe the expanded text (see response below) introduces the quantities in a clearer manner, and the new SI allows readers to dig into the mechanics.

As to whether this is an “inertial mechanism,” the literature is not particularly consistent and therefore useful here. Hence, we define our terms in the manuscript, and now provided some additional mathematical reasoning.

I believe that their statements on the contribution of different components contributions to conservation of momentum should be backed up by the actual calculation. It comes off as qualitative rather than quantitative.

We have expanded upon the logic within the text and added SI to address this. We can add the equations discussed in the SI to the manuscript, but we currently feel it is unnecessary with the additions made.

Line 135: We calculated the magnitude of gust rejection from the perspective of minimising the perturbation applied to the torso. *Inertial gust rejection* describes the difference between the observed movement of the torso and that of the whole-bird centre of mass, as calculated using the mass-calibrated CT scan (Fig 3A,B); inertial gust rejection is due to forces internal to the bird, *i.e.*, passive or active elevation of its wing masses with respect to the torso. *Aerodynamic gust rejection* describes the difference between the observed movement of the centre of mass and that expected from the quasi-steady, rigid-bird glide model with CFD-derived polar (Fig 3A,C); aerodynamic gust rejection is due changes in the external aerodynamic force. Both gust rejection metrics compare theoretical expectations for a rigid bird to observations of our live articulated bird. We quantified these gust rejection magnitudes by their impulse (Supplementary text).

SI:

Gust rejection

We compute impulse as

$$\int_0^t F dt$$

We separate the impulse acting on the torso into two components: the external impulse acting on the centre of mass of the system, *i.e.*, the bird; and the internal impulse acting on the torso applied within the system. The internal impulse has no effect on the centre of mass, and is a result of the bird changing shape, *i.e.*, movement of inertial appendages. The external impulse produces movement of the centre of mass, and is a result of the net aerodynamic and gravitational forces acting on the bird.

An important aspect of this approach is that it is agnostic to any specific mechanism. There are no assumptions about why or how the bird changes shape, nor does it provide any specific insight into that question. It merely separates the resultant impulses/forces.

We compute aerodynamic rejection from the difference between the expected and observed aerodynamic force. The expected force was computed from the glide simulation utilizing the aerodynamic polar derived from CFD. The observed force was computed from movement of the bird's centre of mass.

$$\int_0^t (F_{CFD} - F_{CoM}) dt$$

$$M * (v_{CFD} - v_{CoM})|_0^t$$

where M is the total mass. To determine the fraction of the external impulse attributed to the torso, we normalize by the mass fraction m_{torso}/M .

$$m_{torso} * (v_{CFD} - v_{CoM})|_0^t$$

We compute inertial rejection as the impulse acting within the system on the torso. We'll first discuss the logic to the approach and then discuss the physics.

The inertial rejection does not explicitly consider any effects of aerodynamic force. Inertial rejection is the difference between the expected force on the torso if the bird did not elevate its wings, and the observed force acting on the torso. If the wings did not elevate, the torso would accelerate at the same rate as the observed centre of mass:

$$F_{torso, expected} = m_{torso} a_{CoM}$$

And the observed force acting on the torso would be:

$$F_{torso, observed} = m_{torso} a_{torso}$$

The rejected inertial impulse would then be

$$m_{torso} \int_0^t (a_{CoM} - a_{torso}) dt$$

$$m_{torso} * (v_{CoM} - v_{torso})|_0^t$$

We can find that the quantity above is due to internal forces responding to movement of the wings through manipulating and differentiating the equation for the centre of mass.

$$m_{torso} x_{torso} + m_{wing} x_{wing} = M x_{CoM}$$

$$m_{torso} a_{torso} + m_{wing} a_{wing} = M a_{CoM} = F_{external}$$

$$m_{torso} a_{torso} + m_{wing} a_{wing} = (m_{torso} + m_{wing}) a_{CoM}$$

$$m_{torso} a_{torso} + m_{wing} a_{wing} = m_{torso} a_{CoM} + m_{torso} a_{CoM}$$

$$m_{torso} (a_{torso} - a_{CoM}) + m_{wing} (a_{wing} - a_{CoM}) = 0$$

Yielding that the internal force acting on the torso through acceleration of the wings (and Newton's third law makes the opposite true as well) is:

$$m_{torso} (a_{torso} - a_{CoM}) = -1 * m_{wing} (a_{wing} - a_{CoM})$$

Which, as discussed above, is the inertial rejection once integrated over time.

Regarding CFD, while the code is listed it might be nice to mention what flow assumptions are assumed for those not familiar with the code, i.e. is it low speed? Etc

We expanded the methods with more detail:

Line 509: The CFD simulations treated air as an incompressible and viscous flow. At the speed of the flights, Mach number was approximately 0.02, which suggests air compressibility was negligible. Turbulent effects though are not negligible. For viscous flow at Reynolds numbers for the gliding flights ($\sim 8.8 \times 10^4$ as defined by mean chord length and flight speed), turbulent effects

are significant. To fully simulate the turbulence would require direct numerical simulation (DNS) and a fluid mesh density to capture the Kolmogorov scale. As DNS is computationally impractical for bird flight, we approximated the behaviour of the fluid with the $k-\omega$ SST turbulence model in ANSYS Fluent (version 19.1; ANSYS Inc., Canonsburg, USA). The SST model uses a blending function which combines the merits of the standard $k-\omega$ model, for near-surface simulation, and the standard $k-\epsilon$ model, for the domain away from the surface, which is appropriate for bird flight simulations that require a high accuracy boundary layer. We compared the effects of different turbulence models (Spalart-Allmaras, Reynolds Stress, $k-\epsilon$ standard, and $k-\omega$ standard) on the computed values of lift and drag, and found all models were within 5% of each other for a 5° angle of attack simulation. The criteria for simulation convergence was determined by the unitless ‘scaled’ residuals of three quantities, when: the continuity residual reached $<5 \times 10^{-3}$; the orthogonal velocity-component residuals reached $<5 \times 10^{-7}$; and the turbulence kinetic energy reached $<3 \times 10^{-4}$. Turbulence kinetic energy was typically the last criterion met. The ‘scaled’ residuals are determined by normalising the absolute residuals by the maximum residual value among the first five iterations of the simulation.

Specific line references:

Line 55: Tennekes book is not a reliable source of information. This book does not cite any of the original work that it references, nor does it explain how it arrives at its stated conclusions. See Pennycuick’s review of the original work: <https://www.nature.com/articles/381126b0.pdf?origin=ppub>

We have changed the citation to include a pair of papers focused on 1) wind gust speeds and 2) bird flight speeds.

Hewston R, Dorling SR. 2011 An analysis of observed daily maximum wind gusts in the UK. *J. Wind Eng. Ind. Aerod.* **99**, 845-856. (DOI: 10.1016/j.jweia.2011.06.004)

Alerstam T, Rosén M, Bäckman J, Ericson PGP, Hellgren O. 2007 Flight speeds among bird species: allometric and phylogenetic effects. *PLoS Biol.* **5**, e197. (DOI: 10.1371/journal.pbio.0050197)

Line 92: About what axis is wing pitch angle defined?

We use the shoulder to the wingtip as the pitching axis. We have added a supplemental figure to show the owl’s reference postures, as below in miniature.

Figure 2A: There visually appears to be asymmetries in the wing shape on either side. Are the presented results for one wing or for an average of both wings?

There are asymmetries between wing shape and pose, not just during the gust as in 2A, but even in steady gliding. The kinematics in figure 2 display the average of the two wings. However, when aligning the CT scan, we aligned it to the left wing only, so calculated inertial quantities depend upon only the left wing's inertia.

We clarify this in figure 2's caption and the methods.

Line 131: (C,D) Kinematics represent the average of the two wings.

Line 456: We assumed that the inertial dynamics of the left wing equal those of the right wing.

Figure 2B,C,D: It may be more informative to plot these panels with the mean gust velocity similar to Fig. 3a

The plots are cropped to contain the kinematics during the gust, and do not cover any region beyond this. We have added to the caption to better orient the reader to this:

Line 130: (B-D) Presented kinematic data encompasses the space occupied by the '1-cosine' gust, and does not extend beyond it.

We chose not to add the gust velocity in the background. The plots are already quite busy visually and adding to them seems unnecessary when they only contain the '1-cosine' profile of the gust. We are open to it and will leave it as a decision for the editor.

Line 109: Where is the origin of the torso and the CG of the whole bird? The materials section (Line 351) states that a single glide configuration was scanned. So, does that mean that the whole bird CG and wing CG was calculated by assuming a rigid body rotation of the wings about the shoulder? Has the effect of mass distribution change due to the wing morphing been accounted for in this calculation? If so how?

While we refer to the CG movement of the CG of the torso, our calculation does not explicitly require the position of the CG of the torso. We focus entirely on the time-derivatives of position. For a rigid model that cannot rotate, the vertical velocity/acceleration of all points are equal.

When we compute the CG of the whole bird, we do allow for the wings to undergo both rigid-body rotation and translation. We allow translation for two reasons; 1) many joints are not well represented by a perfect pivot at either the obvious anatomical region or elsewhere for large rotations, making translations necessary (many XROMM studies demonstrate this); and 2) we did not want to entirely neglect wing morphing, and by allowing the rigid body to translate, it would at least partially capture flexion of the elbow, when it does occur. As we discuss above, this elbow flexion does not occur over the time-domain of interest, but it likely would have some small effect on the exact aerodynamic rejection quantities late in the gust.

We have added to the methods

line 448: The mass properties of the bird were calculated based on CT-scan data from a naturally deceased barn owl of similar overall size. The cadaver weighed 296 g, and the live owl 319 g. The cadaver, with wings extended was depth-scanned (LightSpeed RT16 CT scanner; General Electric, Boston, USA) to produce a 3D voxel array in 16-bit grayscale. Calibration phantoms

(Gamex 467, Sun Nuclear, Melbourne, Australia), scanned alongside the bird, supplied the necessary calibration for conversion of grayscale values to tissue density and, in turn, mass distribution of the bird using the known voxel dimensions. The segmented torso and left wing from the scan were aligned with the point clouds to estimate the mass distribution of the bird in flight. We assumed that the inertial dynamics of the left wing were equal to those of the right wing. The CT and point cloud alignment was performed for a single typical glide posture to establish the mass and centre of mass position for each point cloud segment. Centre of mass position was then tracked subsequently when aligning point clouds using rigid-body transforms that allowed for rotation and translation. This approach was ideal for the duration that wing flexion was negligible (*e.g.*, the time-course dominated by the inertial rejection mechanics. Movie S3). However, the effect of wing shape change on the centre of mass was not completely ignored because we allowed the wing to translate. For example, when the elbow flexed, the rigid-body transform that best fit the flexed state of the wing moved the centre of mass proximally as expected.

Line 115/117: The nomenclature of “impulse” suggests that the impulse due to the gust has been calculated which is the time integral of the resultant force in Newton-seconds however in Fig. 3 this is a velocity measurement of meters/second. I assume this is the velocity difference between the different components of the bird. Perhaps change the nomenclature throughout the paper to “rejected velocity”?

We have made changes to be more consistent and clarify that the quantity of gust rejection is impulse or, for plotting purposes, mass-normalized impulse. Additionally, we expanded upon the SI the approach we use to compute impulse, and how differences in impulse lead to rejection terms. In figure 3B-D, we display the data as a mass-normalized impulse, which results in an intuitive quantity: a change in velocity. Figure 3A) does display absolute velocity, which is why all values are not zero at onset of gust.

We have modified the caption of figure 3 to explain this:

Line 181: Vertical arrows in the centre of the plot indicate how differences in paired parameters determine the components of gust rejection in the plots below (coded by colour). Mass-normalised impulse was computed from the onset of the gust.

Line 170: Is this elliptical lift distribution in contrast to the recently published JEB paper? <https://jeb.biologists.org/content/223/3/jeb214809>

No, it is wholly consistent. As you can see in that paper’s figure 5: on the wing, both models of optimum efficiency are similar. The distribution of lift coefficient deviates behind the tail, not meaningfully on the wings. Note in our figure 4, there is no tail/torso.

We have deleted that statement as it is an unnecessary comparison. Assuming an elliptical distribution over the wing would produce a similar centre of pressure as the one we found, but we do not need to assume here. Instead we present a more rigorously determined value.

Line 201: What is the minimum time required for muscle activation? How do you know that 80ms was too fast for the muscles to activate? Or that they hadn’t actively “tensed” up before entering the gust?

Provide citations and further discussion to support this sentence. This is a big result of your paper and mentioned multiple times in the abstract and intro.

The time-scale of interest does not concern only muscle activation speed, but the full behavioural complement of sensing, processing, and muscle activation. This manifests as a time lag between the stimulus and the action. We are not concerned about whether there is a time lag between the gust and the rejection, because the whole inertial rejection event occurs within the domain of the time lag. While this demonstrates that inertial rejection is likely a passive reflex, it needs a mechanism to explain the effect, which is the centre of percussion mechanics.

We have added to the discussion the explanation that the mechanics are intrinsically instantaneous. The mechanism is a byproduct of having a shoulder. The passive mechanics of a hinged-wing require that the wing elevate and reject the gust at the torso. What we have demonstrated is that this rapid mechanism plays a substantial role in torso stabilisation.

We have added two additional citations concerning the minimum reaction time in birds, both concern motor delays during flight: 1) the delay of the visuomotor system in pigeons; and 2) the delay in head turning as response to an acoustic stimulus in barn owls.

Line 243: The inertial-based rejection reaches a maximum after a mean period of 80.7 ms, which is equal to the minimum behavioural reaction time measured in other bird species (27-29). The proposed mechanism for gust rejection is intrinsically instantaneous, and is an effect of having a shoulder that is free to elevate under increased load, which helps to explain how the entire inertial response can occur within the reflex delay. The speed of the event suggests that the CNS cannot modulate shoulder muscle activity sufficiently quickly and, therefore, that the passive mechanical properties of the shoulder joint and wing are responsible for immediate gust rejection. Consistent with this, wing shape is essentially constant from the start of the gust through the moment of peak inertial rejection (Movie S3). After this inertial-based rejection, the pitch and shape of the wing begin to change (Movie S2), altering the aerodynamic forces produced by the wings on a time scale that allows for potential CNS control.

If the bird modified the pre-tension of the agonist and antagonist muscles of the shoulder, this would tune the reflex - the passive mechanical response of the system (somewhat analogous to a feed-forward response). We are preparing a further manuscript dedicated to detailing the effect of muscle stiffness on gust rejection.

Line 228: Does this mean that this bird was found to be optimally loaded? This is a big statement that requires further explanation and supporting numerical data.

No, far from it. It's a reason (one of many) why birds may not be optimally loaded.

Line 383: It is not clear how comparison of the CFD mesh to trial recordings demonstrated "that the bird exhibited similar and consistent approach postures in all flights"

We have rewritten that statement to improve its clarity.

Line 487: Together these demonstrate that the geometry is representative of the posture prior to the gust, and throughout the glide when no gust was present.

Line 378: CFD at lower Reynolds numbers can be difficult. Provide... some validation of the CFD method and discuss how this could impact the results.

We have added some validating information for our CFD:

Line 529: The output of the CFD simulations was sufficient to provide weight support, produced a wake similar to published wakes for *T. alba*, and produced a polar similar to those published for other birds. Lift accounted for 94.2% of body weight after accounting for acceleration. The wake, as visualised by q-criterion, was similar to wakes measured behind the same individual, and the spanwise downwash distribution was also qualitatively and quantitatively similar to that measured using particle tracking (31, 32). Finally, the simulated lift-drag polars were consistent with other published data (33). Importantly, they exhibit similar values for coefficient of lift (~0.8) at high (>20) angle of attack, for which much of the glide model operates; but note our drag values are larger due to the inclusion of the body (Fig S5).

We now also describe the sensitivity of our centre of pressure calculation (as a function of angle of attack), and the sensitivity of our results to the aerodynamic polar computed from CFD.

Line 215: The position of the centre of pressure was stable with regards to changes in angle of attack, moving outboard by 0.3% of wing length per degree increase.

Line 270: The conclusion that inertial rejection is both a rapid and strong gust rejection mechanism in birds is robust to systematic error in the CFD-derived aerodynamic polar. At its peak, inertial rejection exceeded aerodynamic rejection regardless of whether we applied a 10% increase in drag across the drag curve, a 20% increase in lift to the linear portion of the lift curve, or a moderate 6.5% increase in lift across the entire lift curve (detailed in supplementary text). None of these changes affected the absolute magnitude of inertial rejection, but they affected the quasi-steady glide model, which in turn determined the relative magnitudes of inertial and aerodynamic rejection. The relative impact of inertial rejection and its insensitivity to our aerodynamic model provides confidence in the substantial role played by inertial effects in avian gust rejection. Further, we have conservatively estimated the effect of inertial rejection by ignoring the added fluid mass around the wings; accounting for the added mass would have further enhanced inertial rejection.

And within the SI:

Sensitivity of gust rejection to potential error in the aerodynamic polar

Our results are insensitive to systematic error in the CFD-derived aerodynamic polar. For this analysis we only examined the average relative contribution of inertial and aerodynamic rejection at the instant of peak inertial rejection. A 10% increase in drag decreased inertial rejection from 32% to 31% and increased aerodynamic rejection from 6% to 8%. Similarly, increasing the lift curve slope by even 20% over the linear portion (<10°) decreased inertial rejection by just 2% (to 30%) and increased aerodynamic rejection by 6% (to 12% of the total rejection). Increasing lift across the whole aerodynamic polar demonstrates that it is more sensitive to results at higher angle of attack, but even applying a moderate increase still did not affect our conclusions. We applied a 6.5% increase in lift coefficient—the amount required to achieve 100% weight support in the CFD simulations at 5 degrees angle of attack (the observed orientation and at the observed

speed), but treated as systematic error across the whole aerodynamic polar— and this decreased inertial rejection from 32% to 27% and increased aerodynamic rejection from 6% to 21%. Even with this systematic increase in lift applied across the polar, inertial rejection was still 30% more effective than aerodynamic rejection at its peak.

Provide the CFD test Reynolds number,

The Reynolds number:

Line 511: Reynolds numbers for the gliding flights ($\sim 8.8 \times 10^4$ as defined by mean chord length and flight speed)...

Line 558: What method was used to calculate the appropriate moment of inertia about the hinge calculated? And was this assumed to be constant throughout the sequence?

Within the mass-calibrated CT scan we could resolve the humeral head. We used this point to resolve whether voxels belonged to the torso or to the wing. The moment of inertia was only used in the calculation of the centre of percussion in figure 4. The centre of gravity of the wing, also computed from the mass-calibrated CT scan, was tracked throughout a sequence and no strict pivot was enforced, which allowed for translation.

The inertial rejection of the wing occurs in response to the wing moving. The moment of inertia and centre of percussion discussion explains how the wing moves passively with negligible lag or delay. These are general passive mechanical phenomena that happen when an object experiences a force about a hinge.

We have added more information about the tracking at:

Line 457: The CT and point cloud alignment was performed for a single typical glide posture to establish the mass and centre of mass position for each point cloud segment. Centre of mass position was then tracked subsequently when aligning point clouds using rigid-body transforms that allowed for rotation and translation. This approach was ideal for the duration that wing flexion was negligible (*e.g.*, the time-course dominated by the inertial rejection mechanics. Movie S3). However, the effect of wing shape change on the centre of mass was not completely ignored because we allowed the wing to translate. For example, when the elbow flexed, the rigid-body transform that best fit the flexed state of the wing moved the centre of mass proximally as expected.

Referee: 2

Comments to the Author(s)

This work assesses how birds change their wing shape in response to vertical gusts by combining experimental testing of live owls and computational fluid dynamic modeling of a rigid owl model. The authors aim to show that bird wings experience passive upwards translation and rotation (preflex) in response to vertical gusts in order to reduce the translation of the head and body, allowing for more stable flight. This response is said to mimic a 'suspension system'. The authors have undoubtedly taken on an ambitious task to understand a complex problem. While the experiments are fairly strong and show interesting results, the integration of the CFD analysis is flawed. It is my opinion that this paper would be much stronger if the authors eliminate the CFD analysis and expanded the assessment of their experimental analysis.

Major comments:

1. This analysis ignores the effects of aeroelasticity on the gust response and attributes the gust rejection solely to the preflex mechanism. The reduction in peak vertical velocity is calculated by comparing the results of the in-flight (aeroelastic) bird data to a completely rigid CFD model. Wing and feather flexibility will play some, if not a large, role in the vertical translation/acceleration. Aeroelastic deformations are even observable in the supplemental S1 movie. The authors cannot attribute the rejection they are calculating solely to the preflex mechanism. The only way to numerically quantify the role of preflex from this CFD analysis would be to make the CFD model into an FSI model which accounts for the role of aeroelasticity. However, accurately modeling the aeroelasticity of a complete bird is an entirely different challenge in and of itself which I would not recommend. The simpler solution would be to report your observations about the wing/body translation and rotation directly from your experimental measurements. While I understand that this would eliminate Fig. 3, you could replace it with an assessment of how the mass distribution is changing in time as the owl flies through the gust. Plots on the tail's response during the gust would also be useful, as video S1 shows that the tail experiences an upwards pitching motion followed by a downwards pitching motion, the latter of which I presume is a landing response.

We do not attribute the whole rejection to the preflex. The preflex explains how the wing can respond to the gust perturbation in such a short time scale. The data show that the preflex primarily influences gust rejection by inertial dynamics. While we are aware that the wing elevation of the preflex movement will also modify aerodynamic force production, there are additional aerodynamic rejection mechanisms acting beyond the preflex. We specifically highlight one of those by identifying the 20 degree change in wing pitch; aeroelasticity is yet another mechanism.

In the introduction and results, we have elaborated further to better differentiate the mechanics of gust rejection due to inertia and aerodynamics.

Line 87: Based on these observations we propose two independent mechanisms for gust rejection in birds: 1) an inertial mechanism, whereby the relative motion of the mass of the wings reduces the motion of the torso and head; and 2) an aerodynamic mechanism based on changes in the orientation and shape of the wings, including decreasing their angle of attack, which reduces the lift generated. These two mechanisms separate the forces into those acting internally, the forces acting on the joints responsible for moving the segments of the bird relative to the centre of mass,

and those acting externally, responsible for moving the centre of mass, and producing locomotion.

2. The CFD analysis has no validation. The authors have attempted to justify the validity of their results by claiming that the lift and drag is similar to other reported values, but the paper cited does not include data from an owl and the other species tested in that paper have a wide spread of lift and drag results (ranging from C_L max of 0.8 to 1.4). This is neither validation nor justification of the use of CFD. Comparison to other lift/drag data specifically from an owl would be needed.

411: The authors justify their CFD results by claiming that their data is similar to the results presented in [25]. However, this citation does not have any aerodynamic coefficient data for an owl. Furthermore, the lift curves for all of the species tested range from a C_{Lmax} of 1.4 for the woodcock, to a minimum of C_{Lmax} of around 0.8 for the swift. This is a huge range of lift curves to compare results to, so it is not sufficient to simply say that your results “are similar to other published data (25)”.

We have further validating information for our CFD:

Line 531: Lift accounted for 94.2% of body weight after accounting for acceleration. The wake, as visualised by q-criterion, was similar to wakes measured behind the same individual, and the spanwise downwash distribution was also qualitatively and quantitatively similar to that measured using particle tracking (31, 32). Finally, the simulated lift-drag polars were consistent with other published data (33).

Concerning the variance across species’ lift-curves:

Withers’ plots are somewhat challenging to follow. The woodcock actually reaches peak C_L around ~ 1 (1.05?), as seen most clearly on the left plot (also visible on the right plot, but it can be easily confused for the C_L/C_D curve.).

The nighthawk and quail are the best performers, but they only reach C_{Lmax} of 1.1 to 1.05 (depending on whether you fit a curve through the data or accept each point as true). This does not dismiss this point, but it does moderate the perceived variance in the data.

Stimulated by this concern, we have added a discussion of the sensitivity of our results to the aerodynamic polar (two comments below). The greatest variation in Withers’ polar data is at the lowest angle of attack, and our results are insensitive to that portion of the curve.

Figure 1C: The leading edge of the point clouds and the corresponding mesh have a significant amount of rough edges and features. Are you confident that these represent physical features observed on the bird or are they the result of errors in the 3D reconstruction? This is important because leading edge protrusions

are well known to affect the flow, for example by tripping the flow from laminar to turbulent. This could affect the results of your CFD analysis.

There may be possible imperfections in the mesh at the edges. Error at the edges could affect the specific details of the flow around the wing. Despite these potential errors in flow detail, the lift produced essentially supports body weight, the polar is similar to those observed in other birds (clarified above), and the wake is similar to that observed from flow visualization.

We'll further address the sensitivity of our results with the comment below:

Line 402: The authors justify their turbulent model by claiming that it “is appropriate for bird flight simulations that require a high accuracy boundary layer”; however, their first inflation layer near the wall has a y^+ value of 10 which does not resolve the viscous sublayer. Y^+ values of approximately 1 are recommended for best results using low Re (k-omega) models. The authors report a difference of 3% in lift between $y^+ 10$ and y^+3 results, but there would presumably be an even larger error between $y^+ 10$ and y^+1 . And how does this affect the drag error?

We have run four simulations at $y^+ \sim 10, 7, 3,$ and 2 at the observed gliding orientation and speed (angle of attack: 5° ; speed: 7.8 m/s). Linearly extrapolating out to $y^+ \sim 1$, suggests that our approximation underestimates lift by 6.5% (which also adjusts the weight support estimate for this geometry to a fairly reassuring 100.7%), and overestimates drag by 5.5% . Drag plays a relatively minor role in the gusted impulse applied to the bird, providing at most 17% of the impulse (and as little as 5%). But note that drag reduction will cause the rigid-model to be perturbed slightly less, while a lift increase will cause the bird to be perturbed more. Finally, we have also run simulations at 20° angle of attack at $y^+ \sim 10$ and 3 to determine if the error was greater at high angles of attack. Linearly extrapolating the increase in lift with increasing y^+ at $y^+ \sim 1$, we estimate the lift error at 20° angle of attack to be only 3.2% , approximately half of the estimation at 5° .

We now report on the sensitivity of our results:

Line 270: The conclusion that inertial rejection is both a rapid and strong gust rejection mechanism in birds is robust to systematic error in the CFD-derived aerodynamic polar. At its peak, inertial rejection exceeded aerodynamic rejection regardless of whether we applied a 10% increase in drag across the drag curve, a 20% increase in lift to the linear portion of the lift curve, or a moderate 6.5% increase in lift across the entire lift curve (detailed in supplementary text). None of these changes affected the absolute magnitude of inertial rejection, but they affected the quasi-steady glide model, which in turn determined the relative magnitudes of inertial and aerodynamic rejection. The relative impact of inertial rejection and its insensitivity to our aerodynamic model provides confidence in the substantial role played by inertial effects in avian gust rejection. Further, we have conservatively estimated the effect of inertial rejection by ignoring the added fluid mass around the wings; accounting for the added mass would have further enhanced inertial rejection.

And within the SI:

Sensitivity of gust rejection to potential error in the aerodynamic polar

Our results are insensitive to systematic error in the CFD-derived aerodynamic polar. For this analysis we only examined the average relative contribution of inertial and aerodynamic rejection

at the instant of peak inertial rejection. A 10% increase in drag decreased inertial rejection from 32% to 31% and increased aerodynamic rejection from 6% to 8%. Similarly, increasing the lift curve slope by even 20% over the linear portion ($<10^\circ$) decreased inertial rejection by just 2% (to 30%) and increased aerodynamic rejection by 6% (to 12% of the total rejection). Increasing lift across the whole aerodynamic polar demonstrates that it is more sensitive to results at higher angle of attack, but even applying a moderate increase still did not affect our conclusions. We applied a 6.5% increase in lift coefficient—the amount required to achieve 100% weight support in the CFD simulations at 5 degrees angle of attack (the observed orientation and at the observed speed), but treated as systematic error across the whole aerodynamic polar—and this decreased inertial rejection from 32% to 27% and increased aerodynamic rejection from 6% to 21%. Even with this systematic increase in lift applied across the polar, inertial rejection was still 30% more effective than aerodynamic rejection at its peak.

3. The details on how the CFD code was implemented are very fuzzy, especially in the main portion of the text. Even in the methods section, some key aspects of the modeling remain unclear.

&

What was the convergence criteria for the numerical solution? What mesh shape is used (C-grid, o-grid, etc.)?

We have expanded the methods section to contain additional information about boundary conditions, mesh type, Reynolds number, validating information, and introductory material. Given that this is one component of a larger experiment, we maintain the detail should be reported within the methods section:

Line 481: CFD modelling was used to generate aerodynamic force data for the quasi-steady glide model. Commercial software (Mimics, Materialise NV, Leuven, Belgium; and SpaceClaim, version 19.1, ANSYS Inc., Canonsburg, USA) was used to create a high-fidelity surface geometry from a point cloud of the owl in a typically neutral glide posture. This point cloud measurement was performed prior to the experiment. Comparisons of this single geometry to the configurations adopted by the bird are provided for: all trials, pre-gust (Movie S4); and control trials, evenly sampled through the measurement region (Movie S5). Together these demonstrate that the geometry is representative of the posture prior to the gust, and throughout the glide when no gust was present.

The fluid mesh was generated in ANSYS Mesh as a hybrid mesh (Ansys Inc., Canonsburg, USA). The simulation domain (domain size: $9000 \times 6000 \times 6000 \text{ mm}^3$) was discretised by approximately 15 million non-uniform volume elements. Two bodies of influence controlled mesh size: mesh size was 5 mm for the inner body of influence and 12 mm for the outer body of influence. Within the inner body, adjacent to the bird surface, 15 o-shaped inflation layers were used to resolve the boundary layer: the first layer thickness was 0.5 mm ($y^+ \sim 10$) and subsequent layers had a growth ratio of 1.2. Except for the inflation layer, we used an unstructured tetrahedron mesh to reduce the number of mesh elements and computing cost. Several images of the tessellated fluid mesh across a plane through the arm-wing are shown in figure S6. We found that if we further refined the near-surface mesh by decreasing the first layer thickness to 0.1 mm ($y^+ \sim 3$), there was only a 3% difference in lift.

The boundary conditions for the simulation constrained velocity, pressure, and turbulence intensity. The velocity at the inlet was constrained to 7.88 m/s, and at the outlet followed the Neumann boundary condition, while pressure at both the inlet and outlet followed the Neumann boundary condition. The far-field boundary conditions, perpendicular to the flow, were constrained to be symmetric. Turbulence intensity was specified as 1% at the inlet.

The CFD simulations treated air as an incompressible and viscous flow. At the speed of the flights, Mach number was approximately 0.02, which suggests air compressibility was negligible. Turbulent effects though are not negligible. For viscous flow at Reynolds numbers for the gliding flights ($\sim 8.8 \times 10^4$ as defined by mean chord length and flight speed), turbulent effects are significant. To fully simulate the turbulence would require direct numerical simulation (DNS) and a fluid mesh density to capture the Kolmogorov scale. As DNS is computationally impractical for bird flight, we approximated the behaviour of the fluid with the $k-\omega$ SST turbulence model in ANSYS Fluent (version 19.1; ANSYS Inc., Canonsburg, USA). The SST model uses a blending function which combines the merits of the standard $k-\omega$ model, for near-surface simulation, and the standard $k-\epsilon$ model, for the domain away from the surface, which is appropriate for bird flight simulations that require a high accuracy boundary layer. We compared the effects of different turbulence models (Spalart-Allmaras, Reynolds Stress, $k-\epsilon$ standard, and $k-\omega$ standard) on the computed values of lift and drag, and found all models were within 5% of each other for a 5° angle of attack simulation. The criteria for simulation convergence was determined by the unitless ‘scaled’ residuals of three quantities, when: the continuity residual reached $< 5 \times 10^{-3}$; the orthogonal velocity-component residuals reached $< 5 \times 10^{-7}$; and the turbulence kinetic energy reached $< 3 \times 10^{-4}$. Turbulence kinetic energy was typically the last criterion met. The ‘scaled’ residuals are determined by normalising the absolute residuals by the maximum residual value among the first five iterations of the simulation.

The output of the CFD simulations was sufficient to provide weight support, produced a wake similar to published wakes for *T. alba*, and produced a polar similar to those published for other birds. Lift accounted for 94.2% of body weight after accounting for acceleration. The wake, as visualised by q-criterion, was similar to wakes measured behind the same individual, and the spanwise downwash distribution was also qualitatively and quantitatively similar to that measured using particle tracking (31, 32). Finally, the simulated lift-drag polars were consistent with other published data (33). Importantly, they exhibit similar values for coefficient of lift (~ 0.8) at high ($> 20^\circ$) angle of attack, for which much of the glide model operates; but note our drag values are larger due to the inclusion of the body (Fig S5). The lift and drag polars were computed from transient simulations for ten angles of attack between -5 and 50 degrees. The simulation timestep was 0.2 ms and lift and drag values were determined from the final, steady timestep.

What wing/body geometry is used? Is it from the collected in flight data and if so, at what stage in flight before the gust? How are the aerodynamic gusts applied? Does it consider just the 2D variation in gust intensity like the data shown in Figure 1b, or the 3D variation in intensity like the data shown in Figure

S2? Does the CFD model assume that the bird flies directly over the center of the gust? (From the results presented in Figure 2, the wing deflections are highly asymmetric which implies that it is not directly flying over the point of symmetry of the gust (or that the gust itself is asymmetric)).

We have emphasized in the text that the CFD was used to compute the aerodynamic polar, and added additional information about the geometry. We used spanwise-mean flow in our glide model, as we believe that is representative of what is experienced by the bird due to the bird's relatively large sampling volume. Small spatio-temporal variation in the flow will affect the aerodynamics at small spatial and time scales, but the broader dynamics of the bird are better explained by the mean flow.

Line 481: CFD modelling was used to generate aerodynamic force data for the quasi-steady glide model. Commercial software (Mimics, Materialise NV, Leuven, Belgium; and SpaceClaim, version 19.1, ANSYS Inc., Canonsburg, USA) was used to create a high-fidelity surface geometry from a point cloud of the owl in a typically neutral glide posture. This point cloud measurement was performed prior to the experiment. Comparisons of this single geometry to the configurations adopted by the bird are provided for: all trials, pre-gust (Movie S4); and control trials, evenly sampled through the measurement region (Movie S5). Together these demonstrate that the geometry is representative of the posture prior to the gust, and throughout the glide when no gust was present.

It is not safe to attribute asymmetry in an organism purely to asymmetry in force. Note as shown in the SI, even without a gust, the bird is not symmetric.

Line 73: More detail is needed about the CFD model earlier on in the text. Is it a static or dynamic model? Is it rigid or flexible? What turbulence model was implemented? Some of these details are later discussed in the methods section, but the reader needs a better understanding of the CFD was performed before introducing your results so they can assess them properly.

We have expanded upon the CFD model information within the methods (two comments above). Importantly, we have also better distinguished the role of CFD simulation and the glide model. The two different approaches were confusing as noted by reviewer 3. The precise role of CFD was to determine the essential aerodynamic polar used by the glide model, and the centre of pressure in a steady glide. We have expanded upon the roles of CFD, glide modelling, the CT scan, and kinematic analysis in the introduction to better prepare the reader, but we have left the physical reasoning for the results.

Line 66: Here, we investigate how birds cope with rapidly changing airflow by studying their flight through a controlled gust with a combination of high-speed video-based 3-dimensional (3D) surface reconstruction, computational fluid dynamics (CFD), and quasi-steady glide modelling to understand how birds reject gusts through wing morphing, *i.e.*, changing the shape and posture of their wings. Our experiment consisted of having a barn owl (*Tyto alba*) (Fig 1A) glide through a range of fan-generated vertical gusts to impose a transient perturbation of different magnitudes (Fig 1B) on the bird, with the strongest gust having a similar magnitude to the flight speed of the bird (7.72 ± 0.25 m/s, mean \pm s.d., n=13 flights). We measured rapid wing morphing and the perturbation impulse experienced by the torso and head using 3D reconstruction (Fig 1C) from ten high speed video cameras. From these measurements, we interpreted the effects of wing inertia on the glide path of the head and torso using a mass-calibrated CT scan to compute the movement of the wings relative to the centre of mass of the whole bird. We computed the aerodynamic effects of wing movement by contrasting the glide

results of the live articulated bird to a model of the expected glide path for a rigid bird, with an aerodynamic polar computed from CFD (Fig 1D).

4. Although the title claims that this reflex mechanism acts as a suspension system, this is neither quantitatively or qualitatively defined in the text. The only justification for why this acts as a suspension system is because “the bird consistently maintained a stable trajectory of its head and torso”. The only way to quantitatively claim this is to use the results to mathematically model the owl as a suspension system which is done using a mass-spring-damper analysis. If you wish to keep the suspension analogy, I recommend doing this.

Line 90: This is the only time that the suspension system is discussed somewhat qualitatively and in my opinion, this does not adequately demonstrate that the reflex mechanism acts as one. To truly prove that this mechanism acts a suspension system, you should use your experimental results to model it as a mass-spring-damper system which mathematically describes a suspension system.

Suspension refers to mechanical isolation (pragmatically closer to mechanical insulation); specifically, systems in which a main mass is isolated from other elements through a linkage that allows for relative motion. These other elements take the brunt of bumps and jolts from the surface (or medium) with which they are in contact, thereby delaying and/or softening the transfer of load to the main mass. The spring-damper system is indeed a common example of a suspension system. In our case, the wings provide isolation for the torso from aerodynamic perturbations, as shown both qualitatively and quantitatively.

The suspension system in birds is linked through muscle, a tissue that acts as a spring and damper in parallel (and in series). So, while we demonstrate mechanical isolation, our system also meets the expectations for a classical suspension system as well.

General Comments:

Line 39: “Implementing a similar reflex mechanism in future aircraft...”. Phrasing is important here, and ‘future aircraft’ to the public implies transonic aircraft. The reflex shown here undergoes large scale displacement which would almost certainly be unsuitable for transonic aircraft. It’s important to clarify that the target application is (presumably) small scale aircraft or UAVs.

Change made.

Line 55: “The gusts encountered by birds can be of a similar magnitude to their flight speed (4)”. There is no qualitative data (or reference) of gust speeds and correspond flight speeds in Tennekes’ paper. To support this claim, the authors can either conduct a literature review of papers reporting the flight speed and gust speed of a variety of species, or find a paper which does. We have changed the citation to include a pair of papers focused on 1) wind gust speeds and 2) bird flight speeds.

Hewston R, Dorling SR. 2011 An analysis of observed daily maximum wind gusts in the UK. *J. Wind Eng. Ind. Aerod.* **99**, 845-856. (DOI: 10.1016/j.jweia.2011.06.004)

Alerstam T, Rosén M, Bäckman J, Ericson PGP, Hellgren O. 2007 Flight speeds among bird species: allometric and phylogenetic effects. *PLoS Biol.* **5**, e197. (DOI: 10.1371/journal.pbio.0050197)

Fig. 1B: The measurements at the peak of the gusts appear to have no error bars. Is this because they are small or missing? Similarly, the error bars on the edge of the gusts are quite large (up to 0.3-0.4 m/s (or upwards of 10-15%) at a distance of 0.4m into the gust) and even overlap with the error bars of the data points for the other flow velocities, yet no discussion of error in the analysis is really presented. The authors report an n of 6 (presumably for the error calculations) but in the methods section they state that anemometer measurements were taken at 50 hz for 30 s which totals 1.5k measurements. This is confusing. Shouldn't the latter number be used in error calculations? Lastly, the methods section states that the velocity measurements were collected in a 5x6 grid, so why is only one cross sectional profile shown? The full velocity could be plotted as a 3D contour map and would be more informative. We present the span-averaged flow, which is produced from 6 time-averaged measurements. We would argue that the error of interest is not in the flow around a small volume, but the flow experienced by the bird. At an instant in time, the anemometer samples a trivial volume of flow relative to the bird. The spatial average of the flow is closer to that experienced by the bird and therefore more meaningful.

Line 92: “..A substantial reduction in wing pitch angle” is not the only mechanism that may “modify the lift generated by the wings”.

Agreed. Wing pitch is not the only aerodynamic rejection mechanism. We had an imprecise statement that we have changed.

Line 111: The bird achieved this, in part, by two dramatic movements: an almost instantaneous change in wing elevation (Fig 2A,C, Movie S2), followed by a substantial reduction in wing pitch angle (Fig 2D).

Line 92 (continued) Assuming the pitching axis does not coincide with the aerodynamic center, changing the angle of attack will also change the pitching moment of the owl. This should not be ignored. Additionally, was any tail motion observed during the in-flight tests? If so, this motion might be a mechanism to counteract a change in pitching moment from the gust.

No doubt there are interesting aeromechanics to be explored with pitching. It is clearly an important topic, and one we continue to pursue, but it is beyond the scope of this paper and not required to support our conclusion of inertial gust rejection.

Figure 2: It would be useful to superimpose the gust profile onto these plots so the reader can compare the velocity, elevation, and pitch response relative to the spatial strength of the gust.

The plots are cropped to contain the kinematics during the gust, and do not cover any region beyond this. We have added to the caption to better orient the reader to this:

Line 130: (B-D) Presented kinematic data encompasses the space occupied by the ‘1-cosine’ gust, and does not extend beyond it. (C,D) Kinematics represent the average of the two wings.

We chose not to add the gust velocity in the background. The plots are already quite busy visually and adding to them seems unnecessary when they only contain the ‘1-cosine’ profile of the gust. We are open to it and will leave it as a decision for the editor.

Figure 2: Furthermore, there is a fair amount of variability between test runs, so it would be useful to, for example, plot shaded error bands on the plotted averages.

We avoid error bands when we cannot justify arguing for a normal distribution. We argue that it is more informative to see the raw data to assess variation between trials in this case.

Paragraph @106: The classification of how the inertial vs aerodynamic gust response was determined is not sufficiently detailed. I presume mechanism 1) @ line 109 corresponds to the former, and mechanism 2) corresponds to the latter? Furthermore, do the mechanisms presented @ 109 “Calculating the difference between 1)...2)” mean the differences in the vertical velocity? As written, this statement implies the metric is taken by calculating the difference between the centers of mass, not velocity.

Thank you. In an effort to speak broadly about centre of mass dynamics, we managed to instead speak imprecisely. We have edited the text for better precision and clarity.

Line 135: We calculated the magnitude of gust rejection from the perspective of minimising the perturbation applied to the torso. *Inertial gust rejection* describes the difference between the observed movement of the torso and that of the whole-bird centre of mass, as calculated using the mass-calibrated CT scan (Fig 3A,B); inertial gust rejection is due to forces internal to the bird, *i.e.*, passive or active elevation of its wing masses with respect to the torso. *Aerodynamic gust rejection* describes the difference between the observed movement of the centre of mass and that expected from the quasi-steady, rigid-bird glide model with CFD-derived polar (Fig 3A,C); aerodynamic gust rejection is due changes in the external aerodynamic force. Both gust rejection metrics compare theoretical expectations for a rigid bird to observations of our live articulated bird. We quantified these gust rejection magnitudes by their impulse (Supplementary text). The internal impulse acting on the torso would equal that of the observed centre of mass of the whole bird if the bird’s wings were not mobile about the shoulder. The external impulse acting on the bird would equal that acting on the glide model if the bird did not reduce its aerodynamic force production through complex morphing.

Line 113: The conservation of moment calculations would be useful to perform here to physically show the reader how this is true. However, I believe you also need to include the momentum of the fluid especially since you are assessing a dynamic gust.

The fluid mass is an interesting point that we had neglected here. Using the common assumption that the added mass of air is that of a cylinder about the wing chord (Fung 1993, *An introduction to the theory of aeroelasticity*), we estimate an added mass of 10g. We ran a calculation where we added 10 grams of mass to the wings, the results did not change the conclusion of the study: the inertial effect was enhanced and average gust rejection at peak was 42%. We have added:

Line 279: we have conservatively estimated the effect of inertial rejection by ignoring the added fluid mass around the wings; accounting for the added mass would have further enhanced inertial rejection.

Line 115: The use of the word ‘impulse’ throughout the text is misleading. It appears that this calculation is simply the difference between the CFD and the experimental results. Impulse is defined as the force acting over time, but it does not appear that the authors performed time integration to actually calculate

the impulse.

It is indeed impulse. We present mass-normalized impulse on plots, which is equivalent to a change in velocity. This method of normalization is convenient as it is far more intuitive than scaling it to body weight. We have edited the manuscript throughout to be consistent and clear on this point, as well as add additional SI to further explain the approach.

Line 117/119: “At this stage...” “Later, as the wing pitched down...”. It would be useful to mark these stages on the plots to direct the reader exactly to where these trends are occurring.

We have edited the text to be more precise.

Line 157: At the instant of peak inertial rejection,...

Line 159: At this same instance,...

Line 120-122: “As the wing pitched downwards, there was an increase in the difference between the measured vertical velocity of the whole bird center of mass and the rigid body simulation, indicating large aerodynamic-based gust rejection of $48 \pm 4\%$ of the potential impulse (Fig 3c).” The authors cannot confirm that this is due to the translation and rotation of the wings. The wing and feather aeroelasticity plays a substantial role in mitigating gusts, which is not accounted for here since the calculations are performed by taking the difference between the rigid CFD model and the flexible in-flight experiments. Agreed. The quote above merely refers to the timing of aerodynamic rejection and the timing of wing pitch: the two events happen in relative synchrony.

Line 188: The authors state that “birds may then be able to tune the dynamics involved with gust rejection by modifying the stiffness of their shoulder joint”. If the wing does indeed act as a suspension system as the authors claim, then changing the stiffness of the shoulder would effect the timing of the inertial rejection (dictated by mass-spring-damper equations of motion). Stiffening the shoulder muscle would cause the inertial rejection to reach it’s maximum sooner. This is contradictory the statement made in line 200, claiming that the inertial response must be passive since it is not faster than the reaction time measured in other studies. That is, unless the shoulder stiffness is assumed to already be at a maximum for these in-flight tests.

Birds can modify the stiffness of their shoulder through agonist-antagonist interactions. Tuning the dynamics would be a ‘feed-forward’ approach. This may be feasible as turbulence is more predictable in certain environments.

Figure 3: The results in Figure 3 are only presented for one gust intensity (medium), right? Do these results change at the lower and higher intensities? Since this was only assessed at one gust intensity, I don’t think you can conclude that “passive mechanical properties of the shoulder joint and wing are responsible for immediate gust rejection” because this time frame was “equal to the minimum behavioural reaction time measured in other bird species”. The data shown in Figure 2c indicates that for the high gust intensities, the wing elevation changes more rapidly (the response onset is sooner and the slope is steeper) but you have not yet proven that this difference between gust intensities is only due to the aerodynamic rejection.

All of the data are presented together. We have added a statement to make this clear:

Line 154: This inertial effect reduced the impulse to the torso (Fig 3B). The dose-dependent kinematics resulted in wing inertia rejecting a greater magnitude of the gust impulse with increasing gust intensity, but the proportion of the total expected impulse rejected by wing inertia did not discernably differ across gust intensities (Fig S1).

The new supplement figure is presented below.

Line 292: The spacing of the velocity measurements is .45m which appears to be approximately the same as the span of the owl. Some comment would be useful on the spatial sampling of the velocity profiles. Figure S2 shows only 3 velocity measurements overlap with where the bird should fly (when the wings are fully extended). Presumably when the wings are elevated during a gust, no data points capture the velocity over the wings.

The plane presented in figure 1B is located above the bird in all flights, within 20 cm of all but one flight. The wings rotate upwards into the plane. At the moment of peak inertial rejection, the wings will have rotated upwards 15 degrees toward the plane. It is an appropriate and very relevant plane.

We do not attempt to explain all of the aerodynamic effects involved in gust rejection, and particularly avoid discussing the mechanics of aerodynamic gust rejection beyond the first 80 ms. We demonstrate a correlation between wing pitch and aerodynamic rejection, which is likely needed to produce such a sharp change in aerodynamic rejection. Second-order aspects such as feather bending could be attributed to the equivalent of a few degrees change in angle of attack. By contrast, the wing pitches ~ 20 degrees. Our sampling of 2-3 points across the span during the first 80ms seems reasonable for the conclusions drawn.

Line 332: The surface accuracy of the reconstruction is said to be 1.67mm, however there are many features of the bird that will be well below this thickness (namely, the trailing edge feather thickness). How does that effect your reconstruction?

Our surface accuracy was -0.68 ± 1.67 mm (mean \pm s.d.; where 50% of error $<$ 1.1 mm & 95% $<$ 3.5 mm) for a fibreglass bird model. This does not necessarily explicitly limit the thickness that we can resolve. We measured the upper and lower surface of the bird independently, the two must then be combined. Combining without smoothing could create places where the two surfaces penetrate or move apart, so we smooth the geometry adjusting parameters to prevent any shapes that are physically implausible.

Line 341: Are the mass properties segmented into each wing section (i.e. handwing, armwing etc.) or is the mass tracking done by translation and rotation of the whole wing mass? This could yield very different results if the whole wing is treated as one lump mass that can only undergo translation/rotation. Upon looking at the supplementary material, it appears that the mass is recorded for each voxel, so is each voxel mapped to the deformed bird geometry?

We have added some additional information to the methods to clarify this. We used rigid-body transforms to track the centre of mass. These of course are only valid if the wing does not deform substantially. We have added an SI video that displays the wings from each glide pre-gust and at the moment of peak inertial rejection. The outlines all generally look similar despite the imposed gust on half of them. We concluded therefore that wing deformation effects on the centre of mass are negligible. Beyond the moment of peak inertial rejection, this statement begins to lose validity, but that has little bearing on our conclusions.

Line 454: The segmented torso and left wing from the scan were aligned with the point clouds to estimate the mass distribution of the bird in flight. We assumed that the inertial dynamics of the left wing were equal to those of the right wing. The CT and point cloud alignment was performed for a single typical glide posture to establish the mass and centre of mass position for each point cloud segment. Centre of mass position was then tracked subsequently when aligning point clouds using rigid-body transforms that allowed for rotation and translation. This approach was ideal for the duration that wing flexion was negligible (*e.g.*, the time-course dominated by the inertial rejection mechanics. Movie S3). However, the effect of wing shape change on the centre of mass was not completely ignored because we allowed the wing to translate. For example, when the elbow flexed, the rigid-body transform that best fit the flexed state of the wing moved the centre of mass proximally as expected.

Line 370: “The model was free to accelerate both horizontally and vertically, but could not rotate, as the real bird displayed no tendency to pitch during the gust.” This is a major assumption. The lack of observation of the bird pitching during a gust does not mean that, in the absence of wing motions, the bird would not pitch. Elaborating on my comment regarding line 92, the wing’s reduction in angle of attack is likely an active mechanism to, in part, minimize the pitching moments caused by the gust. When the gust hits the nose end of the bird first, this would cause a nose-up pitching moment. This has been shown in similar experiments done with gliders, so it is not accurate to assume that there would be no pitch in your rigid model.

There would be pitch in a passive rigid model; we do not suggest otherwise. Our investigation is in the linear motions. We mention that the bird did not pitch, because if the bird did pitch and the model did not, then we would be neglecting those rotational effects on the aerodynamics. Your comment about wing pitch and pitching moment are interesting and will be considered in detail in future work.

403: From my perspective, comparing the results of different turbulent models is not sufficient for validating the results of the modelling.

We have added details that discuss the CFD simulation producing sufficient lift for approximate body weight support, a qualitative and quantitative match between CFD and measured wakes, and a similarity to the aerodynamic polar of other birds.

Line 529: The output of the CFD simulations was sufficient to provide weight support, produced a wake similar to published wakes for *T. alba*, and produced a polar similar to those published for other birds. Lift accounted for 94.2% of body weight after accounting for acceleration. The wake, as visualised by q-criterion, was similar to wakes measured behind the same individual, and the spanwise downwash distribution was also qualitatively and quantitatively similar to that measured using particle tracking (31, 32). Finally, the simulated lift-drag polars were consistent with other published data (33). Importantly, they exhibit similar values for coefficient of lift (~0.8) at high (>20°) angle of attack, for which much of the glide model operates; but note our drag values are larger due to the inclusion of the body (Fig S5).

Were any corrections made to account for the bird flying too high/low left/right? Since the gust intensity is dependent on distance from the gust source, differences in flight height above the gust could change your measurements. Some mention of this in the results would be sufficient.

No. The change in peak gust intensity was relatively small. We have clarified this in the methods:

Line 384: All flights passed between these planes, with all but one within 0.2 m of the upper plane. Fitted peak intensity was similar between the two planes: peak speed of the lower plane was greater by 6%, 4% and 3% for low, medium, and high intensity, respectively. Due to the similarities in intensity and the proximity of flights to the upper plane, we considered the upper plane as representative of the perturbation experienced by the bird.

Referee: 3

Comments to the Author(s)

A thorough study with an innovative experimental design. I am convinced of the general conclusions (but see my comments about the aerodynamic consequences of wing elevation + retraction). The figures are clean, and the methods are detailed. Overall, my suggestions are to emphasize the importance of the work to help draw in a general audience, more explicitly state the rationale behind the experimental design (to give yourselves due credit), and explain the interpretation of the results in more detail to help hold the attention/understanding of the general audience. Given the page limitations of Proceedings B and the complexity of the methods of the study, it seems that you are stuck between a rock and a hard place. With that in mind, I have tried to avoid asking for additional details unless it seems vitally important for the work to be readable by a general audience. Specific examples below.

General comments:

- The text switches between referring to the head and torso and just the torso. I suspect that you are referencing the head + torso throughout (i.e. the whole bird minus the wings). Maybe use something like “torso + head” or “torso/head” throughout, just to avoid any confusion.

We have added:

Line 118: (for the remaining results, references to the torso also apply to the head, as they were tracked as a whole).

- As a suggestion, I think that a more detailed introduction would help to draw the interest of non-experts and help them to appreciate the importance of this study. Specifically, I think that the introduction could be improved by discussing the importance of rejecting perturbations in flight more concretely. You mention in the abstract that reduced motion of the torso simplifies certain tasks, but the actual costs of failing to reject perturbations remain somewhat opaque. Not rejecting gusts has implications for cost of transport and risk of injury - both for animals and engineered vehicles. I think that putting your work in this context early will help emphasize the importance for the general audience. But do what you can with the space permitted.

The introduction has seen a large number of edits. Relevant to this suggestion, we have expanded upon the ecological implications that gusts have on energetics and nesting habitat in birds. It is a small, but important addition to include some of the recent work by Sheppard and colleagues. Within movement ecology, ‘mean wind speed’ still dominates the literature due to the challenges of measuring variation in wind speed experienced in flight.

Line 45: Birds routinely fly in gusty wind flows, and often in close proximity to obstacles such as terrain and buildings, conditions which challenge engineered air vehicles of a similar size (1, 2). The gusts encountered by birds can be of a similar magnitude to their flight speed (3, 4) and are largely unpredictable. As such deviations in flight path and/or orientation caused by gusts have the potential to disrupt critical behaviours such as landing, prey capture and visual tracking. Therefore, the capacity of birds to cope with gusts could limit their foraging success (5), the conditions in which they can forage, their nesting sites, as well as increase their flight costs (6). The robust flight of birds observed in gusty unpredictable winds leads to the hypothesis that most flying birds must possess fast, stabilising responses that reject gust effects.

- Re: “The time-varying magnitude of the two gust rejection mechanisms was quantified by calculating the difference between 1) the torso and the whole bird centre of mass computed using the mass calibrated CT scan (Fig 3a,b); and 2) the whole bird centre of mass and the rigid body CFD-based simulation (Fig 3c).”

I don't think that it would be immediately evident to a general reader that any difference between the torso and the whole bird center of mass indicates an inertial response and that any difference between the whole bird center of mass and the center of mass of the CFD model indicates an aerodynamic response. I don't disagree. But I do think that it would be very helpful to walk through that logic in a bit more detail either here or in the introduction, rather than stating these concepts as evident.

We have expanded the text to make the work more accessible and provided additional information in the SI.

Line 133 We calculated the magnitude of gust rejection from the perspective of minimising the perturbation applied to the torso. *Inertial gust rejection* describes the difference between the observed movement of the torso and that of the whole-bird centre of mass, as calculated using the mass-calibrated CT scan (Fig 3A,B); inertial gust rejection is due to forces internal to the bird, *i.e.*, passive or active elevation of its wing masses with respect to the torso. *Aerodynamic gust rejection* describes the difference between the observed movement of the centre of mass and that expected from the quasi-steady, rigid-bird glide model with CFD-derived polar (Fig 3A,C); aerodynamic gust rejection is due changes in the external aerodynamic force. Both gust rejection metrics compare theoretical expectations for a rigid bird to observations of our live articulated bird. We quantified these gust rejection magnitudes by their impulse (Supplementary text). The internal impulse acting on the torso would equal that of the observed centre of mass of the whole bird if the bird's wings were not mobile about the shoulder. The external impulse acting on the bird would equal that acting on the glide model if the bird did not reduce its aerodynamic force production through complex morphing.

In other words, I think that the experimental design is an innovative way to separate between inertial and aerodynamic responses, but I don't think that you give yourselves due credit for coming up with a powerful experimental design!

Thank you very much

- Why is the velocity of the whole bird, wing, and torso + head the important factor in measuring gust rejection, rather than their position? Is it because the gust is too brief to cause major changes to position? I think it would be helpful if you could briefly mention why the focus is placed on vertical velocity rather than position (or acceleration, for that matter).

We have replaced a few instances where we describe a change in velocity and modified it to describe impulse. The important factor is the acceleration integrated over time. We initially chose to look at force/acceleration, but realized that instantaneous force can be easily reduced, while at the same time responding poorly to the gust. One simple example of where instantaneous acceleration is misleading would be: if the bird experienced a square-wave gust and instantaneously accelerated until its vertical speed equaled that of the gust, all other instances in time the bird would experience zero acceleration in response to the gust. Over most of the gust and time, this solution is ideal, but not one that is useful.

Whereas, if we integrate the acceleration over time, computing the mass-normalized impulse or change in velocity, it becomes clear that that approach is a poor one.

Position would be a reasonable quantity but the flight paths are unconstrained and, as such, change in position due to the impulse is influenced by their initial velocity. Nor do we use absolute velocity as a measure, for similar reasons.

While we haven't added specific commentary within the manuscript, we think correcting the instances that describe 'changes in velocity' to 'impulse' will streamline and clarify things without requiring extra text.

- I would suggest making the distinction between the CFD simulation and the CFD-based mathematical model clearer. Perhaps you could refer to the mathematical model without using the word "CFD". It took a bit of time before I understood the distinction, so I worry that others will have the same trouble.

We agree: the references to the model at the end of the introduction and at the beginning of the results were imprecise. We now also differentiate the two by referring to the CFD as a "simulation", and the rigid-body, quasi-steady glide calculations as a "model".

In addition, we made the following changes

Line 78: We computed the aerodynamic effects of wing movement by contrasting the glide results of the live articulated bird to a model of the expected glide path for a rigid bird, with an aerodynamic polar computed from CFD (Fig 1D).

Line 118: compared to that predicted by a quasi-steady glide model of a rigid bird

- Unless I am missing something about the mathematical model, I think that you are missing two important mechanisms of aerodynamic rejection, and so should not be attributing the response entirely to changes in wing pitch. In addition to the change in wing pitch, the elevation of the wings also means that the vertical component of the lift vector acting on each wing is reduced quite a bit (though the overall lift force is roughly the same). Also, the wings are retracted over the gust (at least based on the example Fig. 2A). The combination of wing elevation and wing retraction means that the effective wingspan is greatly reduced over the gust. For example, the effective wingspan of the red wings in Figure 2A is about 50% of that of the blue wings, which must have a huge effect on the vertical force they produce. Therefore, I think that elevating the wings has both an inertial and aerodynamic effect, which makes it harder to cleanly separate the two mechanisms of gust rejection.

Luckily, this does not alter your major conclusion about the importance of inertial rejection. However, I do think that you need to discuss these two other mechanisms of aerodynamic rejection rather than put all the emphasis on changes in wing pitch.

&

Line 92 – See my above comment about the aerodynamic impact of changes in wing elevation.

We certainly don't think wing pitch is the only important mechanism, and we accept that our writing was ambiguous/misleading; pitching is just a mechanism that will certainly explain a substantial portion of the aerodynamic rejection in the latter phases of the gust response, and the timing of pitching coincides with the rise in aerodynamic rejection.

We have rectified the statement that suggested wing pitch was sufficient to explain the aerodynamic rejection. It is likely a substantial effect (20 degree change in angle of attack), and it aligns well with the timing of aerodynamic rejection, but it is certainly not the only effect.

Line 111: The bird achieved this, in part, by two dramatic movements: an almost instantaneous change in wing elevation (Fig 2A,C, Movie S2), followed by a substantial reduction in wing pitch angle (Fig 2D).

We have also added details about the broad array of aerodynamic mechanisms available to birds.

Line 285: Why is aerodynamic rejection initially weaker than inertial rejection? Despite a multitude of possible aerodynamic rejection mechanisms, inertial rejection increases more rapidly than aerodynamic rejection. The most likely aerodynamic rejection mechanisms that could reduce the experienced vertical force/impulse include: 1) changes to wing posture, which can reject gusts by reducing angle of attack, and wing elevation, which tilts the resultant aerodynamic force vector inward; 2) changes to wing shape, where feather bending modifies wing camber and angle of attack, and wing retraction, which reduces wing area; and 3) the dynamics of wing elevation, which reduces the relative velocity between the wings and the gust, and effectively reduces the local angle of attack of the wing, most notably at the tip.

- The discussion is great, overall. One additional topic that you might consider including (space permitting) is the speed at which an inertial response leads to a change in position relative to the speed at which an increment of force leads to a change in position. Is there something about the lag time between a change in net force and the subsequent change in position (versus throwing one's weight around) that makes an inertial response especially effective? The answer may be no. Just something to think about.
&

- It seems strange that the rapid elevation of the wings does not have any effect on the whole bird center of mass relative to the CFD model. If the wings are moving upward with the air, that should mean that the angle of incidence of the wings is much lower in the actual bird than in the CFD-based model, which I would expect would cause the whole bird to fall a bit (relative to the CFD-based model). Evidently, that is not the case ("there was negligible aerodynamic rejection accounting for only $6 \pm 6\%$ of the total rejection"). Do you have any thoughts on why the aerodynamic rejection was so low even while the wings were rapidly elevated?

We have added a section in the discussion to discuss this topic and the one above:

Line 283:

The delay in aerodynamic rejection

Why is aerodynamic rejection initially weaker than inertial rejection? Despite a multitude of possible aerodynamic rejection mechanisms, inertial rejection increases more rapidly than aerodynamic rejection. The most likely aerodynamic rejection mechanisms that could reduce the experienced vertical force/impulse include: 1) changes to wing posture, which can reject gusts by reducing angle of attack, and wing elevation, which tilts the resultant aerodynamic force vector inward; 2) changes to wing shape, where feather bending modifies wing camber and angle of attack, and wing retraction, which reduces wing area; and 3) the dynamics of wing elevation,

which reduces the relative velocity between the wings and the gust, and effectively reduces the local angle of attack of the wing, most notably at the tip.

We hypothesise that the relatively slow aerodynamic response was due, in part, to the way it scales. Aerodynamic rejection rises slowly early on and then grows more strongly later in the gust. This delay implies that the dominant mechanisms do not scale with force on the wings, but rather with one or more time-integrals of force (a change in velocity or position/posture; as discussed above). On the other hand, inertial rejection scales directly with force for a given alignment between the centres of percussion and pressure (Fig 4A, Supplementary text) and, as such, begins instantly, regardless of gust magnitude.

Specific comments:

line 32 – I recommend changing "found that their wings act as a suspension system" to "found that their wings can act as a suspension system". I recommend adding "can" as the study involved only one individual, so the authors should be cautious in extending their findings to all birds.

&

Line 53 – I suggest starting a new paragraph at “Birds...”

&

Line 143 – Typo, add ‘the’ in “centre of _ plot”

Changes made.

Line 63 – Preference thing, but it is not immediately obvious if “our observations” refers to observations during this study or before this study. It is not all that important, but changing the wording slightly (perhaps by removing “Based on our observations”) might clear up any minor confusion.

We have edited the introduction to clarify this and expand upon the rationale of the approach:

Change made. Plus, we expanded upon the rationale of the approach:

Line 83: the trajectory of the barn owl was relatively unaffected by even the strongest gust. Instead the bird rejected the potentially negative consequences of gusts by immediate, likely passive, wing elevation about the shoulder. This fast response was then followed by more complex changes in wing shape, potentially under active musculoskeletal control. Based on these observations we propose two independent mechanisms for gust rejection in birds:

Line 79 – Please specify at what height from the gust generator these measurements were taken.

We have added information in the methods:

Line 382: A sonic anemometer (HS-50; Gill Instruments Ltd., Lymington, UK) was used to survey the gust velocity field across two planar horizontal grids of 30 points (5×6) spaced 0.45 m vertically apart, with the lower plane being 0.40 m above the nozzle exit. All flights passed between these planes, with all but one within 0.2 m of the upper plane. Fitted peak intensity was similar between the two planes: peak speed of the lower plane was greater by 6%, 4% and 3% for low, medium, and high intensity, respectively. Due to the similarities in intensity and the proximity of flights to the upper plane, we considered the upper plane as representative of the perturbation experienced by the bird.

Line 93 – Wing pitch also seems to be proportionate to gust strength, as stated in the legend for figure 2. I would suggest rewording slightly to make that clearer.

Corrected:

Line 113: Both movements are likely to be important for gust alleviation and change in proportion to gust intensity, with wing elevation delivering a rapid rejection of the gust due to inertial effects, and the delayed wing pitching acting to modify the lift generated by the wings.

Line 118 – Specify s.e.m or standard deviation

Added: s.e.m.

Figure 2 – The purple-green arrows on top of figure 2 are a bit confusing to me. I am not sure if they indicate a continuum between ‘inertia dominated’ and ‘aerodynamics dominated’ rejection, or if they are simply labeled for the arrows below. I feel like the latter is true. But you might consider the unintended interpretation.

Thank you for the feedback. They were added to assist the reader in appreciating the transition between the two rejection mechanisms. We have added their description and rationale in the figure 3 caption:

Line 205: (A) Top arrows indicate the transition from gust rejection dominated by inertial effects to that by aerodynamic effects.

Line 167 – I would appreciate a more detailed presentation of how the center of percussion is calculated for the owl wing. Perhaps you could include an excel sheet or script in which this parameter is calculated from the mass distribution of the wing.

We have added this information to the SI, and added reference to it:

Line 213: The centre of percussion was located 39.4% of the wing length from the shoulder (Supplementary Text).

SI:

Centre of percussion calculation

The centre of percussion P derives from the wing mass distribution. It is given by

$$P = \frac{I_h}{m_w l_G},$$

where m_w is the mass, I_h the moment of inertia, and l_G the centre of mass of the wing (the latter two being taken about the shoulder).

The wing CT-scan data comprise thousands of elemental voxels of identical volume δV , each with a unique position $\mathbf{p}(x, y, z)$ and density ρ . x points along the shoulder hinge axis, y along the wingspan, and z in the dorsoventral direction. We obtain the mass of each voxel as $\delta m = \rho \delta V$, from which the inertial properties are obtained from the following sums:

$$m_w = \sum \delta m,$$

$$I_h = \sum r^2 \delta m,$$

$$l_G = \frac{\sum r \delta m}{m_w}.$$

r is the distance from the shoulder hinge to each voxel once projected onto the plane of rotation. By substitution, we find that the centre of percussion is

$$P = \frac{\sum r^2 \delta m}{\sum r \delta m}.$$

The paragraph at 166 – Are there any differences in the center of percussion and center of pressure between the two wings? And is there any variation in the span-wise position of the center of pressure with the angle of attack in the CFD model?

We do not have the CT scan of the right wing aligned with the flight posture. Our data is based on the left wing only. However, regarding the CFD, we have added information about the sensitivity of the spanwise position of the centre of pressure to angle of attack:

Line 215: The position of the centre of pressure was stable with regards to changes in angle of attack, moving outboard by 0.3% of wing length per degree increase.

Line 170 – Please specify that these measurements are from the CFD model.

We have added that fact:

Line 214: The spanwise centre of pressure, as computed from CFD, was located 46.1% of the wing length from the shoulder.

Line 378 – How did the overall lift force calculated via CFD compare to the animal's weight?

Line 531: Lift accounted for 94.2% of body weight after accounting for acceleration.

Figure S2 – I have trouble interpreting the top row of the figure without referencing the methods. Perhaps including labels like "top view" and "side view", and a profile of the owl (in gray, like in the lower row) in the top row would help.

The edited figure is reproduced below:

[Final comment] Line 533 – Because you have space in the supplemental material, I would suggest that you proceed very slowly through this calculation. It is an important component of your conclusion, so it would be great if a general reader could follow the math without too much effort. For example, the addition of K in the same step as substituting and rearranging takes some work to follow. A drawing may help. Also, should it be $l'F$ or simply lF ?

Yes, there was an error. We have corrected that and have greatly expanded upon this section within the SI to aid a broad audience of readers. The remaining text introduces centre of percussion and then goes into depth about the application to wings.

SI:

Centre of percussion

Gust force on the wing is transmitted to the torso as reaction at the shoulder hinge. If the gust force acts through the centre of percussion, though, initial reaction on the torso will be zero. We present here a simple, intuitive explanation of how this works.

Imagine a pendulum-like bar (thick grey line) whose pivot O is free to translate sideways. If we push the bar transversely with force F at the centre of mass G , it will accelerate linearly, without rotation, by amount a_G . The dashed green line shows the position of the bar an instant later (figure: left).

If we push the bar lower down, however, we can cause anticlockwise rotation about G . This accelerates O leftwards relative to G by amount $a_{O/G}$, in opposition to a_G (which has the same sense and value as before because F is identical). Now, there exists a special ‘push point’ for which these initial accelerations cancel: the pivot remains still and the bar moves in pure rotation about O . The solid green line shows the bar in this condition at the same time instant as before (figure: right). Let us proceed to find this special ‘push point’.

The linear equation of motion for the bar is

$$F = ma_G.$$

The moment equation about G is

$$\sum M_G = F(P - l_G) = I_G \ddot{\theta},$$

where θ denotes the rotation angle of the bar. By the notation above, we may now write the linear acceleration of the pivot O as

$$a_O = a_G + a_{O/G}.$$

In other words, the motion of O is that at G plus that between O and G .

We require $a_O = 0$, or $a_{O/G} = -a_G = -F/m$. As mentioned, $a_{O/G}$ is the leftward (negative) acceleration of O relative to G by rotation, which is simply $-l_G \ddot{\theta}$ while angle θ is small. Thus $-l_G \ddot{\theta} = -a_G$, so (we cancel the minus signs)

$$\frac{Fl_G(P - l_G)}{I_G} = \frac{F}{m}.$$

By the parallel axis theorem, $I_O = I_G + ml_G^2$, after expanding brackets, we then find

$$mPl_G - ml_G^2 = I_O - ml_G^2,$$

which ultimately yields

$$P = \frac{I_O}{ml_G}.$$

Location P is the ‘push point’ we seek. It is known as the *centre of percussion*. A force at this point will not accelerate the pivot initially.

The idea also applies to a hinged wing. For a more detailed treatment of the problem, please refer to the subsequent derivation.

--

We now present a linearised model, comprising a point mass (torso) and two rigid beams (wings) moving in the plane, that demonstrates, for those interested in further detail, the same idea but for a bird-like system. The linearisation makes the dynamics more accessible, with the caveats that (i) change in wing elevation angle stays below 15-20 degrees, and (ii) the flow remains broadly attached to the lifting surface.

Term	Description	Positive sense
m_t	Torso mass	---
m_w	Wing mass	---
I_G, I_h	Moment of wing inertia, as given	---
l_G	Distance from hinge to centre of wing mass	---
$l_F, l_{F'}$	Distance from hinge to centre of pressure, as given	---
F, F'	Aerodynamic force on each wing	Up
R, R'	Vertical reaction force between each wing and torso	Up
T_h, T'_h	Hinge torque	Clockwise
a_t	Linear vertical acceleration of centre of torso mass	Up
a_w	Linear acceleration of centre of wing mass	Up
z	Vertical torso position	Up
θ	Angular wing elevation from horizontal	Up
P	Distance from hinge to centre of percussion	---

An exploded free body diagram of the model is shown below. Only one wing is drawn; the forces and geometry are the same on both sides. We omit horizontal forces because they are not relevant here.

Figure: Free body diagram of the model. Left: ‘point’ torso. Right: rigid wing.

The equation of motion for vertical force on the torso is

$$2R - m_t g = m_t a_t = m_t \ddot{z}.$$

We use z for vertical torso position. For each wing, the linearised equation is

$$F - R - m_w g = m_w a_w = m_w (\ddot{z} + l_G \ddot{\theta}).$$

It is linear because a_w is approximated as $\ddot{z} + l_G \ddot{\theta}$. The moment equation about the wing centre of mass G is

$$F(l_F - l_G) - T_h + R l_G = I_G \ddot{\theta},$$

We may substitute the force equation, along with $a_m = \ddot{z} + l_G \ddot{\theta}$ and the parallel axis theorem, $I_h = I_G + m_w l_G^2$, to obtain

$$F l_F - T_h - m_w l_G g = m_w l_G \ddot{z} + I_h \ddot{\theta}.$$

Now, before the gust strikes, all time-dependencies disappear and each term assumes its equilibrium (0) value. We thereby obtain the equations of static equilibrium,

$$2R_0 - m_t g = 0,$$

$$F_0 - R_0 - m_w g = 0,$$

and, for moments,

$$F_0 l_{F0} - T_{h0} - m_w l_G g = 0.$$

This is the steady level flight condition. If we subtract equilibrium forces, as is usual for linear systems, then

$$2(R - R_0) = m_t \ddot{z},$$

$$(F - F_0) - (R - R_0) = m_w \ddot{z} + m_w l_G \ddot{\theta},$$

and, finally,

$$(F l_F - F_0 l_{F0}) - (T_h - T_{h0}) = m_w l_G \ddot{z} + I_h \ddot{\theta}.$$

Notice that all bracketed terms now denote increments about equilibrium. We recast these with a prime (‘).

$$2R' = m_t \ddot{z}, \quad (1)$$

$$F' - R' = m_w \ddot{z} + m_w l_G \ddot{\theta} \quad (2)$$

$$F' l_{F'} - T'_h = m_w l_G \ddot{z} + I_h \ddot{\theta} \quad (3)$$

The key point here is that, with the linear model, we can subtract static loads and deal only with increments. By doing this, we effectively remove the influence of gravity from the model.

The fact that $F l_F = F_0 l_{F0} + F' l_{F'}$ is not immediately obvious. If f denotes the distributed force that makes up F , then, by the centroid formula,

$$F l_F = \int_0^b f y dy.$$

b is the wing semi-span. But $f = f_0 + f'$ (total = static + gust) in our linear model, so we deduce

$$F l_F = \int_0^b f_0 y dy + \int_0^b f' y dy = F_0 l_{F0} + F' l_{F'},$$

as required. Now, putting \ddot{z} from (1) into (2) and rearranging, we find

$$\ddot{\theta} = \frac{F'}{m_w l_G} - \frac{R'}{m_w l_G} - \frac{2R'}{m_t l_G},$$

which, when inserted into (3), gives

$$F' l_{F'} - T'_h = m_w l_G \left(\frac{2R'}{m_t} \right) + I_h \left(\frac{F'}{m_w l_G} - \frac{R'}{m_w l_G} - \frac{2R'}{m_t l_G} \right)$$

(We have also substituted \ddot{z} using (1) here.) Collecting force terms,

$$F' l_{F'} - T'_h = R' \left(\frac{2m_w l_G}{m_t} - \frac{I_h}{m_w l_G} - \frac{2I_h}{m_t l_G} \right) + F' \left(\frac{I_h}{m_w l_G} \right).$$

The red portion contains only inertial properties and is constant for any given flyer. We shall absorb it as new constant K . After rearrangement,

$$R' = \frac{F'}{K} \left(F' l_{F'} - \frac{I_h}{m_w l_G} \right) - \frac{T'_h}{K}.$$

Complete torso stabilisation (zero acceleration) requires that the reacted force increment R' be eliminated. Setting $R' = 0$ and solving for $l_{F'}$ yields

$$l_{F'} = \frac{I_h}{m_w l_G} + \frac{T'_h}{F'}.$$

In the classic case of the ‘free’ hinge under zero torque, the necessary condition is simply

$$l_{F'} = \frac{I_h}{m_w l_G} = P.$$

In other words, F' must act at a specific point located $l_{F'} = P$ from the wing root. This point is known as the *centre of percussion*. It is an inertial property and, as such, constant for a given wing mass distribution.

For the owl, $l_{F'} = P$ works only at first. Once T'_h and F' develop, stabilisation requires that

$$l_{F'} = P + \frac{T'}{F'}$$

i.e. that the gust force act beyond the centre of percussion by an amount equal to the force-normalised hinge torque.

Note: in mechanical systems, T' is usually produced by springs and dampers (passive) or prescribed by actuators (active). Here, mechanical elements are replaced by muscles, tendons, etc.

Appendix B

Comments to Author:

Associate Editor: Doug Altshuler

The manuscript has now been seen by the three original referees who bring considerable expertise in both biology and engineering. The three referees did not reach consensus. Two are strongly in favor of acceptance, one as is and one with some additional minor concerns. The other referee raises three significant concerns that were not addressed from the first review. I am agreement with the majority perspective that the work is sufficiently well performed, valuable and of broad interest for the readership of Proceedings B. However, it would be helpful to see direct responses to the remaining major concerns. This will be valuable for the editorial board as we consider the final decision. In the event of eventual acceptance, it will be important to see the authors' responses to these concerns in the online review history that will accompany the article.

Thank you for the feedback. We have addressed the three major comments made by referee two by 1) modifying a paragraph to include additional information about aeroelasticity; 2) verifying that we have citations for each of the components of work that we use to validate our CFD; and 3) amending our conclusions to specify that the hinged wings of birds specifically perform the same role as a suspension system, to differentiate this from the idea that the wings are a classical spring-damper system. We have also addressed the suggestions made by referee 3, which includes additional information within the SI about why this study examine impulse instead of other related metrics.

Reviewer(s)' Comments to Author:

Our responses reference the line numbers after “accepting all changes”

Referee: 2

Comments to the Author(s).

It's disheartening that the authors ignored the overwhelming majority of my initial critiques and presumably the critiques of the other reviewers as well. It's especially evident that the authors didn't even take time to correct some of the softball/easy fixes like changing wording or adding clarifying labels to figures. As such, my comments largely remain the same; however, the few changes that were made to this manuscript only bring up additional concerns.

We are somewhat puzzled by this review: Of the lesser comments that we received, only one was wholly new, 17 were repeated from the previous review; including one that was a request to change wording that was already implemented. Additionally, among the comments below, it seems that some portion of the SI (including the sensitivity analysis) was missed. We hope that in this response, you will be able to find all of those changes, and see that your comments were well considered. Unfortunately, space limitations and the need to focus on inertial rejection, prevented us from discussing all of them within the text of the manuscript.

We want to further clarify that your contributions were quite helpful. At the heart of many of the comments was the concern that the CFD may not be sufficiently accurate. To address that we added references that provide some degree of validation, but more importantly ran a sensitivity analysis, which due to space limitations was placed in the SI, and we included a summary discussion of it in the main text because it is so important. We apologise if we did not clearly highlight our rationale and interpretation of your concerns in our response.

We've reproduced our previous responses to the repeated 17 comments, but in some places you will see additional text if we identified clear gaps in our response.

Similarly, the major comments are repeated from the previous version, with the exception of two sentences. We have focused our response to these new additions (**bolded** for ease of reading).

Major comments:

1. This analysis ignores the effects of aeroelasticity on the gust response and attributes the gust rejection solely to the preflex mechanism.

Our previous response focused on the initial statement that we solely attributed gust rejection to the preflex. We assume that we have addressed this, and instead assume that the major comment concerns aeroelasticity.

1. (CONTINUED) The reduction in peak vertical velocity is calculated by comparing the results of the in-flight (aeroelastic) bird data to a completely rigid CFD model. Wing and feather flexibility will play some, if not a large, role in the vertical translation/acceleration. Aeroelastic deformations are even observable in the S1 movie. The authors cannot attribute the rejection they are calculating solely to the preflex mechanism. The only way to numerically quantify the role of preflex from this CFD analysis would be to make turn the CFD model into an FSI model which accounts for the role of aeroelasticity.

However, accurately modeling the aeroelasticity of a complete bird is an entirely different challenge in and of itself which I would not recommend. The simpler solution would be to report your observations

directly from your experimental measurements. **At the VERY LEAST, the authors should have addressed the role of aeroelasticity in their revised paper's discussion even if they cannot quantify how much it effects the aerodynamic rejection.**

The effects of aeroelasticity would be part of the aerodynamic rejection, but as you state, we cannot parse it apart from other wing morphing effects. Hence, we do not go into much discussion about aeroelasticity, as we cannot say much directly about it. We have already acknowledged that feather aeroelasticity contributed to the aerodynamic rejection:

LINE 292: Despite a multitude of possible aerodynamic rejection mechanisms, inertial rejection increases more rapidly than aerodynamic rejection. The most likely aerodynamic rejection mechanisms that could reduce the experienced vertical force/impulse include: ... 2) changes to wing shape, where feather bending modifies wing camber and angle of attack...

In an effort to satisfy the referee, we have modified our discussion of “the delay in aerodynamic rejection” to specifically refer more to aeroelasticity, but also speculate why the effect may be less pronounced in this study.

Line 302: Aerodynamic rejection did undergo an immediate rise, similar to inertial rejection, but it saturated before it had an appreciable affect (~0.05 m/s; Fig 3C) and maintained that saturated level until around the instant of peak inertial rejection. This saturation may have been a response to soft stall, a feature found in many bird wings and observable in our simulated lift curve from CFD (Fig S5.; 30). Soft stall describes a feature of the lift curve, where the lift coefficient becomes insensitive to changes in angle of attack, *i.e.*, at high angles of attack, the curve forms a nominally flat horizontal line. The majority of the mechanisms discussed above reject gusts by means of reducing the lift coefficient through reductions in angle of attack. We hypothesise that the delay in aerodynamic gust rejection persists until each wing's lift coefficient becomes sensitive to changes in angle of attack, *i.e.*, the wing is no longer stalled. This difference in lift coefficient sensitivity may explain why aeroelastic feather bending, which confers nearly instantaneous load alleviation/gust rejection under small perturbations (31), did not have a more pronounced effect in this study. Future work could explore the relative contributions of inertial and aerodynamic mechanisms under weaker gusts that do not produce stall.

We feel rather constrained in our response as we didn't want to speculate too much about the complexities and interactions involved in aerodynamic rejection for the same reason laid out by the referee; we cannot parse them apart. That said, it makes little sense to specifically refer to one mechanism more than another, except as an example, and an acknowledgement that aeroelastic feather bending should be nearly instantaneous, the same with inertial rejection, but there is little more that we feel comfortable adding.

2. The CFD analysis has no validation. The authors have attempted to justify the validity of their results by claiming that the lift and drag is similar to other reported values, but the paper cited does not include data from an owl and the other species tested in that paper have a wide spread of lift and drag results (ranging from Cl max of 0.8 to 1.4). This is neither validation nor justification of the use of CFD. Comparison to other lift/drag data specifically from an owl would be needed.

We previously addressed this comment by adding validating information for our CFD:

Line 551: The output of the CFD simulations was sufficient to provide weight support, produced a wake similar to published wakes for *T. alba*, and produced a polar similar to those published for other birds. Lift accounted for 94.2% of body weight after accounting for acceleration. The wake, as visualised by Q-criterion (34), was similar to wakes measured behind the same individual, and the spanwise downwash distribution was also qualitatively and quantitatively similar to that measured using particle tracking (33, 35). Finally, the simulated lift-drag polars were consistent with other published data (30). Importantly, they exhibit similar values for coefficient of lift (~0.8) at high (>20°) angle of attack, for which much of the glide model operates; but note our drag values are larger due to the inclusion of the body (Fig S5).

We also noted that the range in CL_max data published by Withers (1988) (~.8 - ~1.1) was half of what was stated in the comment (~.8 - ~1.4). This substantially reduced range suggests the aerodynamic polars among birds are much more similar and our statement comparing our polar to those published is much more reasonable.

2. (CONTINUED) To make matters worse, the revised paper added that the "CFD simulations...produced a wake similar to published wakes for T. alba, and produced a polar similar to those published for other birds." without citing what data is being referred to.

The first sentence lists an array of reasons, but each argument is backed up with an explanation and/or citation in a following sentence. Citation 31 and 32 provide the measured wake of *T. alba*, citation 33 (Withers 1988) provides published polars of many birds that are similar in profile and magnitude to what we computed from CFD.

Line 551: The output of the CFD simulations was sufficient to provide weight support, produced a wake similar to published wakes for *T. alba*, and produced a polar similar to those published for other birds. Lift accounted for 94.2% of body weight after accounting for acceleration. The wake, as visualised by Q-criterion (34), was similar to wakes measured behind the same individual, and the spanwise downwash distribution was also qualitatively and quantitatively similar to that measured using particle tracking (33, 35). Finally, the simulated lift-drag polars were consistent with other published data (30). Importantly, they exhibit similar values for coefficient of lift (~0.8) at high (>20°) angle of attack, for which much of the glide model operates; but note our drag values are larger due to the inclusion of the body (Fig S5).

3. Although the title claims that this reflex mechanism acts as a suspension system, this is neither quantitatively or qualitatively defined in the text. Their only justification for why this acts as a suspension system is because "the bird consistently maintained a stable trajectory of its head and torso". But that's not the definition of a suspension system. Luckily, suspension systems are very easily defined mathematically and thus, the only way to quantitatively claim this is to use the results to mathematically model the owl as a suspension system which is done using a mass-spring-damper analysis. Otherwise, you have not actually proven the claim stated by the title of your paper. You could, alternatively, change the title and remove comparisons to a suspension system.

The mass-spring-damper model is an implementation of a suspension system. The goal of a suspension system is not to replicate mass-spring-damper dynamics, but to reduce perturbations. Modern cars have actuators as part of the suspension system; the actuator is not there to reproduce an oscillating spring but smooth the perturbation. The function and purpose of a suspension system (smoothing) is therefore different than its implementation (spring-damper).

We've added a sentence to our conclusions, to explicitly state that the wings performed the same role as a suspension system in a terrestrial vehicle to clarify that we do not mean that they replicate the design of a suspension system in terrestrial vehicles.

Line 321: The hinged wings performed the same role as a suspension system in a terrestrial vehicle, by dramatically reducing the perturbation applied to the body.

Our previous response to this was:

Suspension refers to mechanical isolation (pragmatically closer to mechanical insulation); specifically, systems in which a main mass is isolated from other elements through a linkage that allows for relative motion. These other elements take the brunt of bumps and jolts from the surface (or medium) with which they are in contact, thereby delaying and/or softening the transfer of load to the main mass. The spring-damper system is indeed a common example of a suspension system. In our case, the wings provide isolation for the torso from aerodynamic perturbations, as shown both qualitatively and quantitatively.

The suspension system in birds is linked through muscle, a tissue that acts as a spring and damper in parallel (and in series). So, while we demonstrate mechanical isolation, our system also meets the expectations for a classical suspension system as well.

General Comments:

Only one long comment was wholly new and previously unaddressed. We have parsed that into smaller comments for clarity. We examined replicated comments to assess if our response was incomplete, and we added additional thoughts where we felt we needed to expand; otherwise we replicated our previous response.

Line 270: The authors have attempted to discuss the potential sensitivity of their main conclusion (specifically that the inertial effects are 'rapid and strong gust rejection mechanisms') to errors in their CFD simulations. First and foremost, a proper error or sensitivity assessment is not meant to determine whether or not the main conclusion is categorically true or false; instead, it is meant to quantify by HOW MUCH the data and/or conclusion is affected. I would also argue that the findings regarding the aerodynamic rejections are just as important as the inertial rejections (the data in Fig. 3 shows that the aerodynamic rejection substantially exceeds the inertial rejection, so perhaps it is actually more important?).

Within the SI we elaborated upon the sensitivity values (the "how much"); however, for the primary discussion, due to space limitations, we focused upon what we concluded from that analysis.

We are not stating that inertial rejection is absolutely more important than aerodynamic rejection at any point other than early gust rejection. The two work together, and interestingly at different time scales;

prior to this work, this was not appreciated/known. Future work, will elucidate why these effects are complementary in time.

Secondly, no documentation or data from this assessment is included with the manuscript's supplementary material.

Rest assured, we did add this to the SI. It is unfortunate that you did not see it, as it's key to our response to many of your concerns about possible error in the CFD. We have produced it below (Line 748 in the previous submission, with 'track changes' accepted):

Line 783: *Sensitivity of gust rejection to potential error in the aerodynamic polar*

Our results are insensitive to systematic error in the CFD-derived aerodynamic polar. For this analysis we only examined the average relative contribution of inertial and aerodynamic rejection at the instant of peak inertial rejection. A 50% increase in drag decreased inertial rejection from 32% to 29% and increased aerodynamic rejection from 6% to 15%. The inertial rejection was still double aerodynamic rejection. Similarly, increasing the lift curve slope by even 20% over the linear portion ($<10^\circ$) decreased inertial rejection by just 2% (to 30%) and increased aerodynamic rejection by 6% (to 12% of the total rejection). Increasing lift across the whole aerodynamic polar demonstrates that it is more sensitive to results at higher angle of attack, but even applying a moderate increase still did not affect our conclusions. We applied a 6.5% increase in lift coefficient—the amount required to achieve 100% weight support in the CFD simulations at 5 degrees angle of attack (the observed orientation and at the observed speed), but treated as systematic error across the whole aerodynamic polar—and this decreased inertial rejection from 32% to 27% and increased aerodynamic rejection from 6% to 21%. Even with this systematic increase in lift applied across the polar, inertial rejection was still 30% more effective than aerodynamic rejection at its peak.

Furthermore, drag is frequently associated with errors substantially larger than 10% because the magnitude of drag forces is so small, so this does not seem like an appropriate % increase to test. It also seems unusual that the sensitivity analysis done for a 20% increase in lift but only a 10% increase in drag. Drag has high uncertainty, but it is unclear what value would be most appropriate. As we wrote in our previous response, when we increased y^+ in our CFD simulation, we reduced our drag. Applying a large decrease in drag would produce implausible results, so instead applying a modest 10% increase seemed reasonable for the sensitivity analysis. However, we agree that the error could be much more substantial than 10%, so we repeated the analysis with a 50% increase in drag. The change is reflected in the SI:

Line 786: A 50% increase in drag decreased inertial rejection from 32% to 29% and increased aerodynamic rejection from 6% to 15%. The inertial rejection was still double aerodynamic rejection.

Lastly, the conclusion that inertial rejection is "insensitive to our aerodynamic model" is arguably common sense. Since you were assessing the effects of errors in your aerodynamic loadings, it would have been more appropriate to assess the changes in aerodynamic rejection. Discussion on the sensitivity to the aerodynamic rejection is almost entirely ignored which is where you would logically see the largest impact of increasing the lift and drag values.

We agree that it would be common sense that absolute inertial rejection would be insensitive to the aerodynamic model. As we wrote on line 281, in our sensitivity analysis:

None of these changes affected the absolute magnitude of inertial rejection, but they affected the quasi-steady glide model, which in turn determined the relative magnitudes of inertial and aerodynamic rejection.

That is why our analysis concerned the relative effect of inertia and aerodynamics. The SI goes into the changes to both. In each of the three cases, inertial rejection exceeded aerodynamic rejection, at the moment of peak inertial rejection:

Line 278: At its peak, inertial rejection exceeded aerodynamic rejection regardless of whether we applied a 50% increase in drag across the drag curve, a 20% increase in lift to the linear portion of the lift curve, or a moderate 6.5% increase in lift across the entire lift curve (detailed in supplementary text).

The discussion assesses whether errors in the CFD-derived polar would change the conclusions of our study, which lies at the fundamental core of your questions about the CFD model elsewhere and in the previous questions.

--- The comments below were all responded to in our previous submission, but were repeated by the referee...

“Implementing a similar reflex mechanism in future aircraft...”. Phrasing is important here, and ‘future aircraft’ to the public implies transonic aircraft. The reflex shown here undergoes large scale displacement which would almost certainly be unsuitable for transonic aircraft. It’s important to clarify that the target application is (presumably) small scale aircraft or UAVs.

This text did not occur in the revised submission. It already read as: future small-scale aircraft (at lines 39, 337, and 340).

Fig. 1B: The measurements at the peak of the gusts appear to have no error bars. Is this because they are small or missing? Similarly, the error bars on the edge of the gusts are quite large (up to 0.3-0.4 m/s (or upwards of 10-15%) at a distance of 0.4m into the gust) and even overlap with the error bars of the data points for the other flow velocities, yet no discussion of error in the analysis is really presented. The authors report an n of 6 (presumably for the error calculations) but in the methods section they state that anemometer measurements were taken at 50 hz for 30 s which totals 1.5k measurements. This is confusing. Shouldn’t the latter number be used in error calculations? Lastly, the methods section states that the velocity measurements were collected in a 5x6 grid, so why is only one cross sectional profile shown? The full velocity could be plotted as a 3D contour map and would be more informative.

Our previous response missed a question by the reviewer:

The error bars at peak gust are small enough that they are hidden behind the size of the marker. We don’t think it requires an additional sentence in the caption stating that fact, but do appreciate that it’s something to examine in review, as errors do happen.

We respect your desire for more information about the gust, some of which is included as SI (since it’s not critical to understand the study). We maintain that our approach to displaying and discussing the gust is more informative, relevant, and less confusing than showing the entire measurement array. Our

quasi-steady glide model is perturbed by the span-averaged flow, which is another reason that we display the data as we have.

Our previous response

We present the span-averaged flow, which is produced from 6 time-averaged measurements. We would argue that the error of interest is not in the flow around a small volume, but the flow experienced by the bird. At an instant in time, the anemometer samples a trivial volume of flow relative to the bird. The spatial average of the flow is closer to that experienced by the bird and therefore more meaningful.

Figure 1C: The leading edge of the point clouds and the corresponding mesh have a significant amount of rough edges and features. Are you confident that these represent physical features observed on the bird or are they the result of errors in the 3D reconstruction? This is important because leading edge protrusions are well known to affect the flow, for example by tripping the flow from laminar to turbulent. This could affect the results of your CFD analysis.

Our previous response:

There may be possible imperfections in the mesh at the edges. Error at the edges could affect the specific details of the flow around the wing. Despite these potential errors in flow detail, the lift produced essentially supports body weight, the polar is similar to those observed in other birds, and the wake is similar to that observed from flow visualization.

This was explained in the methods (line 551-557):

The output of the CFD simulations was sufficient to provide weight support, produced a wake similar to published wakes for *T. alba*, and produced a polar similar to those published for other birds. Lift accounted for 94.2% of body weight after accounting for acceleration. The wake, as visualised by Q-criterion (34), was similar to wakes measured behind the same individual, and the spanwise downwash distribution was also qualitatively and quantitatively similar to that measured using particle tracking (33, 35). Finally, the simulated lift-drag polars were consistent with other published data (30).

More detail is needed about the CFD model earlier on in the text. Is it a static or dynamic model? Is it rigid or flexible? What turbulence model was implemented? Some of these details are later discussed in the methods section, but the reader needs a better understanding of the CFD was performed before introducing your results so they can assess them properly.

Those details are now all discussed within the methods. The CFD is one component of a larger experiment and we maintain that the detail should be reported within the methods section. We did make substantial changes to the introduction to expand upon the role of CFD in this study (as well as the glide modelling, the CT scan, and kinematic analysis).

Figure 2: It would be useful to superimpose the gust profile onto these plots so the reader can compare the velocity, elevation, and pitch response relative to the spatial strength of the gust. Furthermore, there is a fair amount of variability between test runs, so it would be useful to, for example, plot shaded error bands on the plotted averages.

Our previous response:

The plots are cropped to contain the kinematics during the gust, and do not cover any region beyond this. We have added to the caption to better orient the reader to this:

Line 133: (B-D) Presented kinematic data encompasses the space occupied by the '1-cosine' gust, and does not extend beyond it. (C,D) Kinematics represent the average of the two wings.

We chose not to add the gust velocity in the background. The plots are already quite busy visually and adding to them seems unnecessary when they only contain the '1-cosine' profile of the gust. We are open to it and will leave it as a decision for the editor.

And:

We avoid error bands when we cannot justify arguing for a normal distribution. We argue that it is more informative to see the raw data to assess variation between trials in this case.

The conservation of moment calculations would be useful to perform here to physically show the reader how this is true. However, I believe you also need to include the momentum of the fluid especially since you are assessing a dynamic gust.

In the latest revision, we have added a section to the SI that covers the basic physical reasoning and maths behind the calculation (Line 706 – 758).

We also discussed the effect of added mass (line 279):

Our previous response:

The fluid mass is an interesting point that we had neglected here. Using the common assumption that the added mass of air is that of a cylinder about the wing chord (Fung 1993, An introduction to the theory of aeroelasticity), we estimate an added mass of 10g. We ran a calculation where we added 10 grams of mass to the wings, the results did not change the conclusion of the study: the inertial effect was enhanced and average gust rejection at peak was 42%. We have added:

Line 286: we have conservatively estimated the effect of inertial rejection by ignoring the added fluid mass around the wings; accounting for the added mass would have further enhanced inertial rejection.

The use of the word 'impulse' throughout the text is misleading. It appears that this calculation is simply the difference between the CFD and the experimental results. Impulse is defined as the force acting over time, but it does not appear that the authors performed time integration to actually calculate the impulse.

Our previous response:

It is indeed impulse. We present mass-normalized impulse on plots, which is equivalent to a change in velocity. This method of normalization is convenient as it is far more intuitive than scaling it to body weight. We have edited the manuscript throughout to be consistent and clear on this point, as well as add additional SI to further explain the approach.

This refers to the same SI as mentioned in the comment directly above (Line 706 – 758).

“At this stage...” “Later, as the wing pitched down...”. It would be useful to mark these stages on the plots to direct the reader exactly to where these trends are occurring.

Our previous response:

We have edited the text to be more precise.

Line 161: At the instant of peak inertial rejection, ...

Line 163: At this same instance, ...

Line 111: This is the only time that the suspension system is discussed somewhat qualitatively. The kinematic characteristics or mathematical representation of a suspension system aren't even defined in the introduction or analysis. Furthermore, this data does not adequately demonstrate that the reflex mechanism acts as one. To truly prove that this mechanism acts a suspension system, you should use your experimental results to model it as a mass-spring-damper system which mathematically describes a suspension system.

Our response to major comment 3 addresses this.

“As the wing pitched downwards, there was an increase in the difference between the measured vertical velocity of the whole bird center of mass and the rigid body simulation, indicating large aerodynamic-based gust rejection of $48 \pm 4\%$ of the potential impulse (Fig 3c).” The authors cannot confirm that this is due to the translation and rotation of the wings. The wing and feather aeroelasticity plays a substantial role in mitigating gusts, which is not accounted for here since the calculations are performed by taking the difference between the rigid CFD model and the flexible in-flight experiments. *At the very least, the authors could acknowledge that the effects of aeroelasticity is present and that it's effects can't be isolated.*

Our previous response addressed the replicated portion of this comment.

Our previous response:

Agreed. The quote above merely refers to the timing of aerodynamic rejection and the timing of wing pitch: the two events happen in relative synchrony.

We address the added final sentence in major comment 1.

The authors state that “birds may then be able to tune the dynamics involved with gust rejection by modifying the stiffness of their shoulder joint”. If the wing does indeed act as a suspension system as the authors claim, then changing the stiffness of the shoulder would effect the timing of the inertial rejection (dictated by mass-spring-damper equations of motion). Stiffening the shoulder muscle would cause the inertial rejection to reach it's maximum sooner. This is contradictory the statement made earlier, claiming that the inertial response must be passive since it is not faster than the reaction time measured in other studies. That is, unless the shoulder stiffness is assumed to already be at a maximum for these in-flight tests.

Our previous response:

Birds can modify the stiffness of their shoulder through agonist-antagonist interactions. Tuning the dynamics would be a 'feed-forward' approach. This may be feasible as turbulence is more predictable in certain environments.

Figure 3: The results in Figure 3 are only presented for one gust intensity (medium), right? Do these results change at the lower and higher intensities? Since this was only assessed at one gust intensity, I don't think you can conclude that “passive mechanical properties of the shoulder joint and wing are

responsible for immediate gust rejection” because this time frame was “equal to the minimum behavioural reaction time measured in other bird species”. The data shown in Figure 2c indicates that for the high gust intensities, the wing elevation changes more rapidly (the response onset is sooner and the slope is steeper) but you have not yet proven that this difference between gust intensities is only due to the aerodynamic rejection.

Our previous response:

All of the data are presented together. We have added a statement to make this clear:

Line 158: This inertial effect reduced the impulse to the torso (Fig 3B). The dose-dependent kinematics resulted in wing inertia rejecting a greater magnitude of the gust impulse with increasing gust intensity, but the proportion of the total expected impulse rejected by wing inertia did not discernably differ across gust intensities (Fig S1).

The new supplement figure is presented below.

The surface accuracy of the reconstruction is said to be 1.67mm, however there are many features of the bird that will be well below this thickness (namely, the trailing edge feather thickness). How does that effect your reconstruction?

Our previous response:

Our surface accuracy was -0.68 ± 1.67 mm (mean \pm s.d.; where 50% of error < 1.1 mm & 95% < 3.5 mm) for a fibreglass bird model. This does not necessarily explicitly limit the thickness that we can resolve. We measured the upper and lower surface of the bird independently, the two must then be combined. Combining without smoothing could create places where the two surfaces penetrate or move apart, so we smooth the geometry adjusting parameters to prevent any shapes that are physically implausible.

Are the mass properties segmented into each wing section (i.e. handwing, armwing etc.) or is the mass tracking done by translation and rotation of the whole wing mass? This could yield very different results if the whole wing is treated as one lump mass that can only undergo translation/rotation. Upon looking at the supplementary material, it appears that the mass is recorded for each voxel, so is each voxel mapped

to the deformed bird geometry? Basically, are you calculating the mass distribution of the morphed wing by translating and rotating each voxel, or by translating and rotating each wing segment?

Our previous response:

We have added some additional information to the methods to clarify this. We used rigid-body transforms to track the centre of mass. These of course are only valid if the wing does not deform substantially. We have added an SI video that displays the wings from each glide pre-gust and at the moment of peak inertial rejection. The outlines all generally look similar despite the imposed gust on half of them. We concluded therefore that wing deformation effects on the centre of mass are negligible. Beyond the moment of peak inertial rejection, this statement begins to lose validity, but that has little bearing on our conclusions.

Line 476: The segmented torso and left wing from the scan were aligned with the point clouds to estimate the mass distribution of the bird in flight. We assumed that the inertial dynamics of the left wing were equal to those of the right wing. The CT and point cloud alignment was performed for a single typical glide posture to establish the mass and centre of mass position for each point cloud segment. Centre of mass position was then tracked subsequently when aligning point clouds using rigid-body transforms that allowed for rotation and translation. This approach was ideal for the duration that wing flexion was negligible (e.g., the time-course dominated by the inertial rejection mechanics. Movie S3). However, the effect of wing shape change on the centre of mass was not completely ignored because we allowed the wing to translate. For example, when the elbow flexed, the rigid-body transform that best fit the flexed state of the wing moved the centre of mass proximally as expected.

“The model was free to accelerate both horizontally and vertically, but could not rotate, as the real bird displayed no tendency to pitch during the gust.” This is a major assumption. The lack of observation of the bird pitching during a gust does not mean that, in the absence of wing motions, the bird would not pitch. The wing’s reduction in angle of attack is likely an active mechanism to, in part, minimize the pitching moments caused by the gust. When the gust hits the nose end of the bird first, this would cause a nose-up pitching moment. This has been shown in similar experiments done with gliders, so it is not accurate to assume that there would be no pitch in your rigid model.

Our previous response:

There would be pitch in a passive rigid model; we do not suggest otherwise. Our investigation is in the linear motions. We mention that the bird did not pitch, because if the bird did pitch and the model did not, then we would be neglecting those rotational effects on the aerodynamics. Your comment about wing pitch and pitching moment are interesting and will be considered in detail in future work.

The authors justify their turbulent model by claiming that it “is appropriate for bird flight simulations that require a high accuracy boundary layer”; however, their first inflation layer near the wall has a y^+ value of 10 which does not resolve the viscous sublayer. Y^+ values of approximately 1 are recommended for best results using low Re (k-omega) models. The authors report a difference of 3% in lift between $y^+ 10$ and $y^+ 3$ results, but there would presumably be an even larger error between $y^+ 10$ and $y^+ 1$. And how

does this affect the drag error? I appreciate the added details about convergence that the authors added in the resubmission

Our previous response:

We have run four simulations at $y^+ \sim 10, 7, 3,$ and 2 at the observed gliding orientation and speed (angle of attack: 5° ; speed: 7.8 m/s). Linearly extrapolating out to $y^+ \sim 1$, suggests that our approximation underestimates lift by 6.5% (which also adjusts the weight support estimate for this geometry to a fairly reassuring 100.7%), and overestimates drag by 5.5% . Drag plays a relatively minor role in the gusted impulse applied to the bird, providing at most 17% of the impulse (and as little as 5%). But note that drag reduction will cause the rigid-model to be perturbed slightly less, while a lift increase will cause the bird to be perturbed more. Finally, we have also run simulations at 20° angle of attack at $y^+ \sim 10$ and 3 to determine if the error was greater at high angles of attack. Linearly extrapolating the increase in lift with increasing y^+ at $y^+ \sim 1$, we estimate the lift error at 20° angle of attack to be only 3.2% , approximately half of the estimation at 5° .

We now report on the sensitivity of our results:

Line 277: The conclusion that inertial rejection is both a rapid and strong gust rejection mechanism in birds is robust to systematic error in the CFD-derived aerodynamic polar. At its peak, inertial rejection exceeded aerodynamic rejection regardless of whether we applied a 10% increase in drag across the drag curve, a 20% increase in lift to the linear portion of the lift curve, or a moderate 6.5% increase in lift across the entire lift curve (detailed in supplementary text). None of these changes affected the absolute magnitude of inertial rejection, but they affected the quasi-steady glide model, which in turn determined the relative magnitudes of inertial and aerodynamic rejection. The relative impact of inertial rejection and its insensitivity to our aerodynamic model provides confidence in the substantial role played by inertial effects in avian gust rejection. Further, we have conservatively estimated the effect of inertial rejection by ignoring the added fluid mass around the wings; accounting for the added mass would have further enhanced inertial rejection.

And within the SI:

Sensitivity of gust rejection to potential error in the aerodynamic polar

Our results are insensitive to systematic error in the CFD-derived aerodynamic polar. For this analysis we only examined the average relative contribution of inertial and aerodynamic rejection at the instant of peak inertial rejection. A 10% increase in drag decreased inertial rejection from 32% to 31% and increased aerodynamic rejection from 6% to 8% . Similarly, increasing the lift curve slope by even 20% over the linear portion ($<10^\circ$) decreased inertial rejection by just 2% (to 30%) and increased aerodynamic rejection by 6% (to 12% of the total rejection). Increasing lift across the whole aerodynamic polar demonstrates that it is more sensitive to results at higher angle of attack, but even applying a moderate increase still did not affect our conclusions. We applied a 6.5% increase in lift coefficient—the amount required to achieve 100% weight support in the CFD simulations at 5 degrees angle of attack (the observed orientation and at the observed speed), but treated as systematic error across the whole aerodynamic polar— and this decreased inertial rejection from 32% to 27% and increased

aerodynamic rejection from 6% to 21%. Even with this systematic increase in lift applied across the polar, inertial rejection was still 30% more effective than aerodynamic rejection at its peak.

The authors justify their CFD results by claiming that their data is similar to the results presented in [33]. However, this citation does not have any aerodynamic coefficient data for an owl. Furthermore, the lift curves for all of the species tested range from a CL_{max} of 1.4 for the woodcock, to a minimum of CL_{max} of around 0.8 for the swift. This is a huge range of lift curves to compare results to, so it is not sufficient to simply say that your results “are similar to other published data (33)”. *Furthermore, in the resubmitted manuscript, the authors claim that their CFD produced wake distributions and polars similar to other works but did not cite the works to which they were referring to.*

We reference three papers that we use as reference comparisons for our CFD and CFD-derived polar (30, 33, 35). They are referenced as we go through each of the three topics that we introduced in the first sentence of the paragraph.

Line 551-559. The output of the CFD simulations was sufficient to provide weight support, produced a wake similar to published wakes for *T. alba*, and produced a polar similar to those published for other birds. Lift accounted for 94.2% of body weight after accounting for acceleration. The wake, as visualised by Q-criterion (34), was similar to wakes measured behind the same individual, and the spanwise downwash distribution was also qualitatively and quantitatively similar to that measured using particle tracking (33, 35). Finally, the simulated lift-drag polars were consistent with other published data (30). Importantly, they exhibit similar values for coefficient of lift (~ 0.8) at high ($>20^\circ$) angle of attack, for which much of the glide model operates; but note our drag values are larger due to the inclusion of the body (Fig S5).

Our previous response:

We have further validating information for our CFD:

Line 553: Lift accounted for 94.2% of body weight after accounting for acceleration. The wake, as visualised by Q-criterion, was similar to wakes measured behind the same individual, and the spanwise downwash distribution was also qualitatively and quantitatively similar to that measured using particle tracking (33, 35). Finally, the simulated lift-drag polars were consistent with other published data (30).

Concerning the variance across species' lift-curves:

Withers' plots are somewhat challenging to follow. The woodcock actually reaches peak C_L around ~ 1 (1.05?), as seen most clearly on the left plot (also visible on the right plot, but it can be easily confused for the C_L/C_D curve.).

The nighthawk and quail are the best performers, but they only reach CL_{max} of 1.1 to 1.05 (depending on whether you fit a curve through the data or accept each point as true). This does not dismiss this point, but it does moderate the perceived variance in the data.

Stimulated by this concern, we have added a discussion of the sensitivity of our results to the aerodynamic polar (two comments below). The greatest variation in Withers' polar data is at the lowest angle of attack, and our results are insensitive to that portion of the curve.

Referee: 1

Comments to the Author(s).

A good idea. Changes are acceptable.

Referee: 3

Comments to the Author(s).

Review for RSPB-2020-1748-Proof-hi

The manuscript is much improved, and I have no major concerns with the methods or conclusions. I do have quite a few comments to improve the clarity of the text, most of which are suggestions.

One overarching theme of those comments concerns the use of “trajectory” and “path” in the text. If possible, I would suggest removing these words wherever possible and focusing on impulse. It would also help to concisely explain why impulse is used to quantify gust rejection, rather than position, as early as the introduction. Maybe it is just me, but the plots in Figure 2B and Figure 3 evoke in my mind the flight paths of a bird through an upward gust. With the references to trajectory and path, it is hard to break away from viewing these plots as flight trajectories.

The appropriate metric for describing perturbation and rejection are, at some level, arbitrary (they all relate), but at the level of intuitive interpretation quite distinct. It is an issue that we have spent some time considering during this study. We have expanded the SI with some of those considerations, as a response to one of your concerns. However, trajectory/path are critical, and we do not want the reader to forget their importance. Near an obstacle deviation may be life-or-death, so we do think it is important that we

maintain the general framework of maintain a path/trajectory, even if we quantify it with a time-derivative.

To address this comment though, we have emphasized impulse as the selected metric now in the introduction (line 81), discussed in greater detail the rationale for considering impulse over position and acceleration in the SI (line 708), and still remind the reader of it in the first paragraph of the results (no change, but at line 120).

Thank you for addressing my comments from the previous version! The supplemental is much easier to follow and the text is much clearer.

Specific comments:

-Line 36 – Specify that the center of percussion and mass are for the wing(s). The torso and head mentioned last, which may cause some confusion.

We clarified that by:

Line 36: For each wing, the centre of aerodynamic loading aligns with the centre of percussion, consistent with enhancing passive inertial gust rejection.

-Line 114 – Unclear what “proportion” refers to. Do the movements change in proportion relative to one another, or are they both proportional to gust intensity? Maybe “Both movements are likely to be important for gust alleviation and increase proportionally with gust intensity.”

-Line 188 – Change to “The white dot” and add a period at the end of the sentence.

-Figure 2 – I think it would be helpful to indicate that vertical impulse, wing elevation, and wing pitch plots are relative to pre-gust values. Including a delta symbol on the axes’ labels and a brief comment in the figure caption would accomplish this. To help the general reader understand wing pitch and wing elevation, consider adding a small cartoon version of each response to the corner of plots 2B through 2D. We implemented those three suggestions. Great idea on the cartoon in the corner.

- Was the plane fitted to the top or bottom surface of the wing? What software was used segmentation and for the ICP Alignment method?

We added that the segmentation software was custom-written Matlab GUIs, and that ICP is an in-built function of Matlab.

We removed the word “surface” to clarify that the fit was to the whole reconstructed wing. We only reconstruct the surface, but here that specificity created confusion.

Paragraph at Line 66 – Two references are made to the glide path, but you have focused on the vertical impulse rather than the position of the bird. I think it would help if you explained how/why impulse was used to measure gust rejection. Right after the sentence ending in “ten high-speed video cameras,” you could add something like, “to quantify gust rejection, we compared the change in vertical velocity, hereafter impulse...”

Suggestions to improve the clarity of this paragraph:

“Here, we investigate how birds cope with rapidly changing airflow by studying the flight of a barn owl (*Tyto alba*), tracked using high-speed video-based 3-dimensional (3D) surface reconstruction, through a range of fan-generated vertical gusts. To successfully reject a gust perturbation, a bird must maintain the velocity of its torso and head through wing morphing, i.e. changing the shape and posture of their wings.

We quantified how well the barn owl rejected the controlled gust through inertial mechanisms by comparing the impulse (change in velocity) of the barn owl's torso and head to that of its center of mass, which includes the mass of the torso, head, and wing. In other words, we investigated how the movement of the mass of the wings mitigated the impulse experienced by the torso and head of the animal through the conservation of momentum. We quantified the aerodynamic effects of wing movement by contrasting the glide results of the live articulated bird to a model of the expected glide path for a rigid bird, with an aerodynamic polar computed from CFD (Fig 1D)."

We think/hope that the two instances of glide path in this paragraph are useful for maintaining the bigger picture of the study. However, this paragraph is an excellent place to emphasize the importance of impulse:

Line 80: We quantified the effect, and rejection, of the perturbation over time using mass-normalised impulse (change in velocity).

Thank you for the time that you put into the paragraph, it is excellent and accurate. We are hesitant to make such a large change to a critical paragraph at this stage, but we appreciate the thoughtfulness.

Line 114-116 – Wing elevation also modifies the lift generated by the wings. You have done a great job at addressing the concerns of reviewers RE: the importance of wing pitch relative to other factors, but you are less successful here. I am guessing that the aerodynamic effect of wing elevation is also substantial. The effects of wing pitch might be more substantial. Either way, it might help to use the data you have from your CFD polar and data on wing elevation/pitch to roughly estimate the contribution of each mechanism. If the wings elevate up to 30 deg, that should be about a 15% drop in the vertical component of the lift vector ($\cos(30 \text{ deg}) = 0.866$). The change in lift based on wing pitch depends on the initial angle of attack but could be much larger based on your CFD-based lift polar. Pointing that out with some numbers might help placate reviewers/readers.

With that in mind, it might be interesting/convincing to include a plot of angle of attack vs. distance into the gust, accounting for the upward velocity of the gust. This is not required, but it might be convincing if the owl maintained a consistent angle of attack by pitching its wings forward despite the perturbation.

Wing elevation will contribute to aerodynamic rejection, so we have added that it aids wing pitching to reduce the lift generated. We wish to keep the emphasis on the inertial rejection, although the aerodynamic rejection is also interesting, and we will follow up with a later study modeling some of the aerodynamic phenomena. Here, we suspect it would lead to many questions just due to the many ways that we think about wing morphing: changing thickness, pitch, camber, twist, wing area, etc..

Line 117: Both movements are likely to be important for gust alleviation and increase proportionally with gust intensity, with wing elevation delivering a rapid rejection of the gust due to inertial effects, and also aiding the delayed wing pitching to modify the lift generated by the wings.

Line 295 – The text in this paragraph is not clear, at least to me. If space allows, it would be helpful to expand on this section.

We've rewritten this section. Initially, we had hoped to avoid going into too much detail about specific mechanisms, and spoke abstractly. Now it reads with more specific, but speculative details, and we have added details about aeroelasticity based on the request by reviewer 2.

Line 302: Aerodynamic rejection did undergo an immediate rise, similar to inertial rejection, but it saturated before it had an appreciable affect (~ 0.05 m/s; Fig 3C) and maintained that saturated level until around the instant of peak inertial rejection. This saturation may have been a response to soft stall, a feature found in many bird wings and observable in our simulated lift curve from CFD (Fig S5.; 30). Soft stall describes a feature of the lift curve, where the lift coefficient becomes insensitive to changes in angle of attack, *i.e.*, at high angles of attack, the curve forms a nominally flat horizontal line. The majority of the mechanisms discussed above reject gusts by means of reducing the lift coefficient through reductions in angle of attack. We hypothesise that the delay in aerodynamic gust rejection persists until each wing's lift coefficient becomes sensitive to changes in angle of attack, *i.e.*, the wing is no longer stalled. This difference in lift coefficient sensitivity may explain why aeroelastic feather bending, which confers nearly instantaneous load alleviation/gust rejection under small perturbations (31), did not have a more pronounced effect in this study. Future work could explore the relative contributions of inertial and aerodynamic mechanisms under weaker gusts that do not produce stall.

Line 111-114 – Briefly define wing pitch and wing elevation. Alternatively, you could include a cartoon/drawing in Figure 2 with arrows to illustrate wing pitch and elevation. Currently, a lot is going on in Figure 2A and in Movie S2, so it is not clear to a novice reader what movements are elevation versus pitching (e.g. in pitching, the back of the wing “elevates” and in elevating, the wings pitch toward the midline).

We've added a brief description of each movement. Combined with the suggested addition to the methods and the current figures, we think the two motions will be now much clearer.

Line 113: The bird achieved this, in part, by two dramatic movements: an almost instantaneous change in wing elevation (Fig 2A,C, Movie S2), where rotation about the shoulder caused the wing to rise with the gust, followed by a substantial reduction in wing pitch angle, where wing rotation was about its long axis (Fig 2D).

Line 144 – Add a statement explaining why impulse, rather than position, was used to quantify gust rejection. Both “perturbation” and “movement” in this paragraph suggest that you tracked position instead of impulse. This would also be a good place to specify that velocity was calculated at the center of mass for the wing, torso/head, and whole bird, which I assume was the case.

We don't have the space in the main document to elaborate upon why we selected mass-normalised impulse (Δ velocity) over Δ position/acceleration/jerk. As impulse is an intuitive quantity for any object which has a force applied to it over time, we think this addition would be appropriate to the SI:

Line 707: We quantified the gust perturbation by its delivered impulse or, after mass-normalisation, the added vertical velocity. The effect of the perturbation could also be described by other metrics, such as a change in position or instantaneous acceleration. Impulse is clearer than position as it is not affected by variation in initial conditions, such as flight velocity. We did not select instantaneous acceleration because it neglected the previous effects of the perturbation. Expanding on this, the gust causes the bird's torso to rise with the flow over time, as the torso rises the relative flow experienced decreases as does the perturbation; therefore, minimising instantaneous perturbation can be achieved by performing poorly in previous time-steps, *i.e.*, early in the gust. Impulse accounts for this.

Line 158 – for clarity, it would be helpful to define the instant of peak inertial rejection (e.g. “the point at which the impulse of the torso is most different from the impulse of the center of mass”)

We appreciate the desire for more orientation, but we are not sure it’s needed here. It is important that we are using a standardized instant in time for comparison, but the exact definition of that standard is less so and may distract from the key result that immediately follows.

Line 210 – I suggest introducing this paragraph briefly, if possible. Otherwise, it comes suddenly, given that the center of percussion is not mentioned in the introduction.

We have added a topic sentence to this section to make the paragraph a bit easier to follow:

Line 214: The mechanics that allow for inertial gust rejection depend upon the mass distribution as well as the spanwise centre of pressure; together they determine how each wing moves, and how the perturbation applied to each wing affects the body.

Line 416 – How was this accuracy defined? I asked because the value is negative.

The accuracy is largely intuitive, and the exact definition is convoluted for nonspecialist readers, but we agree that the sign of the measure requires explanation:

Line 435: A negative accuracy implies that the known geometry is larger in one or more dimensions than the point cloud. Most likely, our primary error is along the vertical axis between the upper and lower camera sets, and our reconstruction has estimated the lower and upper surfaces too close to one another.

Also, in Figure 2, both values start at 0, so I think that these are changes in wing elevation and wing pitch from the start of the gust. Is wing pitch/elevation, as included in the supplemental data sheet, calculated based on the horizontal?

We’ve changed the y-axis label to specify that these are changes in elevation and pitch.

Paragraph at 437 – As wing elevation and wing pitch are central components of this work, more specificity would help here. Specify which reference position (currently, the text says “a reference position”). At 0 m into the gust?

We’ve modified the text to state that the rotations are based on a reference “pose” instead of “position”, as it’s just a series of Euler rotations. We have elaborated upon the zero’d definition of each angle with:

Line 465: Wing elevation is described relative to the horizontal plane, wing pitch is described as the long-axis rotation of the wing after accounting for its swept and elevated orientation.

Line 188 – Figure 2B says Vertical Impulse whereas figure 3A says Normalized Impulse. I would suggest using “Normalized Vertical Impulse”.

Agreed. Figure 2B should state “Normalised vertical impulse (m/s)”.

Paragraph at 402 – Is this python script available upon request? Without it, I think it would be difficult to repeat the methodology described in this paragraph from the text alone. Does this python script take advantage of any specific libraries (OpenCV)? If so, please reference those libraries.

Yes, it is available, but unfortunately, the gradual improvements to PhotoScan, which has now rebranded as Metashape, have meant that the script is no longer functional in the modern software and would require chunks be rewritten to implement. We would offer it, but it would be almost misleading as it would

require an older version of the software. More importantly, much of what was coded can now be done through their GUI and task manager, perhaps almost as quickly.

Let me know if you have any questions about these comments!

Regards,

Tony Lapsansky

anthony.lapsansky@umontana.edu